# PRIMAL-DUAL DIRECT PREFERENCE OPTIMIZATION FOR CONSTRAINED LLM ALIGNMENT

## ABSTRACT

The widespread application of Large Language Models (LLMs) imposes increasing demands on safety, such as reducing harmful content and fake information, and avoiding certain forbidden tokens due to rules and laws. While there have been several recent works studying safe alignment of LLMs, these works either require the training of reward and cost models and incur high memory and computational costs, or need prior knowledge of the optimal Lagrange multiplier. Motivated by this fact, we study the problem of constrained alignment in LLMs, i.e., maximizing the output reward while restricting the cost due to potentially unsafe content to stay below a threshold. For this problem, we propose a novel primal-dual DPO approach, which first trains a model using standard DPO on reward preference data to provide reward information, and then adopts a rearranged Lagrangian DPO objective utilizing the provided reward information to fine-tune LLMs on cost preference data. (Reviewer kvKV) Our approach only needs to train two models rather than three as in prior works that need trained reward and cost models, which significantly saves memory costs, and does not require extra prior knowledge. Moreover, we establish rigorous theoretical guarantees on the suboptimality and constraint violation of the output policy. We also extend our approach to an online data setting by incorporating exploration bonuses, which enables exploration in the uncovered prompt-response space, and provide theoretical results that get rid of the dependence on preference data coverage. Experimental results on the widely-used preference dataset PKU-SafeRLHF demonstrate the effectiveness of our approach.

## 1 INTRODUCTION

Large Language Models (LLMs) (Achiam et al., 2023; Touvron et al., 2023a;b) have achieved a remarkable success in dialogues, summarization, instruction following, etc. Despite the huge success of LLMs, LLMs may also output fabricated information and harmful content, such as texts involving discrimination, crimes and moral issues (Gehman et al., 2020; Lin et al., 2021; Wei et al., 2023). With the extensive application of LLMs, how to align them to enhance safety or impose constraints has become a crucial problem. For example, we want to prevent LLMs from generating content that may have negative societal impacts or ethical concerns. In Agentic AI or AI education applications, we need to avoid certain tokens due to some rules and laws, or course content that has not been taught.

Recently, there are several works studying the safety alignment of LLMs. A popular formulation is the *constrained alignment problem*, which aims to maximize the reward while constraining the cost to stay below a threshold. Dai et al. (2024) proposed a safe reinforcement learning from human feedback (RLHF) framework for this problem, which trains reward and cost models on reward and cost preference data, respectively, and then applies an RL algorithm to fine-tune LLMs to maximize the Lagrangian function under the learned reward and cost functions. Liu et al. (2024b); Wachi et al. (2024); Huang et al. (2024); Kim et al. (2025) designed direct preference optimization (DPO)-based safety alignment approaches. The idea of DPO is to directly fine-tune LLMs using preference data, without training a reward model. However, these works either still require trained reward and cost models (Liu et al., 2024b; Huang et al., 2024), or need prior knowledge of the optimal Lagrange multiplier (Wachi et al., 2024), or are inefficient in cost information learning (Kim et al., 2025).

Motivated by the above facts, we propose a novel and provably efficient primal-dual DPO approach. Our approach first trains a model using standard DPO on reward preference data, and then fine-tunes

LLMs with a rearranged Lagrangian DPO objective on cost preference data, utilizing the reward information provided by the standard DPO-trained model. Unlike prior works (Dai et al., 2024; Liu et al., 2024b; Huang et al., 2024) which require to train and load three models, i.e., reward and cost models and the reward-cost-aligned language model, our approach only needs to train two models, i.e., the reward-aligned and reward-cost-aligned language models, and does not require any prior knowledge on the optimal solution. Moreover, we establish rigorous theoretical guarantees on the suboptimality and constraint violation of the output policy. Finally, we investigate an online setting where collecting preference data online is allowed. In this setting, we adopt exploration bonuses in our primal-dual DPO approach to guide exploration in the uncovered prompt-response space, and provide theoretical results that remove the dependence on preference data coverage. All proofs are deferred to Appendix due to space limits.

The contributions of our work are summarized as follows.

- We propose a novel primal-dual DPO approach for constrained LLM alignment. This approach first trains a model using standard DPO on reward preference data to offer reward information, and then adopts a rearranged Lagrangian DPO objective to fine-tune LLMs utilizing the offered reward information. It neither requires to train reward and cost models, which significantly saves memory costs, nor needs prior knowledge of the optimal Lagrange multiplier. We provide rigorous suboptimality and cost violation guarantees.

- We conduct experiments on the PKU-SafeRLHF preference dataset (Dai et al., 2024). Empirical results show that our approach achieves an effective helpfulness-harmlessness trade-off without training reward and cost models.

- In the online data setting, by incorporating exploration bonuses in our rearranged DPO objective, our approach can effectively explore the uncovered prompt-response space, and enjoys theoretical results that get rid of the dependence on preference data coverage.

## 2 RELATED WORK

In this section, we review the related work to ours. With the rapid development of LLMs, the alignment of LLMs has received extensive attention. RLHF (Ouyang et al., 2022) and DPO (Rafailov et al., 2023) are the two main algorithmic frameworks for LLM alignment. RLHF first trains a reward model, and then applies an RL algorithm with the learned reward model to fine-tune LLMs. DPO does not explicitly train a reward model, but instead directly fine-tunes LLMs using preference data.

Recently, to reduce the harmful content generation of LLMs, there are several works studying safety alignment. Dai et al. (2024) proposed a safe RLHF framework. Safe RLHF trains a reward model and a cost model on reward and cost preference data, respectively, and then applies an RL algorithm, PPO (Schulman et al., 2017), to maximize the Lagrangian function using the learned reward and cost functions. Liu et al. (2024b) used trained reward and cost models to regenerate preference data according to the Lagrangian function, and then applied DPO on regenerated data. Wachi et al. (2024) observed a relationship between the optimal policy of maximizing the Lagrangian function and that of maximizing the reward function, and performed DPO combined with this observation. However, their approach requires prior knowledge of the optimal Lagrange multiplier, and their theoretical results depend on the gap between the used and optimal Lagrange multipliers, which can be unbounded. Kim et al. (2025) reordered preference data if the preferred response is unsafe and the not-preferred response is safe, and ran DPO on reordered data. Their approach is inefficient in cost information learning. Huang et al. (2024); Zhang et al. (2025) investigated constrained LLM alignment from the perspective of dual optimization. Huang et al. (2024) proposed to first learn the optimal Lagrange multiplier via an explicit form of the dual function to avoid the expensive computation of evaluating the optimal policy under every updated Lagrange multiplier, and then compute the optimal policy. Zhang et al. (2025) generalized the algorithms in Huang et al. (2024) to the multi-shot scheme and focused on the primal-dual gap analysis under policy parameterization.

(Reviewer kvKV) (Reviewer VPsQ) In contrast to the above works, our approach only needs to train and load two models, rather than three as in prior works which need trained reward and cost models (Dai et al., 2024; Liu et al., 2024b; Huang et al., 2024; Zhang et al., 2025), or require prior knowledge of the optimal Lagrange multiplier (Wachi et al., 2024). Regarding theoretical results, to the best of our knowledge, only (Wachi et al., 2024; Huang et al., 2024; Zhang et al., 2025) and our

work provide theoretical guarantees on the output policy. Moreover, we provide novel theoretical results which get rid of the dependence on preference data coverage in the online data setting. The results in (Wachi et al., 2024; Huang et al., 2024) have an unbounded term or require an extra assumption, and the results in (Zhang et al., 2025) focus on analyzing the primal-dual gap brought by policy parameterization. Due to the difference in needed assumptions and main focuses, the results in (Wachi et al., 2024; Huang et al., 2024; Zhang et al., 2025) and ours cannot be directly compared. We present a comparison table on the assumptions, the required trained and loaded models, and theoretical guarantees with a more detailed description of related work in Appendix A.

## 3 PRELIMINARIES

**Reinforcement Learning from Human Feedback (RLHF).** The RLHF framework (Christiano et al., 2017; Ouyang et al., 2022) consists of three phases: (i) supervised fine-tuning a pre-trained LLM on a high-quality dataset of downstream tasks, e.g., dialogue and summarization, (ii) reward model learning, and (iii) RL optimization with the learned reward model.

Let $\mathcal{X}$ and $\mathcal{Y}$ denote the sets of all possible prompts and responses. We define a policy $\pi : \mathcal{X} \to \triangle_{\mathcal{Y}}$ as a mapping from $\mathcal{X}$ to a distribution on $\mathcal{Y}$, where $\triangle_{\mathcal{Y}}$ denotes the set of all distributions on $\mathcal{Y}$. We formulate an LLM as a policy, and use $\pi_{\text{ref}}$ to denote the supervised fine-tuned (SFT) model.

In the reward model learning phase, we have access to a reward preference dataset $\mathcal{D}^{\text{r}} = \{x_i^{\text{r}}, y_i^{\text{rw}}, y_i^{\text{rl}}\}_{i=1}^{N^{\text{r}}}$, where $x_i^{\text{r}}$ is a prompt, $y_i^{\text{rw}}, y_i^{\text{rl}}$ are preferred and dispreferred responses under prompt $x_i^{\text{r}}$, and the superscripts r, w and l stand for reward preference, "winner" and "loser", respectively. The generation of preference data is as follows: We assume that there exists an *unknown* reward function $r^*(x, y) \in [-R_{\max}, R_{\max}]$ for some constant $R_{\max}$, which models the helpfulness of response $y$ under prompt $x$. Human annotators compare a pair of responses $y^{\text{rw}}, y^{\text{rl}}$ under prompt $x$. Then, we assume that the probability that $y^{\text{rw}}$ is preferred to $y^{\text{rl}}$ under prompt $x$ follows the Bradley-Terry model (Bradley & Terry, 1952):

$$\Pr\left[y^{\text{rw}} \succ y^{\text{rl}} | x\right] = \frac{\exp(r^*(x, y^{\text{rw}}))}{\exp(r^*(x, y^{\text{rw}})) + \exp(r^*(x, y^{\text{rl}}))} = \sigma\left(r^*(x, y^{\text{rw}}) - r^*(x, y^{\text{rl}})\right), \quad (1)$$

where $\sigma(z) := \frac{1}{1+\exp(-z)}$ denotes the sigmoid function. This Bradley-Terry model is a standard assumption used to characterize human preference in the RLHF literature (Ouyang et al., 2022; Rafailov et al., 2023). With the reward preference data, we train a reward model $r$ via maximum likelihood estimation (MLE), i.e., minimizing the negative log-likelihood loss:

$$\min_r \ -\frac{1}{N^{\text{r}}} \sum_{i=1}^{N^{\text{r}}} \log \sigma\left(r(x_i^{\text{r}}, y_i^{\text{rw}}) - r(x_i^{\text{r}}, y_i^{\text{rl}})\right). \quad (2)$$

In the RL optimization phase, we apply RL algorithms, e.g., PPO (Schulman et al., 2017), to fine-tune the SFT model under the learned reward model $r$:

$$\max_\pi \ \mathbb{E}_{x \sim \mathcal{D}^{\text{p}}}\left[\mathbb{E}_{y \sim \pi(\cdot|x)}\left[r(x, y)\right] - \beta \cdot \text{KL}\left(\pi(\cdot|x) \| \pi_{\text{ref}}(\cdot|x)\right)\right]. \quad (3)$$

Here $\beta$ is a parameter controlling the deviation between the trained model $\pi$ and SFT model $\pi_{\text{ref}}$, since we do not want the trained model to be too far away from the SFT model. $\mathcal{D}^{\text{p}}$ is a distribution of prompts, and the optimal solution to Eq. (3) is independent of $\mathcal{D}^{\text{p}}$, which will be presented in Eq. (4).

**Direct Preference Optimization (DPO).** Recently, Rafailov et al. (2023) designed an direct preference optimization (DPO) approach, which bypasses the reward model training phase in RLHF, and directly fine-tunes LLMs using preference data. The derivation idea of DPO is as follows.

First, the optimal solution to Eq. (3) is (Peters & Schaal, 2007; Peng et al., 2019)

$$\pi_r^*(y|x) = \frac{\pi_{\text{ref}}(y|x) \exp\left(\frac{1}{\beta} r(x, y)\right)}{Z_r(x)}, \quad (4)$$

where $Z_r(x) := \sum_{y' \in \mathcal{Y}} \pi_{\text{ref}}(y'|x) \exp(\frac{1}{\beta} r(x, y'))$ is the partition function. Then, we can rewrite Eq. (4) to express the reward function $r$ by the optimal policy $\pi_r^*$ as

$$r(x, y) = \beta \log \frac{\pi_r^*(y|x)}{\pi_{\text{ref}}(y|x)} + \beta \log Z_r(x). \quad (5)$$

Eqs. (4) and (5) hold for any reward function $r$. Hence, the Bradley-Terry model in Eq. (1) can be expressed by the optimal policy $\pi_{r^*}^*$:

$$\Pr\left[y^{\text{rw}} \succ y^{\text{rl}}|x\right] = \sigma\left(\beta \log \frac{\pi_{r^*}^*(y^{\text{rw}}|x)}{\pi_{\text{ref}}(y^{\text{rw}}|x)} - \beta \log \frac{\pi_{r^*}^*(y^{\text{rl}}|x)}{\pi_{\text{ref}}(y^{\text{rl}}|x)}\right), \qquad (6)$$

where the partition function $Z_{r^*}(x)$ is cancelled out. Now, by expressing the probability that preference data happen by $\pi_{r^*}^*$, we can replace the likelihood in the MLE training objective in Eq. (2) by Eq. (6), and obtain a new objective with the optimization variable directly being the policy:

$$\min_{\pi} -\frac{1}{N^{\text{r}}} \sum_{i=1}^{N^{\text{r}}} \log \sigma\left(\beta \log \frac{\pi(y_i^{\text{rw}}|x_i^{\text{r}})}{\pi_{\text{ref}}(y_i^{\text{rw}}|x_i^{\text{r}})} - \beta \log \frac{\pi(y_i^{\text{rl}}|x_i^{\text{r}})}{\pi_{\text{ref}}(y_i^{\text{rl}}|x_i^{\text{r}})}\right). \qquad (7)$$

Eq. (7) is the training objective of DPO. Thus, DPO directly uses preference data to fine-tune LLMs without training a reward model, and enjoys lower memory and computational costs than RLHF.

**Safe RLHF.** To enhance safety in LLM alignment, Dai et al. (2024) proposed a safe RLHF framework. In safe RLHF, we assume that there exists an *unknown* cost function $c^*(x,y) \in [-C_{\max}, C_{\max}]$ for some constant $C_{\max}$, which characterizes the harmfulness of response $y$ under prompt $x$. (Reviewer EtwF) In addition to reward preference dataset $\mathcal{D}^{\text{r}}$, we also have access to a cost preference dataset $\mathcal{D}^{\text{c}} = \{x_i^{\text{c}}, y_i^{\text{cw}}, y_i^{\text{cl}}\}_{i=1}^{N^{\text{c}}}$, where $y_i^{\text{cw}}$ and $y_i^{\text{cl}}$ denote unsafer and safer responses under prompt $x_i^{\text{c}}$ ($y_i^{\text{cw}}$ has a higher cost than $y_i^{\text{cl}}$), and the superscript c refers to cost preference. We assume that cost preference is generated according to the Bradley-Terry model with cost function $c^*$, i.e.,

$$\Pr\left[y^{\text{cw}} \succ y^{\text{cl}}|x\right] = \frac{\exp(c^*(x, y^{\text{cw}}))}{\exp(c^*(x, y^{\text{cw}})) + \exp(c^*(x, y^{\text{cl}}))} = \sigma\left(c^*(x, y^{\text{cw}}) - c^*(x, y^{\text{cl}})\right). \qquad (8)$$

Similar to Eq. (2), we can also train a cost model $c$ via MLE:

$$\min_{c} -\frac{1}{N^{\text{c}}} \sum_{i=1}^{N^{\text{c}}} \log \sigma\left(c(x_i^{\text{c}}, y_i^{\text{cw}}) - c(x_i^{\text{c}}, y_i^{\text{cl}})\right). \qquad (9)$$

To restrict the costs of LLM outputs within a threshold, we consider the constrained optimization:

$$\max_{\pi} \quad \mathbb{E}_{x \sim \mathcal{D}^{\text{c}}}\left[\mathbb{E}_{y \sim \pi(\cdot|x)}[r(x,y)] - \beta \cdot \text{KL}\left(\pi(\cdot|x) \| \pi_{\text{ref}}(\cdot|x)\right)\right]$$

$$\text{s.t.} \quad c(x, y) \leq 0, \quad \forall x \sim \mathcal{D}^{\text{c}}, y \sim \pi(\cdot|x).$$

Here for simplicity, we set the threshold of harmfulness to $0$. The above problem is hard to solve using neural networks, since it requires the cost of every possible response $y$ to stay below $0$.

To feasibly perform safety alignment, many prior works, e.g., (Dai et al., 2024; Wachi et al., 2024; Liu et al., 2024b; Kim et al., 2025), consider a relaxed optimization problem with an expected cost constraint, which we called *constrained alignment problem*:

$$\max_{\pi} \quad f(\pi) := \mathbb{E}_{x \sim \mathcal{D}^{\text{p}}}\left[\mathbb{E}_{y \sim \pi(\cdot|x)}[r^*(x,y)] - \beta \cdot \text{KL}\left(\pi(\cdot|x) \| \pi_{\text{ref}}(\cdot|x)\right)\right]$$

$$\text{s.t.} \quad g(\pi) := \mathbb{E}_{x \sim \mathcal{D}^{\text{p}}, y \sim \pi(\cdot|x)}[c^*(x,y)] \leq 0. \qquad (10)$$

In this work, we also study this relaxed problem. Then, it is natural to look into the Lagrangian dual problem of the above constrained optimization:

$$\min_{\lambda \geq 0} \max_{\pi} L(\pi; \lambda) := \mathbb{E}_{x \sim \mathcal{D}^{\text{p}}}\left[\mathbb{E}_{y \sim \pi(\cdot|x)}[r^*(x,y) - \lambda \cdot c^*(x,y)] - \beta \cdot \text{KL}\left(\pi(\cdot|x) \| \pi_{\text{ref}}(\cdot|x)\right)\right], \quad (11)$$

where $\lambda \geq 0$ is a Lagrange multiplier. Throughout the paper, we call $L(\pi; \lambda)$ the *Lagrangian function*.

With the above unconstrained formulation, the safe RLHF framework (Dai et al., 2024) regarded $r - \lambda \cdot c$ as a new reward function and applied an RL algorithm PPO (Schulman et al., 2017) to maximize $L(\pi; \lambda)$, and performed subgradient descent (Beck, 2017) to update $\lambda$. Safe RLHF requires to train both reward and cost models, which incurs high memory and computational costs.

## 4 PRIMAL-DUAL DPO UTILIZING STANDARD DPO

In this section, we propose a provably efficient primal-dual DPO approach for the constrained alignment problem (Eq. (10)), utilizing a model trained using standard DPO on reward preference data to provide reward information. We first describe the key idea behind our approach, and present the specific algorithm PD-DPO which has rigorous theoretical guarantees.

### 4.1 OUR APPROACH

First, we have that the optimal solution to $\max_\pi L(\pi; \lambda)$ in Eq. (11) is

$$r(x, y) - \lambda \cdot c(x, y) = \beta \log \frac{\pi^*_{r-\lambda \cdot c}(y|x)}{\pi_{\text{ref}}(y|x)} + \beta \log Z_{r-\lambda \cdot c}(x), \qquad (12)$$

where $Z_{r-\lambda \cdot c}(x) := \sum_{y' \in \mathcal{Y}} \pi_{\text{ref}}(y'|x) \exp(\frac{1}{\beta}\left(r(x, y') - \lambda \cdot c(x, y')\right))$ is the partition function, and $r$ and $c$ can be any reward and cost functions.

When one wants to apply the derivation idea of DPO in Eqs. (6) and (7), a difficulty arises: *We do not have preference data generated according to $r - \lambda \cdot c$*, but only have preference data generated according to $r$ and $c$ separately. Thus, we cannot use $\beta \log \frac{\pi^*_{r-\lambda \cdot c}(y|x)}{\pi_{\text{ref}}(y|x)}$ to directly express data likelihood as in Eq. (7), which means that the DPO derivation idea cannot be directly applied here.

To overcome this difficulty, we first rearrange Eq. (12) as

$$c(x, y) = \frac{1}{\lambda}\left(r(x, y) - \beta \log \frac{\pi^*_{r-\lambda \cdot c}(y|x)}{\pi_{\text{ref}}(y|x)} - \beta \log Z_{r-\lambda \cdot c}(x)\right).$$

Plugging the above equation with $r^*$ and $c^*$ into Eq. (8), the generation of cost preference data can be rewritten as $\Pr[y^{\text{cw}} \succ y^{\text{cl}}|x] =$

$$\sigma\left(\frac{1}{\lambda}\left(r^*(x, y^{\text{cw}}) - \beta \log \frac{\pi^*_{r^*-\lambda \cdot c^*}(y^{\text{cw}}|x)}{\pi_{\text{ref}}(y^{\text{cw}}|x)} - \left(r^*(x, y^{\text{cl}}) - \beta \log \frac{\pi^*_{r^*-\lambda \cdot c^*}(y^{\text{cl}}|x)}{\pi_{\text{ref}}(y^{\text{cl}}|x)}\right)\right)\right),$$

where $Z_{r^*-\lambda \cdot c^*}(x)$ is cancelled out. Then, replacing the cost preference data likelihood in Eq. (9) by the above equation, we can obtain a training objective with the optimization variable directly being the policy which is supposed to get close to $\pi^*_{r^*-\lambda \cdot c^*}$ during training:

$$\min_\pi -\frac{1}{N^{\text{c}}} \sum_{i=1}^{N^{\text{c}}} \log \sigma\left(\frac{1}{\lambda}\left(r^*(x_i^{\text{c}}, y_i^{\text{cw}}) - \beta L_\pi(y_i^{\text{cw}}|x_i^{\text{c}}) - \left(r^*(x_i^{\text{c}}, y_i^{\text{cl}}) - \beta L_\pi(y_i^{\text{cl}}|x_i^{\text{c}})\right)\right)\right), \quad (13)$$

where $L_\pi(y|x) := \log \frac{\pi(y|x)}{\pi_{\text{ref}}(y|x)}$ is the logarithmic ratio of response $y$ under $x$ between $\pi$ and $\pi_{\text{ref}}$.

Now the main challenge lies in that *we do not know $r^*$*, and meanwhile, we do not want to explicitly train a reward model in order to keep memory and computational efficiency. To handle this challenge, we make an observation that $r^*(x_i^{\text{c}}, y_i^{\text{cw}}) - r^*(x_i^{\text{c}}, y_i^{\text{cl}})$ *can be expressed by* $\beta \log \frac{\pi^*_{r^*}(y_i^{\text{cw}}|x_i^{\text{c}})}{\pi_{\text{ref}}(y_i^{\text{cw}}|x_i^{\text{c}})} - \beta \log \frac{\pi^*_{r^*}(y_i^{\text{cl}}|x_i^{\text{c}})}{\pi_{\text{ref}}(y_i^{\text{cl}}|x_i^{\text{c}})}$ according to Eq. (4). Then, $\pi^*_{r^*}$ is what we can learn by training a model using standard DPO on reward preference data.

Therefore, using this observation, the training objective Eq. (13) can be rewritten as

$$\min_\pi -\frac{1}{N^{\text{c}}} \sum_{i=1}^{N^{\text{c}}} \log \sigma\left(\frac{\beta}{\lambda}\left(L_{\pi^*_{r^*}}(y_i^{\text{cw}}|x_i^{\text{c}}) - L_\pi(y_i^{\text{cw}}|x_i^{\text{c}}) - \left(L_{\pi^*_{r^*}}(y_i^{\text{cl}}|x_i^{\text{c}}) - L_\pi(y_i^{\text{cl}}|x_i^{\text{c}})\right)\right)\right), \quad (14)$$

where $\pi^*_{r^*}$ can be learned by first training a model using standard DPO on reward preference data.

Eq. (14) is the main idea of our primal-dual DPO approach utilizing standard DPO. Our approach only needs to train two models, i.e., the reward-aligned and reward-cost-aligned language models, rather than three models (i.e., reward and cost models and the reward-cost-aligned language models) as in prior works (Dai et al., 2024; Liu et al., 2024b; Huang et al., 2024; Zhang et al., 2025), which significantly reduces memory costs. This approach shows even more advantages when there already exists a trained model on reward preference data, which is often the case since there are many high-quality and open-source LLMs (Dubey et al., 2024; Team et al., 2025).

### 4.2 A PROVABLY EFFICIENT ALGORITHM PD-DPO

While Eq. (14) has presented the main idea of our primal-dual DPO approach, to enable rigorous theoretical guarantees, we develop a specific provably efficient algorithm PD-DPO, which imposes policy search constraints based on Eq. (14) and enjoys suboptimality and constraint violation guarantees. Before describing the specific algorithm PD-DPO, we first introduce several assumptions.

---

**Algorithm 1:** PD-DPO

---

**Input:** $\beta, \pi_{\text{ref}}, \rho, \lambda_1, K, N^{\text{CE}}, M^{\text{CE}}, \mathcal{D}^{\text{p}}, \mathcal{D}^{\text{r}} = \{(x_i^{\text{r}}, y_i^{\text{rw}}, y_i^{\text{rl}})\}_{i \in [N^{\text{r}}]}, \mathcal{D}^{\text{c}} = \{(x_i^{\text{c}}, y_i^{\text{cw}}, y_i^{\text{cl}})\}_{i \in [N^{\text{c}}]}$

1 Train a model using standard DPO on reward preference data:

$$\pi_{\hat{r}}^* \leftarrow \underset{\pi \in \Pi^{\text{r}}}{\operatorname{argmin}} -\frac{1}{N^{\text{r}}} \sum_{i=1}^{N^{\text{r}}} \log \sigma \left( \beta \log \frac{\pi(y_i^{\text{rw}}|x_i^{\text{r}})}{\pi_{\text{ref}}(y_i^{\text{rw}}|x_i^{\text{r}})} - \beta \log \frac{\pi(y_i^{\text{rl}}|x_i^{\text{r}})}{\pi_{\text{ref}}(y_i^{\text{rl}}|x_i^{\text{r}})} \right), \qquad (15)$$

where $\Pi^{\text{r}}$ is defined in Eq. (17)

2 **for** $k = 1, 2, \ldots, K$ **do**

3    Train a model using a rearranged Lagrangian DPO objective on cost preference data:

$$\pi_k \leftarrow \underset{\pi \in \Pi_k^{\text{c}}}{\operatorname{argmin}} -\frac{1}{N^{\text{c}}} \sum_{i=1}^{N^{\text{c}}} \log \sigma \left( \frac{1}{\lambda_k} \left( \beta \log \frac{\pi_{\hat{r}}^*(y_i^{\text{cw}}|x_i^{\text{c}})}{\pi_{\text{ref}}(y_i^{\text{cw}}|x_i^{\text{c}})} - \beta \log \frac{\pi(y_i^{\text{cw}}|x_i^{\text{c}})}{\pi_{\text{ref}}(y_i^{\text{cw}}|x_i^{\text{c}})} \right. \right.$$
$$\left. \left. - \left( \beta \log \frac{\pi_{\hat{r}}^*(y_i^{\text{cl}}|x_i^{\text{c}})}{\pi_{\text{ref}}(y_i^{\text{cl}}|x_i^{\text{c}})} - \beta \log \frac{\pi(y_i^{\text{cl}}|x_i^{\text{c}})}{\pi_{\text{ref}}(y_i^{\text{cl}}|x_i^{\text{c}})} \right) \right) \right), \qquad (16)$$

   where $\Pi_k^{\text{c}}$ is defined in Eq. (18)

4    Construct an estimate $\tilde{c}_k$ for $\mathbb{E}_{x \sim \mathcal{D}^{\text{p}}, y \sim \pi_k(\cdot|x)}[c^*(x, y)]$: For $i = 1, \ldots, N^{\text{CE}}$, first sample $x_i \sim \mathcal{D}^{\text{p}}, y_i \sim \pi_k(\cdot|x_i)$. Then, for each $(x_i, y_i)$, sample $\{Z_{i,j}\}_{j=1}^{M^{\text{CE}}} \overset{\text{i.i.d.}}{\sim} \text{Ber}(\sigma(c^*(x_i, y_i)))$. Set $\tilde{c}_k \leftarrow \frac{1}{N^{\text{CE}}} \sum_{i=1}^{N^{\text{CE}}} \sigma^{-1}(\frac{1}{M^{\text{CE}}} \sum_{j=1}^{M^{\text{CE}}} Z_{i,j})$, where $\sigma^{-1}(z) := \log(\frac{1}{1-z} - 1)$ is the inverse of the sigmoid function

5    $\lambda_{k+1} \leftarrow \text{Proj}_{[0,2\rho]}(\lambda_k + \eta \tilde{c}_k)$, where $\eta := \frac{\lambda_1}{C_{\max}\sqrt{K}}$

6 **return** $\pi_K^{\text{out}} := \text{unif}(\pi_1, \ldots, \pi_K)$

---

**Assumption 1** (Slater's Condition). *There exists a policy $\bar{\pi}$ which satisfies $\mathbb{E}_{x \sim \mathcal{D}^{\text{p}}, y \sim \bar{\pi}(\cdot|x)}[c^*(x, y)] < 0$. In addition, we know a constant $\rho \geq \frac{f(\pi^*) - f(\bar{\pi})}{-\mathbb{E}_{x \sim \mathcal{D}^{\text{p}}, y \sim \bar{\pi}(\cdot|x)}[c^*(x,y)]}$.*

This assumption is common in the constrained optimization and learning literature (Beck, 2017; Efroni et al., 2020). In practice, it is reasonable that there exists a safe policy model, e.g., a language model which refuses to answer harmful questions albeit less helpful.

Following prior works (Dai et al., 2024; Kim et al., 2025), we also allow querying cost binary feedback from human annotators, which indicates whether a response $y$ is safe under prompt $x$. Such cost binary feedback is generated according to

$$\Pr[Z(y) = 1|x] = \sigma(c^*(x, y))$$

and $\Pr[Z(y) = 0|x] = 1 - \sigma(c^*(x, y))$, where $Z(y) = 1$ and $Z(y) = 0$ denote that $y$ is unsafe and safe, respectively. We will use cost binary feedback in algorithm PD-DPO to estimate the cost of the current trained model for Lagrange multiplier update.

Now we present an efficient and provably convergent primal-dual DPO algorithm PD-DPO. PD-DPO first trains a model using standard DPO on reward preference data, and leverages this model to provide reward information to train a model using a rearranged Lagrangian DPO objective with policy search constraints. Then, PD-DPO conducts projected subgradient descent to update the Lagrange multiplier. PD-DPO performs such model training and Lagrange multiplier update alternately.

Algorithm 1 illustrates the algorithm procedure of PD-DPO. Specifically, PD-DPO first trains a model $\pi_{\hat{r}}^*$ using the standard DPO objective (Rafailov et al., 2023) on reward preference data $\mathcal{D}^{\text{r}}$ within a constrained policy search range:

$$\Pi^{\text{r}} := \left\{ \pi(y|x) = \frac{\pi_{\text{ref}}(y|x) \cdot \exp\left(\frac{1}{\beta} r(x, y)\right)}{\sum_{y' \in \mathcal{Y}} \pi_{\text{ref}}(y'|x) \cdot \exp\left(\frac{1}{\beta} r(x, y')\right)} : r \in [-R_{\max}, R_{\max}] \right\}. \qquad (17)$$

Since we use only finite preference data, we cannot exactly learn $\pi_{r^*}^*$. Instead, we learn a reward function $\hat{r}$ which is close to $r^*$ and implicitly maintained by the policy $\pi_{\hat{r}}^*$. The notation $\pi_{\hat{r}}^*$ denotes

the optimal policy under the learned reward function $\hat{r}$ (Eq. (4)). The policy search range $\Pi^r$ is used to restrict the learned reward function $\hat{r}$ within $[-R_{\max}, R_{\max}]$ (Line 1). Next, in each iteration $k$, given a Lagrange multiplier $\lambda_k$, PD-DPO utilizes the reward information provided by $\pi^*_{\hat{r}}$ to train a model using a rearranged Lagrangian DPO objective as derived in Section 4.1, but with a constrained policy search range:

$$\Pi^c_k := \left\{ \pi(y|x) = \frac{\pi_{\text{ref}}(y|x) \exp\left(\frac{1}{\beta}\left(\beta \log \frac{\pi^*_{\hat{r}}(y|x)}{\pi_{\text{ref}}(y|x)} - \lambda_k c(x,y)\right)\right)}{\sum_{y' \in \mathcal{Y}} \pi_{\text{ref}}(y'|x) \exp\left(\frac{1}{\beta}\left(\beta \log \frac{\pi^*_{\hat{r}}(y'|x)}{\pi_{\text{ref}}(y'|x)} - \lambda_k c(x,y')\right)\right)} : c \in [-C_{\max}, C_{\max}] \right\}$$

$$\overset{(a)}{=} \left\{ \pi(y|x) = \frac{\pi_{\text{ref}}(y|x) \exp\left(\frac{1}{\beta}\left(\hat{r}(x,y) - \lambda_k c(x,y)\right)\right)}{\sum_{y' \in \mathcal{Y}} \pi_{\text{ref}}(y'|x) \exp\left(\frac{1}{\beta}\left(\hat{r}(x,y') - \lambda_k c(x,y')\right)\right)} : c \in [-C_{\max}, C_{\max}] \right\}. \quad (18)$$

Here equality (a) is due to Eq. (5) and the fact that the partition function $Z_{\hat{r}}(x)$ only depends on $x$ and can be cancelled out. $\Pi^c_k$ is used to restrict the learned cost function within $[-C_{\max}, C_{\max}]$ (Line 3).

After obtaining $\pi_k$, we estimate the cost of $\pi_k$ for Lagrange multiplier update using the following scheme: We i.i.d. draw $N^{\text{CE}}$ prompt-response pairs $\{(x_i, y_i)\}_{i=1}^{N^{\text{CE}}}$ using $\pi_k$, where the superscript CE stands for cost estimation. For each pair $(x_i, y_i)$, we i.i.d. query human annotators whether response $y_i$ is safe under prompt $x_i$ $M^{\text{CE}}$ times, and obtain $M^{\text{CE}}$ cost binary feedback $\{Z_{i,j}\}_{j=1}^{M^{\text{CE}}}$ drawn from $\text{Ber}(\sigma(c^*(x_i, y_i)))$. Then, we take the inverse of the sigmoid function $\sigma^{-1}(\cdot)$ on the average of these $M^{\text{CE}}$ Bernoulli outcomes to obtain an estimate $\tilde{c}_k$ for the expected cost of $\pi_k$ (Line 4). In analysis, we can bound the deviation between this estimate $\tilde{c}_k$ and the expected cost of $\pi_k$ (see Appendix C.2). After cost estimation, PD-DPO performs projected subgradient descent with $\tilde{c}_k$ to update Lagrange multiplier $\lambda_k$, and enters the next iteration (Line 5).

### 4.3 THEORETICAL GUARANTEES OF ALGORITHM PD-DPO

Unlike prior works (Dai et al., 2024; Liu et al., 2024b; Kim et al., 2025) which did not provide theoretical guarantees for their output policy models, we establish rigid suboptimality and constraint violation guarantees for the output policy of algorithm PD-DPO.

First, we note that our rearranged Lagrangian DPO objective (Eq. (16)) and the safe RLHF procedure, which first trains reward and cost models using MLE and maximizes $L(\pi; \lambda_k)$ under the learned reward and cost functions, have the same set of optimal solutions (see Theorem 4 in Appendix C.1 for a formal statement). Next, we present the theoretical results of algorithm PD-DPO.

For any $(x, y) \in \mathcal{X} \times \mathcal{Y}$, let $\phi(x, y)$ denote a $|\mathcal{X}||\mathcal{Y}|$-dimensional vector where the entry corresponding to $(x, y)$ is 1 and all other entries are 0. Let $\alpha(z) := \sqrt{(\exp(z) + \exp(-z) + 2)^2 \left(|\mathcal{X}||\mathcal{Y}| + \log\left(\frac{1}{\delta}\right)\right) + \gamma z^2}$ and

$$B := \rho C_{\max} \sqrt{\frac{\log\left(\frac{K}{\delta}\right)}{N^{\text{CE}}}} + \rho W \sqrt{\frac{\log\left(\frac{|\mathcal{X}||\mathcal{Y}|N^{\text{CE}}K}{\delta}\right)}{M^{\text{CE}}}}$$

$$+ \rho \cdot \alpha(C_{\max})\left(\mathbb{E}_{(x,y)\sim\mathcal{D}^p \times \pi^*}\left[\|\phi(x,y)\|_{(\Sigma_{\mathcal{D}^c} + \gamma I)^{-1}}\right] + \frac{1}{K}\sum_{k=1}^K \mathbb{E}_{(x,y)\sim\mathcal{D}^p \times \pi_k}\left[\|\phi(x,y)\|_{(\Sigma_{\mathcal{D}^c} + \gamma I)^{-1}}\right]\right)$$

$$+ \alpha(R_{\max})\left(\mathbb{E}_{(x,y)\sim\mathcal{D}^p \times \pi^*}\left[\|\phi(x,y)\|_{(\Sigma_{\mathcal{D}^r} + \gamma I)^{-1}}\right] + \frac{1}{K}\sum_{k=1}^K \mathbb{E}_{(x,y)\sim\mathcal{D}^p \times \pi_k}\left[\|\phi(x,y)\|_{(\Sigma_{\mathcal{D}^r} + \gamma I)^{-1}}\right]\right).$$

Here $\Sigma_{\mathcal{D}^\diamond} := \sum_{(x,y,y') \in \mathcal{D}^\diamond} (\phi(x,y) - \phi(x,y'))(\phi(x,y) - \phi(x,y'))^\top$ with $\diamond \in \{r, c\}$. For any $\pi$, $(x, y) \sim \mathcal{D}^p \times \pi$ denotes $x \sim \mathcal{D}^p$, $y \sim \pi(\cdot|x)$. $\gamma > 0$ is an arbitrary regularization parameter. $W$ is a parameter dependent on $C_{\max}$, which is formally defined in Eq. (28) in Appendix C.2.

**Theorem 1** (Result of Algorithm PD-DPO). *With probability at least $1 - \delta$, for any $K \geq 1$, the output policy $\pi^{\text{out}}_K$ of algorithm* PD-DPO *satisfies*

$$f(\pi^*) - f(\pi^{\text{out}}_K) = O\left(\frac{\lambda_1 C_{\max}}{\sqrt{K}} + B\right), \quad g(\pi^{\text{out}}_K) = O\left(\frac{C_{\max}}{\rho\sqrt{K}}\left(\frac{(\lambda_1 - 2\rho)^2}{\lambda_1} + \lambda_1\right) + \frac{B}{\rho}\right).$$

In this result, the $\frac{1}{\sqrt{K}}$ term is an inherent error of the primal-dual method. The $\frac{1}{\sqrt{N^{\mathrm{CE}}}}$ and $\frac{1}{\sqrt{M^{\mathrm{CE}}}}$ terms are the error due to cost estimation. The four $\|\phi(x,y)\|_{(\Sigma_{\mathcal{D}^\diamond}+\gamma I)^{-1}}$ terms are the error due to inferring reward and cost information from preference data. The $\|\phi(x,y)\|_{(\Sigma_{\mathcal{D}^\diamond}+\gamma I)^{-1}}$ factor stands for how broadly the given preference data cover. Theorem 1 shows that the suboptimality and cost violation of the output policy by algorithm PD-DPO can be arbitrarily close to zero, when the given preference data have sufficient coverage, and the number of preference data, the number of iterations $K$, and the number of samples for cost estimation $N^{\mathrm{CE}}, M^{\mathrm{CE}}$ are large enough.

## 5 EXPLORATORY PRIMAL-DUAL DPO WITH EXPLORATION BONUSES

The result of algorithm PD-DPO (Theorem 1) depends on the coverage of preference data, i.e., $\|\phi(x,y)\|_{(\Sigma_{\mathcal{D}^\diamond}+\gamma I)^{-1}}$. If the given preference data do not have sufficient coverage, the suboptimality and constraint violation of PD-DPO can be unbounded.

To resolve this coverage issue, we further investigate an online setting where collecting preference data online is allowed. In this setting, we develop an exploratory primal-dual DPO algorithm O-PD-DPO, which incorporates exploration bonuses $b_k^{\mathrm{c}}(x,y)$ and $b_k^{\mathrm{r}}(x,y)$ in the rearranged Lagrangian DPO and standard DPO objectives. The construction of exploration bonuses is based on the Bradley-Terry model (Eqs. (1) and (8)), which is commonly assumed in many RLHF works, e.g., (Zhu et al., 2023; Wachi et al., 2024; Huang et al., 2024). In algorithm O-PD-DPO, the trained policy has an incentive to explore the uncovered prompt-response space, and gradually expands the used preference data. We defer the pseudo-code and detailed description of O-PD-DPO to Appendix D due to space limits.

We take the incorporation of $b_k^{\mathrm{r}}$ in standard DPO as an example to explain the *intuition behind why including exploration bonuses can encourage exploration*. For the standard DPO objective (Eq. (15)), algorithm O-PD-DPO will subtract a $b_k^{\mathrm{r}}(x,y^{\mathrm{rw}})$ term from the original $\beta \log \frac{\pi(y^{\mathrm{rw}}|x)}{\pi_{\mathrm{ref}}(y^{\mathrm{rw}}|x)}$ term. When preference data do not cover $(x,y^{\mathrm{rw}})$ well, $b_k^{\mathrm{r}}(x,y^{\mathrm{rw}})$ will be large. Then, subtracting a large value from $\beta \log \frac{\pi(y^{\mathrm{rw}}|x)}{\pi_{\mathrm{ref}}(y^{\mathrm{rw}}|x)}$ encourages $\pi$ to put a higher probability on $y^{\mathrm{rw}}$ to maintain the original value of $\beta \log \frac{\pi(y^{\mathrm{rw}}|x)}{\pi_{\mathrm{ref}}(y^{\mathrm{rw}}|x)}$ which achieves the optimal value of the MLE objective function. Thus, by incorporating exploration bonuses in the DPO objective, the trained policy is incentivized to explore the uncovered prompt-response space. This design and its analysis are novel to the RLHF literature.

Now we provide the suboptimality and constraint violation guarantees of algorithm O-PD-DPO. Let $\omega(z) := \sqrt{(\exp(z) + \exp(-z) + 2)^2 \cdot (|\mathcal{X}||\mathcal{Y}| + \log(\frac{K}{\delta}))/N^{\mathrm{on}} + \gamma^{\mathrm{on}} z^2}$ and

$$
B^{\mathrm{on}} := \rho C_{\max} \sqrt{\frac{\log\left(\frac{K}{\delta}\right)}{N^{\mathrm{CE}}}} + \rho W \sqrt{\frac{\log\left(\frac{|\mathcal{X}||\mathcal{Y}|N^{\mathrm{CE}}K}{\delta}\right)}{M^{\mathrm{CE}}}} + (\rho \cdot \omega(C_{\max}) + \omega(R_{\max})) \cdot
$$

$$
\sqrt{\frac{|\mathcal{X}||\mathcal{Y}|}{K}\left(\log\left(\frac{\gamma^{\mathrm{on}} + \max\{|\mathcal{D}_1^{\mathrm{r}}|, |\mathcal{D}_1^{\mathrm{c}}|\} + K}{|\mathcal{X}||\mathcal{Y}|\gamma^{\mathrm{on}}}\right) + \frac{1}{C^{\mathrm{base}}}\log\left(\frac{\gamma^{\mathrm{on}} + \max\{|\mathcal{D}_1^{\mathrm{r}}|, |\mathcal{D}_1^{\mathrm{c}}|\} + C^{\mathrm{base}}K}{|\mathcal{X}||\mathcal{Y}|\gamma^{\mathrm{on}}}\right)\right)}.
$$

Here $N^{\mathrm{on}}$ is the number of preference data collected online in each iteration. $\gamma^{\mathrm{on}} > 0$ is a given regularization parameter. $C^{\mathrm{base}}$ is a parameter related to a baseline policy which is used in online data collection. The definitions of the baseline policy and $C^{\mathrm{base}}$ are in Eq. (34) in Appendix D.

**Theorem 2** (Result of Algorithm O-PD-DPO). *With probability at least $1 - \delta$, for any $K \geq 1$, the output policy $\pi_K^{\mathrm{out}}$ of algorithm O-PD-DPO satisfies*

$$
f(\pi^*) - f(\pi_K^{\mathrm{out}}) = O\left(\frac{\lambda_1 C_{\max}}{\sqrt{K}} + B^{\mathrm{on}}\right), \quad g(\pi_K^{\mathrm{out}}) = O\left(\frac{C_{\max}}{\rho\sqrt{K}}\left(\frac{(\lambda_1 - 2\rho)^2}{\lambda_1} + \lambda_1\right) + \frac{B^{\mathrm{on}}}{\rho}\right).
$$

Compared to Theorem 1, here the results have no dependence on the coverage of preference data, i.e., $\|\phi(x,y)\|_{(\Sigma_{\mathcal{D}^\diamond}+\gamma I)^{-1}}$. Theorem 2 demonstrates that the adoption of exploration bonuses in the rearranged Lagrangian DPO objective effectively incentivizes exploration and expands the used preference data during training. When all problem parameters $K, N^{\mathrm{CE}}, M^{\mathrm{CE}}, N^{\mathrm{on}}$ are large enough, the suboptimality and constraint violation bounds will shrink to zero.

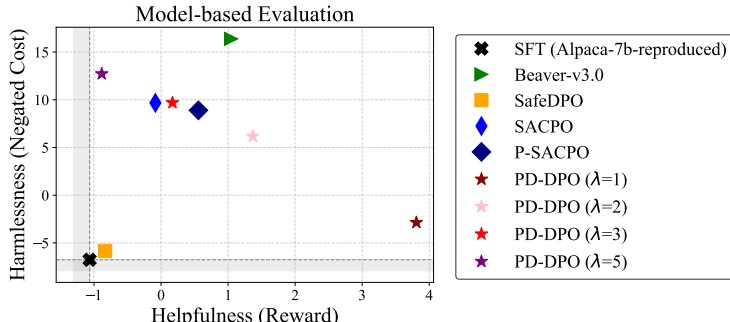

Figure 1: (Reviewer VPsQ) (Reviewer kvKV) (Reviewer QQdF) Rewards and negated costs of responses generated by compared language models when evaluated by Beaver-7b-unified-reward and Beaver-7b-unified-cost (Dai et al., 2024).

While prior works (Huang et al., 2024; Wachi et al., 2024) also provide theoretical results, Huang et al. (2024) require an assumption that the optimal policy is feasible under the estimated reward and cost functions. The results in (Wachi et al., 2024) have a term of the deviation between the used and optimal Lagrange multipliers, which can be unbounded since their algorithm does not contain any scheme to learn the optimal Lagrange multiplier. In addition, the results in prior works depend on preference data coverage. To the best of our knowledge, Theorem 2 is the first result for the constrained alignment problem (Eq. 10) to get rid of the dependence on preference data coverage.

## 6 EXPERIMENTS

In this section, we provide experimental results. Our experiments are run on an Intel Xeon Platinum 8558 CPU and a single NVIDIA GH200 96GB GPU. Following prior works, we use the PKU-SafeRLHF preference dataset (Dai et al., 2024) to train and evaluate models, and take `Alpaca-7b-reproduced` as the SFT model, which is a fine-tuned version of the LLaMA-2-7b model (Touvron et al., 2023b) on the Alpaca dataset (Taori et al., 2023). We compare our algorithm `PD-DPO` with the SFT model and existing open-source safety alignment algorithms `Beaver-v3.0` (Dai et al., 2024), `SafeDPO` (Kim et al., 2025), `SACPO` and `P-SACPO` (Wachi et al., 2024).

Figure 1 presents the model-based evaluation results, i.e., the average reward and negated cost scores of responses generated by compared language models, when evaluated by the reward model Beaver-7b-unified-reward and the cost model Beaver-7b-unified-cost (Dai et al., 2024). (Reviewer kvKV) (Reviewer QQdF) Our `PD-DPO` ($\lambda = 3$) outperforms the SFT model, `SafeDPO` (Kim et al., 2025) and `SACPO` (Wachi et al., 2024) in both harmlessness and helpfulness. The performance of `PD-DPO` ($\lambda = 3$) is comparable to that of `P-SACPO` (Wachi et al., 2024). However, `PD-DPO` does not require prior knowledge of the optimal Lagrange multiplier as in `SACPO` and `P-SACPO`. While `PD-DPO` has worse performance than `Beaver-v3.0` (Dai et al., 2024), `PD-DPO` only needs to train two models rather than three models as in `Beaver-v3.0`. In addition, `Beaver-v3.0` requires much higher memory costs than our algorithm (cannot be run on a single GH200 GPU with 96GB memory), and does not have rigorous theoretical guarantees as our algorithm. This trade-off between performance and memory costs is similar to the trade-offs between DPO and RLHF that have been reported in the literature (Rafailov et al., 2023; Xu et al., 2024).

## 7 CONCLUSION

In this work, we study the constrained alignment problem for LLMs, which aims to maximize the reward while constraining the cost to stay below a threshold. We develop a novel primal-dual DPO approach for the offline and online data settings. Our approach adopts a rearranged Lagrangian DPO training objective, utilizing the reward information provided by a model trained using standard DPO. We establish suboptimality and constraint violation guarantees, and provide experimental results on the PKU-SafeRLHF dataset (Dai et al., 2024) to validate the effectiveness of our approach.

There are several interesting directions for future work. One direction is to extend our theoretical results to the policy parameterization setting. The challenge is that under policy parameterization, the constrained alignment problem can be non-convex. Another direction is to investigate stricter cost constraints, e.g., per-response constraints, which is challenging to tackle using neural networks.

ETHICS STATEMENT

This paper studies the alignment of LLMs to enhance safety or impose certain constraints. The data used in experiments may contain harmful or offensive content.

REPRODUCIBILITY STATEMENT

This paper provides theoretical guarantees and experimental results for the proposed primal-dual DPO approach. All results are reproducible. For theoretical guarantees, the assumptions required are stated in Sections 3 and 4.2 and Appendix D, and all proofs are presented in Appendix. For experimental results, the experimental setup is described in Section 6 and Appendix B, and the code is provided in supplementary materials.

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

# APPENDIX

## A A FULL REVIEW OF RELATED WORK

In this section, we give a more detailed review of related work.

With the extensive application of LLMs, the alignment of LLMs has received widespread attention in the AI community, which aims to make LLMs align with human preference and values, and become more helpful and harmless. RLHF (Christiano et al., 2017; Ouyang et al., 2022) and DPO (Rafailov et al., 2023) are two main algorithmic frameworks for LLM alignment. RLHF first trains a reward model, and then applies RL algorithms with the learned reward model to fine-tune LLMs. DPO directly fine-tune LLMs using preference data, without explicitly training a reward model.

While LLMs have achieved a remarkable success, they may also generate harmful and fabricated content (Gehman et al., 2020; Lin et al., 2021; Wei et al., 2023). Recently, there are several works studying safety or constrained alignment of LLMs. The most related works to ours are (Dai et al., 2024; Liu et al., 2024b; Wachi et al., 2024; Huang et al., 2024; Zhang et al., 2025; Kim et al., 2025). Dai et al. (2024) proposed a safe RLHF framework, which considers maximizing the reward while restricting the cost to be no larger than a threshold. Their approach first trains a reward model and a cost model on reward and cost preference data, respectively, and then applies an RL algorithm, PPO (Schulman et al., 2017), to maximize the Lagrangian function constituted by the learned reward and cost functions. Liu et al. (2024b) regenerated preference data according to the Bradley-Terry model (Bradley & Terry, 1952) with the Lagrangian function using trained reward and cost models, and then performed the standard DPO algorithm (Rafailov et al., 2023) on these regenerated data. Wachi et al. (2024) observed a relationship between the optimal policy of maximizing the Lagrangian function and the optimal policy of maximizing the reward function, and then applied DPO combined with this observation. The algorithm in (Wachi et al., 2024) requires prior knowledge of the optimal Lagrange multiplier, and their theoretical results depend on the deviation between the used Lagrange multiplier and optimal Lagrange multiplier, which can be unbounded. Kim et al. (2025) reordered preference data if the preferred response (in terms of helpfulness) is unsafe and the dispreferred response is safe, and then ran DPO on these reordered data. Their algorithm is inefficient in cost information learning, and thus performs worse than our algorithm in experiments (see Section 6). Huang et al. (2024); Zhang et al. (2025) investigated the constrained alignment problem from the perspective of dual optimization. Huang et al. (2024) derived an explicit form of the dual function, which only involves the SFT model and does not need to compute the optimal policy to the Lagrangian function. Leveraging this derivation, their algorithms use offline data generated by the SFT model to first learn the optimal Lagrange multiplier, which avoids the expensive computation of evaluating the optimal policy at each step, and then compute the optimal policy only after it learns the optimal Lagrange multiplier. However, the algorithms in (Huang et al., 2024) require trained reward and cost models, or need to train the reward-aligned and cost-aligned language models in advance, while our algorithm only needs to train the reward-aligned language model in advance. Zhang et al. (2025) generalized the algorithms in (Huang et al., 2024) to the multi-shot scheme with policy parameterization, and focused on analyzing the primal-dual gap brought by policy parameterizaiton.

(Reviewer kvKV) Table 1 summarizes the assumptions, the number of required trained and loaded models, and theoretical guarantees on the output policy of our work and the most related works. Our algorithm just needs to train and load two models, which significantly reduces memory costs compared to prior works (Dai et al., 2024; Liu et al., 2024b; Huang et al., 2024; Zhang et al., 2025). While Wachi et al. (2024) also only needed to train two models, they required prior knowledge of the optimal Lagrange multiplier, and their theoretical results have an unbounded term due to the lack of schemes to learn the optimal Lagrange multiplier. While Kim et al. (2025) needed to train only one model, their algorithm has worse empirical performance than ours, and they did not provide theoretical guarantees on the output policy.

Regarding theoretical results, since the needed assumptions and main focuses of the analyses in our work and prior works (Wachi et al., 2024; Huang et al., 2024; Zhang et al., 2025) are different, the results cannot be directly compared. The results in Wachi et al. (2024) have an unbounded term of the gap between the used Lagrange multiplier and optimal Lagrange multiplier, since their algorithm does not contain any scheme to learn the optimal Lagrange multiplier. The results in (Huang et al., 2024) rely on the assumption that the optimal policy is feasible under their used cost model, which is

Table 1: Summary of the assumptions, the number of required trained and loaded models, and theoretical guarantees on the output policy in our work and the most related works. In the last column, $r$ and $c$ denote the reward and cost models, respectively, and $\pi^{\mathrm{r}}$, $\pi^{\mathrm{c}}$ and $\pi^{\mathrm{r,c}}$ denote the reward-aligned, cost-aligned and reward-cost-aligned language models, respectively.

| Algorithms | Assumptions | # The required trained and loaded models | Theoretical guarantees on the output policy |
|---|---|---|---|
| PD-DPO (ours) | (i) Bradley-Terry model (ii) Slater's condition | 2: $\pi^{\mathrm{r}}, \pi^{\mathrm{r,c}}$ | Yes[1] |
| Safe RLHF (Dai et al., 2024) | Bradley-Terry model | 3: $r, c, \pi^{\mathrm{r,c}}$ | No |
| C-DPO (Liu et al., 2024b) | Bradley-Terry model | 3: $r, c, \pi^{\mathrm{r,c}}$ | No |
| MoCAN, PeCAN (Huang et al., 2024) | (i) Bradley-Terry model (ii) Slater's condition (iii) $\pi^*$ is feasible under $c$ | 3: $r, c, \pi^{\mathrm{r,c}}$ ($\pi^{\mathrm{r}}, \pi^{\mathrm{c}}, \pi^{\mathrm{r,c}}$) | Yes[2] |
| CAID (Zhang et al., 2025) | (i) Bradley-Terry model (ii) Slater's condition (iii) Boundedness of the policy parameterization gap (iv) Strong convexity of the dual function | 3: $r, c, \pi^{\mathrm{r,c}}$ | Yes[3] |
| SACPO (Wachi et al., 2024) | (i) Bradley-Terry model (ii) Slater's condition (iii) Knowledge of $\lambda^*$ | 2: $\pi^{\mathrm{r}}, \pi^{\mathrm{r,c}}$ | Yes[4] |
| SafeDPO (Kim et al., 2025) | (i) Bradley-Terry model (ii) $\forall x, \exists \bar{y}$ s.t. $c^*(x, \bar{y}) \leq 0$ and $\pi_{\mathrm{ref}}(\bar{y}|x) > 0$ | 1: $\pi^{\mathrm{r,c}}$ | No |

[1] Our results do not require extra assumptions other than the standard Bradley-Terry model and Slater's condition. In addition, our results for the online exploration version of algorithm PD-DPO get rid of the dependence on preference data coverage (Theorem 2).

[2] The results in (Huang et al., 2024) rely on the assumption that the optimal policy $\pi^*$ is feasible under the used cost model $c$.

[3] The results in (Zhang et al., 2025) focus on analyzing the primal-dual gap brought by policy parameterization, instead of the error due to learning reward and cost functions from preference data as in our work, (Wachi et al., 2024) and (Huang et al., 2024).

[4] The results in (Wachi et al., 2024) have an unbounded term of the gap between the used Lagrange multiplier and optimal Lagrange multiplier $\lambda^*$.

hard to verify in practice. Zhang et al. (2025) focused on analyzing the primal-dual gap due to policy parameterization, instead of the error due to learning reward and cost functions from preference data as in our work and (Wachi et al., 2024; Huang et al., 2024). In contrast to prior works, our results do not require extra assumptions and remove the dependence on preference data coverage in the extended online exploration setting (Theorem 2).

There are also other works related to safety or constrained alignment of LLMs, e.g., (Zhou et al., 2023; Ji et al., 2024; Yang et al., 2024; Qi et al., 2025). Most of these works are empirical works, which did not provide theoretical guarantees on the output policy and are less related to our work.

## B  MORE EXPERIMENTAL DETAILS

In this section, we will describe more details of algorithm implementation and experimental setup. Our code is written based on the released code of prior safe RLHF work (Dai et al., 2024) on their GitHub website, and we also open source our code in supplementary materials.

In algorithm implementation, we implement our algorithm PD-DPO (Algorithm 1) without policy search constraints $\pi \in \Pi^{\mathrm{r}}$ in Line 1 and $\pi \in \Pi_k^{\mathrm{c}}$ in Line 3, since these two constraints are mainly used

Table 2: Hyper-parameters of our algorithm PD-DPO and the compared algorithms.

| Hyper-parameters | PD-DPO (ours) | SafeDPO | Beaver-v3.0 | SACPO and P-SACPO |
|---|---|---|---|---|
| $\beta$ | 0.1 | 0.1 | 0.01 | 0.1 ($\frac{\beta}{\lambda} = 0.025$) |
| epochs | 5 | 3 | 2 | 3 |
| max_length | 512 | 512 | 512 | 512 |
| per_device_train_batch_size | 8 | 8 | 16 | 16 |
| per_device_eval_batch_size | 8 | 8 | 16 | 16 |
| gradient_accumulation_steps | 1 | 1 | 1 | 2 |
| gradient_checkpointing | True | True | True | True |
| lr | 2e-5 | 1e-6 | 2e-5 | 2e-5 |
| lr_scheduler_type | cosine | cosine | cosine | cosine |
| lr_warmup_ratio | 0.03 | 0.03 | 0.03 | 0.03 |
| weight_decay | 0.05 | 0.05 | 0.1 | - |
| bf16 | True | True | True | True |
| tf32 | True | True | True | True |

for guaranteeing theoretical performance. In experiments, we set the Lagrange multiplier of algorithm PD-DPO as 5 to save computational costs and time due to our limited computational resources. We find that it works well in practice.

For the compared algorithms, we directly access the released models `Alpaca-7b-reproduced`, `Beaver-v3.0` (Dai et al., 2024), `SACPO` and `P-SACPO` (Wachi et al., 2024) via their Hugging Face websites for evaluation, and do not tune their algorithms. Thus, the hyper-parameters of their algorithms are the same as reported in their paper (Dai et al., 2024). For algorithm `SafeDPO` (Kim et al., 2025), to guarantee fair comparison, we run it on the SFT model `Alpaca-7b-reproduced`, without performing additional supervised fine-tuning on pairwise preference data as described in (Kim et al., 2025). We present the hyper-parameters of our algorithm PD-DPO and the compared algorithms in Table 2.

## C   PROOFS FOR ALGORITHM PD-DPO

In this section, we present the proofs for algorithm PD-DPO, including the proofs for the connection to the RLHF-based procedure, suboptimality, and constraint violation.

We note that our proofs for the connection between our DPO-based procedure and the RLHF-based procedure (Theorems 3, 4, 5 and 6) follow the analysis of Proposition 4 in (Azar et al., 2024). We extend their analysis to the setting with constrained policy search ranges and a Lagrangian objective.

### C.1   CONNECTION BETWEEN OUR DPO-BASED PROCEDURE AND THE RLHF-BASED PROCEDURE

We first give a result which builds a bridge between standard DPO and standard RLHF with constrained policy search ranges.

Let $\mathcal{R} := [-R_{\max}, R_{\max}]$ and $\mathcal{C} := [-C_{\max}, C_{\max}]$. Define the following problem which first learns a reward model and then finds the optimal policy to maximize the learned reward function:

$$\hat{r} \leftarrow \min_{r \in \mathcal{R}} -\frac{1}{N^{\mathrm{r}}} \sum_{i=1}^{N^{\mathrm{r}}} \log \sigma \left( r(x_i, y_i^{\mathrm{rw}}) - r(x_i, y_i^{\mathrm{rl}}) \right) \tag{19}$$

$$\max_{\pi} \ \mathbb{E}_{x \sim \mathcal{D}^{\mathrm{p}}} \left[ \mathbb{E}_{y \sim \pi(\cdot|x)} \left[ \hat{r}(x, y) \right] - \beta \cdot \mathrm{KL}(\pi(\cdot|x) \| \pi_{\mathrm{ref}}(\cdot|x)) \right] \tag{20}$$

**Theorem 3** (Connection between Standard DPO and Standard RLHF with Constrained Policy Ranges). *Problems Eqs. (15) and* (20) *have the same set of optimal solutions.*

*Proof.* **Step (i).** First, we prove that if $\pi$ is an optimal solution to Eq. (20), then $\pi$ is also an optimal solution to Eq. (15).

If $\hat{r} \in \mathcal{R}$ is an optimal solution to Eq. (19), then $\pi^*_{\hat{r}} \in \Pi^{\mathrm{r}}$ (as defined in Eq. (4)) is an optimal solution to Eq. (20). We have that $\pi^*_{\hat{r}}$ is also an optimal solution to Eq. (15). Otherwise, there exists another $\pi' \in \Pi^{\mathrm{r}}$ which achieves a smaller objective value in Eq. (15). Then, there must exist a $r' \in \mathcal{R}$ which satisfies that

$$\pi'(y|x) = \frac{\pi_{\mathrm{ref}}(y|x) \cdot \exp\left(\frac{1}{\beta} r'(x,y)\right)}{\underbrace{\sum_{y' \in \mathcal{Y}} \pi_{\mathrm{ref}}(y'|x) \cdot \exp\left(\frac{1}{\beta} r'(x,y')\right)}_{:=Z_{r'}(x)}},$$

i.e.,

$$r'(x,y) = \beta \log \frac{\pi'(y|x)}{\pi_{\mathrm{ref}}(y|x)} + \beta \log Z_{r'}(x),$$

and the objective value in Eq. (19) achieved by $r'$,

$$-\frac{1}{N^{\mathrm{r}}} \sum_{i=1}^{N^{\mathrm{r}}} \log \sigma\left(\beta \log \frac{\pi'(y_i^{\mathrm{rw}}|x_i)}{\pi_{\mathrm{ref}}(y_i^{\mathrm{rw}}|x_i)} + \beta \log Z_{r'}(x_i) - \left(\beta \log \frac{\pi'(y_i^{\mathrm{rl}}|x_i)}{\pi_{\mathrm{ref}}(y_i^{\mathrm{rl}}|x_i)} + \beta \log Z_{r'}(x_i)\right)\right),$$

is smaller than that achieved by $\hat{r}$, which contradicts the supposition that $\hat{r}$ is the optimal solution to Eq. (19).

**Step (ii).** Next, we prove that if $\pi$ is an optimal solution to Eq. (15), then $\pi$ is also an optimal solution to Eq. (20).

If $\tilde{\pi} \in \Pi^{\mathrm{r}}$ is an optimal solution to Eq. (15), then there exists a $\tilde{r} \in \mathcal{R}$ which satisfies

$$\tilde{\pi}(y|x) = \frac{\pi_{\mathrm{ref}}(y|x) \cdot \exp\left(\frac{1}{\beta} \tilde{r}(x,y)\right)}{\sum_{y' \in \mathcal{Y}} \pi_{\mathrm{ref}}(y'|x) \cdot \exp\left(\frac{1}{\beta} \tilde{r}(x,y')\right)},$$

i.e.,

$$\tilde{r}(x,y) = \beta \log \frac{\tilde{\pi}(y|x)}{\pi_{\mathrm{ref}}(y|x)} + \beta \log Z_{\tilde{r}}(x).$$

We have that $\tilde{r}$ achieves the optimal value in Eq. (19),

$$-\frac{1}{N^{\mathrm{r}}} \sum_{i=1}^{N^{\mathrm{r}}} \log \sigma\left(\beta \log \frac{\tilde{\pi}(y_i^{\mathrm{rw}}|x_i)}{\pi_{\mathrm{ref}}(y_i^{\mathrm{rw}}|x_i)} + \beta \log Z_{\tilde{r}}(x_i) - \left(\beta \log \frac{\tilde{\pi}(y_i^{\mathrm{rl}}|x_i)}{\pi_{\mathrm{ref}}(y_i^{\mathrm{rl}}|x_i)} + \beta \log Z_{\tilde{r}}(x_i)\right)\right). \quad (21)$$

Otherwise, there exists another $r' \in \mathcal{R}$ and then there exists a $\pi' = \pi^*_{\hat{r}} \in \Pi^{\mathrm{r}}$ which gives a smaller objective value than $\tilde{\pi}$ in Eq. (21). Thus, $\tilde{r}$ achieves the optimal value in Eq. (19). Then, the optimal solution to Eq. (20) under cost model $\tilde{r}$ is

$$\pi(y|x) \propto \pi_{\mathrm{ref}}(y|x) \cdot \exp\left(\frac{1}{\beta} \tilde{r}(x,y)\right)$$

$$\overset{(a)}{\propto} \pi_{\mathrm{ref}}(y|x) \cdot \exp\left(\frac{1}{\beta}\left(\beta \log \frac{\tilde{\pi}(y|x)}{\pi_{\mathrm{ref}}(y|x)} + \beta \log Z_{\tilde{r}}(x)\right)\right)$$

$$\propto \pi_{\mathrm{ref}}(y|x) \cdot \exp\left(\log \frac{\tilde{\pi}(y|x)}{\pi_{\mathrm{ref}}(y|x)}\right)$$

$$= \tilde{\pi}(y|x),$$

where (a) uses Eq. (5).

Therefore, $\tilde{\pi}$ is also an optimal solution to Eq. (20). $\square$

In the following, we provide a result which builds a connection between our rearranged Lagrangian DPO objective and the safe RLHF objective.

**Theorem 4** (Connection between Our Rearranged Lagrangian DPO and Safe RLHF). *For any $k \geq 0$, problem Eq. (16) and the following problem*

$$\hat{c} \leftarrow \min_{c \in [-C_{\max}, C_{\max}]} -\frac{1}{N^c} \sum_{i=1}^{N^c} \log \sigma \left( c(x_i^c, y_i^{cw}) - c(x_i^c, y_i^{cl}) \right), \tag{22}$$

$$\max_{\pi} \mathbb{E}_{x \sim \mathcal{D}^p} \left[ \mathbb{E}_{y \sim \pi(\cdot|x)} \left[ \hat{r}(x, y) - \lambda_k \cdot \hat{c}(x, y) \right] - \beta \cdot \mathrm{KL}(\pi(\cdot|x) \| \pi_{\mathrm{ref}}(\cdot|x)) \right]. \tag{23}$$

*have the same set of optimal solutions.*

Theorem 4 demonstrates that our rearranged Lagrangian DPO objective is an effective and alternative way to learn the optimal policy of maximizing the Lagrangian function, while enjoying the advantage of memory and computational efficiency.

*Proof of Theorem 4.* First, note that for any $\hat{c}$, the optimal solution to Eq. (23) is

$$\pi_{\hat{r}-\lambda_k \hat{c}}^*(y|x) = \frac{\pi_{\mathrm{ref}}(y|x) \cdot \exp\left(\frac{1}{\beta} \left(\hat{r}(x,y) - \lambda_k \cdot \hat{c}(x,y)\right)\right)}{\underbrace{\sum_{y' \in \mathcal{Y}} \pi_{\mathrm{ref}}(y'|x) \cdot \exp\left(\frac{1}{\beta} \left(\hat{r}(x,y') - \lambda_k \cdot \hat{c}(x,y')\right)\right)}_{:=Z_{\hat{r}-\lambda_k \hat{c}}(x)}}, \quad \forall x \in \mathcal{X}. \tag{24}$$

Then, we have

$$\hat{c}(x, y) = \frac{1}{\lambda_k} \left( \hat{r}(x,y) - \beta \log \frac{\pi_{\hat{r}-\lambda_k \hat{c}}^*(y|x)}{\pi_{\mathrm{ref}}(y|x)} - \beta \log Z_{\hat{r}-\lambda_k \hat{c}}(x) \right)$$

$$\overset{(a)}{=} \frac{1}{\lambda_k} \left( \beta \log \frac{\pi_{\hat{r}}^*(y|x)}{\pi_{\mathrm{ref}}(y|x)} + \beta \log Z_{\hat{r}}(x) - \beta \log \frac{\pi_{\hat{r}-\lambda_k \hat{c}}^*(y|x)}{\pi_{\mathrm{ref}}(y|x)} - \beta \log Z_{\hat{r}-\lambda_k \hat{c}}(x) \right),$$

where equality (a) uses Eq. (5).

The proof consists of two steps.

**Step (i).** First, we prove that if $\pi$ is an optimal solution to Eq. (23), then $\pi$ is also an optimal solution to Eq. (16).

If $\hat{c} \in \mathcal{C}$ is an optimal solution to Eq. (22), then $\pi_{\hat{r}-\lambda_k \hat{c}}^* \in \Pi_k^c$ (as shown in Eq. (24)) is an optimal solution to Eq. (23). We have that $\pi_{\hat{r}-\lambda_k \hat{c}}^*$ is also an optimal solution to Eq. (16). Otherwise, there exists another $\pi' \in \Pi_k^c$ which achieves a smaller objective value in Eq. (16). Then, there must exist a $c' \in \mathcal{C}$ which satisfies that

$$\pi'(y|x) = \frac{\pi_{\mathrm{ref}}(y|x) \cdot \exp\left(\frac{1}{\beta} \left(\beta \log \frac{\pi_{\hat{r}}^*(y|x)}{\pi_{\mathrm{ref}}(y|x)} - \lambda_k \cdot c'(x,y)\right)\right)}{\underbrace{\sum_{y' \in \mathcal{Y}} \pi_{\mathrm{ref}}(y'|x) \cdot \exp\left(\frac{1}{\beta} \left(\beta \log \frac{\pi_{\hat{r}}^*(y|x)}{\pi_{\mathrm{ref}}(y|x)} - \lambda_k \cdot c'(x,y')\right)\right)}_{:=Z_{\beta \log \frac{\pi_{\hat{r}}^*}{\pi_{\mathrm{ref}}} - \lambda_k c'}(x)}},$$

i.e.,

$$c'(x, y) = \frac{1}{\lambda_k} \left( \beta \log \frac{\pi_{\hat{r}}^*(y|x)}{\pi_{\mathrm{ref}}(y|x)} - \beta \log \frac{\pi'(y|x)}{\pi_{\mathrm{ref}}(y|x)} - \beta \log Z_{\beta \log \frac{\pi_{\hat{r}}^*}{\pi_{\mathrm{ref}}} - \lambda_k c'}(x) \right),$$

and the objective value in Eq. (22) achieved by $c'$,

$$-\frac{1}{N^c} \sum_{i=1}^{N^c} \log \sigma \left( \frac{1}{\lambda_k} \left( \beta \log \frac{\pi_{\hat{r}}^*(y_i^{cw}|x_i)}{\pi_{\mathrm{ref}}(y_i^{cw}|x_i)} - \beta \log \frac{\pi'(y_i^{cw}|x_i)}{\pi_{\mathrm{ref}}(y_i^{cw}|x_i)} - \beta \log Z_{\beta \log \frac{\pi_{\hat{r}}^*}{\pi_{\mathrm{ref}}} - \lambda_k c'}(x_i) \right) \right.$$

$$\left. - \frac{1}{\lambda_k} \left( \beta \log \frac{\pi_{\hat{r}}^*(y_i^{cl}|x_i)}{\pi_{\mathrm{ref}}(y_i^{cl}|x_i)} - \beta \log \frac{\pi'(y_i^{cl}|x_i)}{\pi_{\mathrm{ref}}(y_i^{cl}|x_i)} - \beta \log Z_{\beta \log \frac{\pi_{\hat{r}}^*}{\pi_{\mathrm{ref}}} - \lambda_k c'}(x_i) \right) \right),$$

is smaller than that achieved by $\hat{c}$, which contradicts the supposition that $\hat{c}$ is the optimal solution to Eq. (22).

**Step (ii).** Next, we prove that if $\pi$ is an optimal solution to Eq. (16), then $\pi$ is also an optimal solution to Eq. (23).

If $\pi_k \in \Pi_k^c$ is an optimal solution to Eq. (16), then there exists a $c_k \in \mathcal{C}$ which satisfies

$$\pi_k(y|x) = \frac{\pi_{\text{ref}}(y|x) \cdot \exp\left(\frac{1}{\beta}\left(\beta \log \frac{\pi_{\hat{r}}^*(y|x)}{\pi_{\text{ref}}(y|x)} - \lambda_k \cdot c_k(x,y)\right)\right)}{\sum_{y' \in \mathcal{Y}} \pi_{\text{ref}}(y'|x) \cdot \exp\left(\frac{1}{\beta}\left(\beta \log \frac{\pi_{\hat{r}}^*(y'|x)}{\pi_{\text{ref}}(y'|x)} - \lambda_k \cdot c_k(x,y')\right)\right)},$$

i.e.,

$$c_k(x,y) = \frac{1}{\lambda_k}\left(\beta \log \frac{\pi_{\hat{r}}^*(y|x)}{\pi_{\text{ref}}(y|x)} - \beta \log \frac{\pi_k(y|x)}{\pi_{\text{ref}}(y|x)} - \beta \log Z_{\beta \log \frac{\pi_{\hat{r}}^*}{\pi_{\text{ref}}} - \lambda_k c_k}(x)\right).$$

We have that $c_k$ achieves the optimal value in Eq. (22),

$$-\frac{1}{N^c} \sum_{i=1}^{N^c} \log \sigma \left(\frac{1}{\lambda_k}\left(\beta \log \frac{\pi_{\hat{r}}^*(y_i^{\text{cw}}|x_i)}{\pi_{\text{ref}}(y_i^{\text{cw}}|x_i)} - \beta \log \frac{\pi_k(y_i^{\text{cw}}|x_i)}{\pi_{\text{ref}}(y_i^{\text{cw}}|x_i)} - \beta \log Z_{\beta \log \frac{\pi_{\hat{r}}^*}{\pi_{\text{ref}}} - \lambda_k c_k}(x_i)\right)\right.$$

$$\left. - \frac{1}{\lambda_k}\left(\beta \log \frac{\pi_{\hat{r}}^*(y_i^{\text{cl}}|x_i)}{\pi_{\text{ref}}(y_i^{\text{cl}}|x_i)} - \beta \log \frac{\pi_k(y_i^{\text{cl}}|x_i)}{\pi_{\text{ref}}(y_i^{\text{cl}}|x_i)} - \beta \log Z_{\beta \log \frac{\pi_{\hat{r}}^*}{\pi_{\text{ref}}} - \lambda_k c_k}(x_i)\right)\right). \quad (25)$$

Otherwise, there exists another $c' \in \mathcal{C}$ and then there exists a $\pi' = \pi_{\hat{r}-\lambda_k c'}^* \in \Pi_k^c$ which gives a smaller objective value than $\pi_k$ in Eq. (25). Thus, $c_k$ achieves the optimal value in Eq. (22). Then, the optimal solution to Eq. (22) under cost model $c_k$ is

$$\pi(y|x) \propto \pi_{\text{ref}}(y|x) \cdot \exp\left(\frac{1}{\beta}\left(\hat{r}(x,y) - \beta \log \frac{\pi_{\hat{r}}^*(y|x)}{\pi_{\text{ref}}(y|x)} + \beta \log \frac{\pi_k(y|x)}{\pi_{\text{ref}}(y|x)}\right.\right.$$

$$\left.\left. + \beta \log Z_{\beta \log \frac{\pi_{\hat{r}}^*}{\pi_{\text{ref}}} - \lambda_k c_k}(x)\right)\right)$$

$$\overset{(a)}{\propto} \pi_{\text{ref}}(y|x) \cdot \exp\left(\frac{1}{\beta}\left(\beta \log Z_{\hat{r}}(x) + \beta \log \frac{\pi_k(y|x)}{\pi_{\text{ref}}(y|x)} + \beta \log Z_{\beta \log \frac{\pi_{\hat{r}}^*}{\pi_{\text{ref}}} - \lambda_k c_k}(x)\right)\right)$$

$$\propto \pi_{\text{ref}}(y|x) \cdot \exp\left(\frac{1}{\beta}\left(\beta \log \frac{\pi_k(y|x)}{\pi_{\text{ref}}(y|x)}\right)\right)$$

$$= \pi_k(y|x),$$

where (a) uses Eq. (5).

Therefore, $\pi_k$ is also an optimal solution to Eq. (23). $\qquad\square$

## C.2 Cost Estimation for Lagrangian Multiplier Update

In the following, we bound the estimation error between $\tilde{c}_k$ and $\mathbb{E}_{x \sim \mathcal{D}^p, y \sim \pi_k(\cdot|x)}[c^*(x,y)]$ in Line 4 of Algorithm PD-DPO.

Let $\delta' := \frac{\delta}{4}$. Define events

$$\mathcal{E} := \left\{ |\bar{Z}_i - \sigma(c^*(x_i, y_i))| \leq \sqrt{\frac{\log\left(\frac{2|\mathcal{X}||\mathcal{Y}|N^{\text{CE}}K}{\delta'}\right)}{M^{\text{CE}}}}, \ \forall i \in [N^{\text{CE}}], \forall k \in [K] \right\}, \quad (26)$$

$$\mathcal{F} := \left\{ \left|\frac{1}{N^{\text{CE}}} \sum_{i=1}^{N^{\text{CE}}} c^*(x_i, y_i) - \mathbb{E}_{x \sim \mathcal{D}^p, y \sim \pi_k(\cdot|x)}[c^*(x,y)]\right| \leq C_{\max}\sqrt{\frac{\log\left(\frac{2K}{\delta'}\right)}{N^{\text{CE}}}}, \ \forall k \in [K] \right\}. \quad (27)$$

**Lemma 1.** *It holds that*

$$\Pr\left[\mathcal{E}\right] \geq 1 - \delta',$$
$$\Pr\left[\mathcal{F}\right] \geq 1 - \delta'.$$

*Proof.* Using Hoeffding's inequality, for any $i \in [N^{\mathrm{CE}}]$, for any fixed $(x_i, y_i) = (x, y) \in \mathcal{X} \times \mathcal{Y}$, we have that with probability at least $1 - \tilde{\delta}$,

$$\left|\bar{Z}_i - \sigma(c^*(x_i, y_i))\right| \leq \sqrt{\frac{\log\left(\frac{2}{\tilde{\delta}}\right)}{M^{\mathrm{CE}}}}.$$

Taking a union bound over $(x, y) \in \mathcal{X} \times \mathcal{Y}$, $i \in [N^{\mathrm{CE}}]$ and $k \in [K]$, we can obtain the first statement.

Combining the fact that $c^*(x, y) \in [-C_{\max}, C_{\max}]$ for any $(x, y) \in \mathcal{X} \times \mathcal{Y}$, Hoeffding's inequality, and a union bound over $k \in [K]$, we can obtain the second statement. $\qquad\square$

**Lemma 2.** *Assume that event $\mathcal{E} \cap \mathcal{F}$ holds. Then, we have*

$$\left|\tilde{c}_k - \mathbb{E}_{x \sim \mathcal{D}^{\mathrm{p}}, y \sim \pi_k(\cdot|x)}[c^*(x, y)]\right| \leq C_{\max}\sqrt{\frac{\log\left(\frac{2K}{\delta'}\right)}{N^{\mathrm{CE}}}} + W\sqrt{\frac{\log\left(\frac{2|\mathcal{X}||\mathcal{Y}|N^{\mathrm{CE}}K}{\delta'}\right)}{M^{\mathrm{CE}}}},$$

*where*

$$W := \frac{1}{\left(\frac{1}{1+\exp(-C_{\max})} + \sqrt{\frac{\log\left(\frac{2|\mathcal{X}||\mathcal{Y}|N^{\mathrm{CE}}K}{\delta'}\right)}{M^{\mathrm{CE}}}}\right)\left(\frac{\exp(-C_{\max})}{1+\exp(-C_{\max})} - \sqrt{\frac{\log\left(\frac{2|\mathcal{X}||\mathcal{Y}|N^{\mathrm{CE}}K}{\delta'}\right)}{M^{\mathrm{CE}}}}\right)} \qquad (28)$$

*and $\delta' := \frac{\delta}{4}$.*

*Proof.* For any $i \in [N^{\mathrm{CE}}]$, we have

$$\left|\bar{Z}_i - \sigma(c^*(x_i, y_i))\right| \leq \sqrt{\frac{\log\left(\frac{2|\mathcal{X}||\mathcal{Y}|N^{\mathrm{CE}}K}{\delta'}\right)}{M^{\mathrm{CE}}}}.$$

Since $c^*(x, y) \in [-C_{\max}, C_{\max}]$ for any $(x, y) \in \mathcal{X} \times \mathcal{Y}$, we have

$$\sigma(-C_{\max}) - \sqrt{\frac{\log\left(\frac{2|\mathcal{X}||\mathcal{Y}|N^{\mathrm{CE}}K}{\delta'}\right)}{M^{\mathrm{CE}}}} \leq \bar{Z}_i \leq \sigma(C_{\max}) + \sqrt{\frac{\log\left(\frac{2|\mathcal{X}||\mathcal{Y}|N^{\mathrm{CE}}K}{\delta'}\right)}{M^{\mathrm{CE}}}}.$$

The derivative of $\sigma^{-1}(z)$ is $(\sigma^{-1})'(z) = \frac{1}{z(1-z)}$. For any $z$ lying between $\bar{Z}_i$ and $\sigma(c^*(x_i, y_i))$, we have

$$(\sigma^{-1})'(z) \leq \frac{1}{\left(\frac{1}{1+\exp(-C_{\max})} + \sqrt{\frac{\log\left(\frac{2|\mathcal{X}||\mathcal{Y}|N^{\mathrm{CE}}K}{\delta'}\right)}{M^{\mathrm{CE}}}}\right)\left(\frac{\exp(-C_{\max})}{1+\exp(-C_{\max})} - \sqrt{\frac{\log\left(\frac{2|\mathcal{X}||\mathcal{Y}|N^{\mathrm{CE}}K}{\delta'}\right)}{M^{\mathrm{CE}}}}\right)} := W.$$

According to the Lagrange's Mean Value Theorem, we have

$$\left|\sigma^{-1}(\bar{Z}_i) - c^*(x_i, y_i)\right| = \left|\sigma^{-1}(\bar{Z}_i) - \sigma^{-1}(\sigma(c^*(x_i, y_i)))\right|$$
$$\leq W\left|\bar{Z}_i - \sigma(c^*(x_i, y_i))\right|$$
$$\leq W\sqrt{\frac{\log\left(\frac{2|\mathcal{X}||\mathcal{Y}|N^{\mathrm{CE}}K}{\delta'}\right)}{M^{\mathrm{CE}}}}.$$

Hence, we have

$$c^*(x_i, y_i) - W\sqrt{\frac{\log\left(\frac{2|\mathcal{X}||\mathcal{Y}|N^{\mathrm{CE}}K}{\delta'}\right)}{M^{\mathrm{CE}}}} \leq \sigma^{-1}(\bar{Z}_i) \leq c^*(x_i, y_i) + W\sqrt{\frac{\log\left(\frac{2|\mathcal{X}||\mathcal{Y}|N^{\mathrm{CE}}K}{\delta'}\right)}{M^{\mathrm{CE}}}}.$$

Since the above argument holds for any $i \in [N^{\mathrm{CE}}]$, we have

$$\frac{1}{N^{\mathrm{CE}}}\sum_{i=1}^{N^{\mathrm{CE}}} c^*(x_i, y_i) - W\sqrt{\frac{\log\left(\frac{2|\mathcal{X}||\mathcal{Y}|N^{\mathrm{CE}}K}{\delta'}\right)}{M^{\mathrm{CE}}}} \leq \frac{1}{N^{\mathrm{CE}}}\sum_{i=1}^{N^{\mathrm{CE}}} \sigma^{-1}(\bar{Z}_i)$$

$$\leq \frac{1}{N^{\mathrm{CE}}}\sum_{i=1}^{N^{\mathrm{CE}}} c^*(x_i, y_i) + W\sqrt{\frac{\log\left(\frac{2|\mathcal{X}||\mathcal{Y}|N^{\mathrm{CE}}K}{\delta'}\right)}{M^{\mathrm{CE}}}}.$$

Combining with the definition of event $\mathcal{F}$, we have

$$\mathbb{E}_{x\sim\mathcal{D}^{\mathrm{p}}, y\sim\pi_k(\cdot|x)}\left[c^*(x,y)\right] - C_{\max}\sqrt{\frac{\log\left(\frac{2K}{\delta'}\right)}{N^{\mathrm{CE}}}} - W\sqrt{\frac{\log\left(\frac{2|\mathcal{X}||\mathcal{Y}|N^{\mathrm{CE}}K}{\delta'}\right)}{M^{\mathrm{CE}}}} \leq \frac{1}{N^{\mathrm{CE}}}\sum_{i=1}^{N^{\mathrm{CE}}} \sigma^{-1}(\bar{Z}_i) \leq$$

$$\mathbb{E}_{x\sim\mathcal{D}^{\mathrm{p}}, y\sim\pi_k(\cdot|x)}\left[c^*(x,y)\right] + C_{\max}\sqrt{\frac{\log\left(\frac{2K}{\delta'}\right)}{N^{\mathrm{CE}}}} + W\sqrt{\frac{\log\left(\frac{2|\mathcal{X}||\mathcal{Y}|N^{\mathrm{CE}}K}{\delta'}\right)}{M^{\mathrm{CE}}}}.$$

$\square$

### C.3 SUBOPTIMALITY AND CONSTRAINT VIOLATION

Now we give the proof of the suboptimality and constraint violation guarantees for Algorithm PD-DPO (Theorem 1).

Recall that for any $(x,y) \in \mathcal{X} \times \mathcal{Y}$, $\phi(x,y)$ denotes a $|\mathcal{X}||\mathcal{Y}|$-dimensional vector where the entry corresponding to $(x,y)$ is 1 and all other entries are 0. In addition, let

$$\Sigma_{\mathcal{D}^{\mathrm{r}}} := \sum_{i=1}^{N^{\mathrm{r}}} \left(\phi(x_i^{\mathrm{r}}, y_i^{\mathrm{rw}}) - \phi(x_i^{\mathrm{r}}, y_i^{\mathrm{rl}})\right)\left(\phi(x_i^{\mathrm{r}}, y_i^{\mathrm{rw}}) - \phi(x_i^{\mathrm{r}}, y_i^{\mathrm{rl}})\right)^\top,$$

$$\Sigma_{\mathcal{D}^{\mathrm{c}}} := \sum_{i=1}^{N^{\mathrm{c}}} \left(\phi(x_i^{\mathrm{c}}, y_i^{\mathrm{cw}}) - \phi(x_i^{\mathrm{c}}, y_i^{\mathrm{cl}})\right)\left(\phi(x_i^{\mathrm{c}}, y_i^{\mathrm{cw}}) - \phi(x_i^{\mathrm{c}}, y_i^{\mathrm{cl}})\right)^\top.$$

Define event

$$\mathcal{G} := \Bigg\{ |\hat{r}(x,y) - r^*(x,y)| \leq 4\,\|\phi(x,y)\|_{(\Sigma_{\mathcal{D}^{\mathrm{r}}} + \gamma I)^{-1}} \cdot$$

$$\sqrt{\left(\exp(R_{\max}) + \exp(-R_{\max}) + 2\right)^2\left(|\mathcal{X}||\mathcal{Y}| + \log\left(\frac{2}{\delta'}\right)\right) + \gamma(R_{\max})^2},$$

$$|\hat{c}(x,y) - c^*(x,y)| \leq 4\,\|\phi(x,y)\|_{(\Sigma_{\mathcal{D}^{\mathrm{c}}} + \gamma I)^{-1}} \cdot$$

$$\sqrt{\left(\exp(C_{\max}) + \exp(-C_{\max}) + 2\right)^2\left(|\mathcal{X}||\mathcal{Y}| + \log\left(\frac{2}{\delta'}\right)\right) + \gamma(C_{\max})^2},$$

$$\forall (x,y) \in \mathcal{X} \times \mathcal{Y} \Bigg\}.$$

**Lemma 3** (MLE Guarantee, Lemma 3.1 in (Zhu et al., 2023)). *It holds that*

$$\Pr[\mathcal{G}] \geq 1 - 2\delta'.$$

**Lemma 4.** *For any $k \geq 1$, we have*

$$f(\pi^*; \hat{r}) - f(\pi_k; \hat{r}) \leq -\lambda_k \cdot \mathbb{E}_{x \sim \mathcal{D}^\mathrm{p}, y \sim \pi_k(\cdot|x)}[\hat{c}(x,y)] + \lambda_k \left( \mathbb{E}_{x \sim \mathcal{D}^\mathrm{p}, y \sim \pi^*(\cdot|x)}[\hat{c}(x,y) - c^*(x,y)] \right).$$

*Proof.* It holds that

$$
\begin{aligned}
f(\pi^*; \hat{r}) &\overset{(a)}{\leq} f(\pi^*; \hat{r}) - \lambda_k \cdot \mathbb{E}_{x \sim \mathcal{D}^\mathrm{p}, y \sim \pi^*(\cdot|x)}[c^*(x,y)] \\
&= \mathbb{E}_{x \sim \mathcal{D}^\mathrm{p}} \left[ \mathbb{E}_{y \sim \pi^*(\cdot|x)} [\hat{r}(x,y) - \lambda_k \cdot \hat{c}(x,y)] - \beta \cdot \mathrm{KL}(\pi^*(\cdot|x) \| \pi_\mathrm{ref}(\cdot|x)) \right] \\
&\quad + \lambda_k \cdot \mathbb{E}_{x \sim \mathcal{D}^\mathrm{p}, y \sim \pi^*(\cdot|x)}[\hat{c}(x,y)] - \lambda_k \cdot \mathbb{E}_{x \sim \mathcal{D}^\mathrm{p}, y \sim \pi^*(\cdot|x)}[c^*(x,y)] \\
&\overset{(b)}{\leq} \mathbb{E}_{x \sim \mathcal{D}^\mathrm{p}} \left[ \mathbb{E}_{y \sim \pi_k(\cdot|x)} [\hat{r}(x,y) - \lambda_k \cdot \hat{c}(x,y)] - \beta \cdot \mathrm{KL}(\pi_k(\cdot|x) \| \pi_\mathrm{ref}(\cdot|x)) \right] \\
&\quad + \lambda_k \cdot \mathbb{E}_{x \sim \mathcal{D}^\mathrm{p}, y \sim \pi^*(\cdot|x)}[\hat{c}(x,y)] - \lambda_k \cdot \mathbb{E}_{x \sim \mathcal{D}^\mathrm{p}, y \sim \pi^*(\cdot|x)}[c^*(x,y)] \\
&= f(\pi_k; \hat{r}) - \lambda_k \cdot \mathbb{E}_{x \sim \mathcal{D}^\mathrm{p}, y \sim \pi_k(\cdot|x)}[\hat{c}(x,y)] + \lambda_k \left( \mathbb{E}_{x \sim \mathcal{D}^\mathrm{p}, y \sim \pi^*(\cdot|x)}[\hat{c}(x,y) - c^*(x,y)] \right),
\end{aligned}
$$

where inequality (a) uses the fact that $\lambda_k \geq 0$ and $\pi^*$ is feasible, and inequality (b) comes from the definition of $\pi_k$ and Theorem 4. $\qquad\square$

Now we prove Theorem 1.

*Proof of Theorem 1.* Recall that $\delta' := \frac{\delta}{4}$. Then, according to Lemmas 1 and 3, we have $\Pr[\mathcal{E} \cap \mathcal{F} \cap \mathcal{G}] \geq 1 - \delta$. Hence, it suffices to prove this theorem assuming that event $\mathcal{E} \cap \mathcal{F} \cap \mathcal{G}$ holds. In the following proof, we assume that event $\mathcal{E} \cap \mathcal{F} \cap \mathcal{G}$ holds.

For any $k \geq 1$ and $\bar{\lambda} \in [0, 2\rho]$, we have

$$
\begin{aligned}
\left( \lambda^{k+1} - \bar{\lambda} \right)^2 &= \left( \mathrm{Proj}_{[0,2\rho]} \left( \lambda_k + \eta_k \tilde{c}_k \right) - \mathrm{Proj}_{[0,2\rho]} \left( \bar{\lambda} \right) \right)^2 \\
&\overset{(a)}{\leq} \left( \lambda_k + \eta_k \tilde{c}_k - \bar{\lambda} \right)^2 \\
&= \left( \lambda_k - \bar{\lambda} \right)^2 + 2\eta_k \tilde{c}_k \left( \lambda_k - \bar{\lambda} \right) + (\eta_k)^2 \left( \tilde{c}_k \right)^2,
\end{aligned}
$$

where inequality (a) uses the nonexpansivity of the projection to $[0, 2\rho]$.

Summing the above inequality over $k = 1, \ldots, K$, we have

$$
0 \leq \left( \lambda_{K+1} - \bar{\lambda} \right)^2 \leq \left( \lambda_1 - \bar{\lambda} \right)^2 + \sum_{k=1}^{K} 2\eta_k \cdot \mathbb{E}_{x \sim \mathcal{D}^\mathrm{p}, y \sim \pi_k(\cdot|x)}[c^*(x,y)] \cdot \left( \lambda_k - \bar{\lambda} \right)
$$

$$
- \sum_{k=1}^{K} 2\eta_k \cdot \mathbb{E}_{x \sim \mathcal{D}^\mathrm{p}, y \sim \pi_k(\cdot|x)}[c^*(x,y)] \cdot \left( \lambda_k - \bar{\lambda} \right) + \sum_{k=1}^{K} 2\eta_k \tilde{c}_k \left( \lambda_k - \bar{\lambda} \right) + \sum_{k=1}^{K} (\eta_k)^2 \left( \tilde{c}_k \right)^2.
$$

Hence, we have

$$
\sum_{k=1}^{K} 2\eta_k \cdot \mathbb{E}_{x \sim \mathcal{D}^\mathrm{p}, y \sim \pi_k(\cdot|x)}[c^*(x,y)] \cdot \bar{\lambda} - \sum_{k=1}^{K} 2\eta_k \cdot \mathbb{E}_{x \sim \mathcal{D}^\mathrm{p}, y \sim \pi_k(\cdot|x)}[\hat{c}(x,y)] \cdot \lambda_k
$$

$$
\leq \left( \lambda_1 - \bar{\lambda} \right)^2 + \sum_{k=1}^{K} 2\eta_k \lambda_k \cdot \mathbb{E}_{x \sim \mathcal{D}^\mathrm{p}, y \sim \pi_k(\cdot|x)}[c^*(x,y) - \hat{c}(x,y)]
$$

$$
+ \sum_{k=1}^{K} 2\eta_k \left( \lambda_k - \bar{\lambda} \right) \left( \tilde{c}_k - \mathbb{E}_{x \sim \mathcal{D}^\mathrm{p}, y \sim \pi_k(\cdot|x)}[c^*(x,y)] \right) + \sum_{k=1}^{K} (\eta_k)^2 \left( \tilde{c}_k \right)^2.
$$

Using Lemma 4, we have

$$
\sum_{k=1}^{K} 2\eta_k \left( \mathbb{E}_{x \sim \mathcal{D}^\mathrm{p}, y \sim \pi_k(\cdot|x)}[c^*(x,y)] \cdot \bar{\lambda} + f(\pi^*; \hat{r}) - f(\pi_k; \hat{r}) \right.
$$

$$- \lambda_k \cdot \mathbb{E}_{x \sim \mathcal{D}^\mathrm{p}, y \sim \pi^*(\cdot|x)}[\hat{c}(x,y) - c^*(x,y)]\Big)$$

$$\leq \left(\lambda_1 - \bar{\lambda}\right)^2 + \sum_{k=1}^{K} \left(\eta_k\right)^2 \left(\tilde{c}_k\right)^2 + \sum_{k=1}^{K} 2\eta_k \lambda_k \cdot \mathbb{E}_{x \sim \mathcal{D}^\mathrm{p}, y \sim \pi_k(\cdot|x)}[c^*(x,y) - \hat{c}(x,y)]$$

$$+ \sum_{k=1}^{K} 2\eta_k \left(\lambda_k - \bar{\lambda}\right) \left(\tilde{c}_k - \mathbb{E}_{x \sim \mathcal{D}^\mathrm{p}, y \sim \pi_k(\cdot|x)}[c^*(x,y)]\right).$$

Recall that $\eta_k = \eta$. Then, we have

$$\sum_{k=1}^{K} \left(f(\pi^*) - f(\pi_k)\right) + \bar{\lambda} \sum_{k=1}^{K} \mathbb{E}_{x \sim \mathcal{D}^\mathrm{p}, y \sim \pi_k(\cdot|x)}[c^*(x,y)]$$

$$\leq \frac{1}{2\eta} \left(\lambda_1 - \bar{\lambda}\right)^2 + \frac{\eta}{2} \sum_{k=1}^{K} \left(\tilde{c}_k\right)^2 + \sum_{k=1}^{K} \lambda_k \cdot \mathbb{E}_{x \sim \mathcal{D}^\mathrm{p}, y \sim \pi_k(\cdot|x)}[c^*(x,y) - \hat{c}(x,y)]$$

$$+ \sum_{k=1}^{K} \left(\lambda_k - \bar{\lambda}\right) \left(\tilde{c}_k - \mathbb{E}_{x \sim \mathcal{D}^\mathrm{p}, y \sim \pi_k(\cdot|x)}[c^*(x,y)]\right)$$

$$+ \sum_{k=1}^{K} \lambda_k \cdot \mathbb{E}_{x \sim \mathcal{D}^\mathrm{p}, y \sim \pi^*(\cdot|x)}[\hat{c}(x,y) - c^*(x,y)]$$

$$+ \sum_{k=1}^{K} \left(f(\pi^*) - f(\pi^*; \hat{r})\right) - \sum_{k=1}^{K} \left(f(\pi_k) - f(\pi_k; \hat{r})\right)$$

$$\leq \frac{1}{2\eta} \left(\lambda_1 - \bar{\lambda}\right)^2 + \frac{\eta(C_{\max})^2 K}{2} + \sum_{k=1}^{K} \lambda_k \cdot \mathbb{E}_{x \sim \mathcal{D}^\mathrm{p}, y \sim \pi_k(\cdot|x)}[c^*(x,y) - \hat{c}(x,y)]$$

$$+ \sum_{k=1}^{K} \left(\lambda_k - \bar{\lambda}\right) \left(\tilde{c}_k - \mathbb{E}_{x \sim \mathcal{D}^\mathrm{p}, y \sim \pi_k(\cdot|x)}[c^*(x,y)]\right)$$

$$+ \sum_{k=1}^{K} \lambda_k \cdot \mathbb{E}_{x \sim \mathcal{D}^\mathrm{p}, y \sim \pi^*(\cdot|x)}[\hat{c}(x,y) - c^*(x,y)]$$

$$+ K \cdot \mathbb{E}_{x \sim \mathcal{D}^\mathrm{p}, y \sim \pi^*(\cdot|x)}[r^*(x,y) - \hat{r}(x,y)] - \sum_{k=1}^{K} \mathbb{E}_{x \sim \mathcal{D}^\mathrm{p}, y \sim \pi_k(\cdot|x)}[r^*(x,y) - \hat{r}(x,y)].$$

Let $\bar{\lambda} = 0$. Recall that $\pi_K^{\mathrm{out}}$ is the uniform policy over $\pi_1, \ldots, \pi_K$ and $\eta := \frac{\lambda_1}{C_{\max}\sqrt{K}}$. Then, we have

$$f(\pi^*) - f(\pi_K^{\mathrm{out}})$$

$$= \frac{1}{K} \sum_{k=1}^{K} \left(f(\pi^*) - f(\pi_k)\right)$$

$$= O\left(\frac{\lambda_1 C_{\max}}{\sqrt{K}} + \rho C_{\max}\sqrt{\frac{\log\left(\frac{1}{\delta}\right)}{N^{\mathrm{CE}}}} + \rho W \sqrt{\frac{\log\left(\frac{|\mathcal{X}||\mathcal{Y}|N^{\mathrm{CE}}}{\delta}\right)}{M^{\mathrm{CE}}}}\right)$$

$$+ \rho\left(\mathbb{E}_{x \sim \mathcal{D}^\mathrm{p}, y \sim \pi^*(\cdot|x)}\left[\|\phi(x,y)\|_{(\Sigma_{\mathcal{D}^\mathrm{c}}+\gamma I)^{-1}}\right] + \frac{1}{K}\sum_{k=1}^{K} \mathbb{E}_{x \sim \mathcal{D}^\mathrm{p}, y \sim \pi_k(\cdot|x)}\left[\|\phi(x,y)\|_{(\Sigma_{\mathcal{D}^\mathrm{c}}+\gamma I)^{-1}}\right]\right).$$

$$\sqrt{\left(\exp\left(C_{\max}\right) + \exp\left(-C_{\max}\right) + 2\right)^2 \left(|\mathcal{X}||\mathcal{Y}| + \log\left(\frac{1}{\delta}\right)\right) + \gamma(C_{\max})^2}$$

$$+ \left(\mathbb{E}_{x \sim \mathcal{D}^\mathrm{p}, y \sim \pi^*(\cdot|x)}\left[\|\phi(x,y)\|_{(\Sigma_{\mathcal{D}^\mathrm{r}}+\gamma I)^{-1}}\right] + \frac{1}{K}\sum_{k=1}^{K} \mathbb{E}_{x \sim \mathcal{D}^\mathrm{p}, y \sim \pi_k(\cdot|x)}\left[\|\phi(x,y)\|_{(\Sigma_{\mathcal{D}^\mathrm{r}}+\gamma I)^{-1}}\right]\right).$$

$$\sqrt{(\exp{(R_{\max})} + \exp{(-R_{\max})} + 2)^2 \left(|\mathcal{X}||\mathcal{Y}| + \log{\left(\frac{1}{\delta}\right)}\right) + \gamma(R_{\max})^2}.$$

Let $\bar{\lambda} = 2\rho$. Then, we have

$$f(\pi^*) - f(\pi_K^{\text{out}}) + 2\rho\mathbb{E}_{x\sim\mathcal{D}^{\text{p}}, y\sim\pi_K^{\text{out}}(\cdot|x)}[c^*(x,y)]$$

$$= \frac{1}{K}\sum_{k=1}^{K}(f(\pi^*) - f(\pi_k)) + \frac{2\rho}{K}\sum_{k=1}^{K}\mathbb{E}_{x\sim\mathcal{D}^{\text{p}}, y\sim\pi_k(\cdot|x)}[c^*(x,y)].$$

If $\frac{1}{K}\sum_{k=1}^{K}\mathbb{E}_{x\sim\mathcal{D}^{\text{p}}, y\sim\pi_k(\cdot|x)}[c^*(x,y)] \leq 0$, the second statement of the theorem naturally holds; Otherwise, we can replace the term $2\rho\mathbb{E}_{x\sim\mathcal{D}^{\text{p}}, y\sim\pi_K^{\text{out}}(\cdot|x)}[c^*(x,y)]$ by $2\rho[\mathbb{E}_{x\sim\mathcal{D}^{\text{p}}, y\sim\pi_K^{\text{out}}(\cdot|x)}[c^*(x,y)]]_+$ in the above inequality. Then, using Corollary 1 and Lemma 10, we obtain

$$\mathbb{E}_{x\sim\mathcal{D}^{\text{p}}, y\sim\pi_K^{\text{out}}(\cdot|x)}[c^*(x,y)]$$

$$= O\left(\frac{C_{\max}}{\rho\sqrt{K}}\left(\frac{(\lambda_1 - 2\rho)^2}{\lambda_1} + \lambda_1\right) + C_{\max}\sqrt{\frac{\log{\left(\frac{1}{\delta}\right)}}{N^{\text{CE}}}} + W\sqrt{\frac{\log{\left(\frac{|\mathcal{X}||\mathcal{Y}|N^{\text{CE}}}{\delta}\right)}}{M^{\text{CE}}}}\right.$$

$$+ \left(\mathbb{E}_{x\sim\mathcal{D}^{\text{p}}, y\sim\pi^*(\cdot|x)}\left[\|\phi(x,y)\|_{(\Sigma_{\mathcal{D}^{\text{c}}} + \gamma I)^{-1}}\right] + \frac{1}{K}\sum_{k=1}^{K}\mathbb{E}_{x\sim\mathcal{D}^{\text{p}}, y\sim\pi_k(\cdot|x)}\left[\|\phi(x,y)\|_{(\Sigma_{\mathcal{D}^{\text{c}}} + \gamma I)^{-1}}\right]\right).$$

$$\sqrt{(\exp{(C_{\max})} + \exp{(-C_{\max})} + 2)^2 \left(|\mathcal{X}||\mathcal{Y}| + \log{\left(\frac{1}{\delta}\right)}\right) + \gamma(C_{\max})^2}$$

$$+ \frac{1}{\rho}\left(\mathbb{E}_{x\sim\mathcal{D}^{\text{p}}, y\sim\pi^*(\cdot|x)}\left[\|\phi(x,y)\|_{(\Sigma_{\mathcal{D}^{\text{r}}} + \gamma I)^{-1}}\right] + \frac{1}{K}\sum_{k=1}^{K}\mathbb{E}_{x\sim\mathcal{D}^{\text{p}}, y\sim\pi_k(\cdot|x)}\left[\|\phi(x,y)\|_{(\Sigma_{\mathcal{D}^{\text{r}}} + \gamma I)^{-1}}\right]\right).$$

$$\sqrt{(\exp{(R_{\max})} + \exp{(-R_{\max})} + 2)^2 \left(|\mathcal{X}||\mathcal{Y}| + \log{\left(\frac{1}{\delta}\right)}\right) + \gamma(R_{\max})^2}.$$

$\square$

## D  PSEUDO-CODE AND DETAILED DESCRIPTION OF ALGORITHM O-PD-DPO

In this section, we present the pseudo-code and a more detailed description of algorithm O-PD-DPO.

Algorithm 2 illustrates the algorithm procedure of O-PD-DPO. Compared to algorithm PD-DPO, O-PD-DPO includes exploration bonuses $b_k^{\text{c}}(x,y)$ and $b_k^{\text{r}}(x,y)$ in the standard DPO and standard rearranged Lagrangian DPO training objectives (Lines 3 and 4). We define the exploration bonuses $b_k^{\diamond}(x,y)$ as

$$b_k^{\diamond}(x,y) := 4\|\phi(x,y)\|_{(\tilde{\Sigma}_{\mathcal{D}_k^{\diamond}} + \gamma^{\text{on}}I)^{-1}}\sqrt{\frac{(\exp{(z)} + \exp{(-z)} + 2)^2}{N^{\text{on}}}\left(|\mathcal{X}||\mathcal{Y}| + \log{\left(\frac{2}{\delta'}\right)}\right) + \gamma^{\text{on}}z^2},$$

where

$$\tilde{\Sigma}_{\mathcal{D}_k^{\diamond}} := \frac{1}{N^{\text{on}}}\sum_{(x,y,y')\in\mathcal{D}_1^{\diamond}}(\phi(x,y) - \phi(x,y'))(\phi(x,y) - \phi(x,y'))^{\top}$$

$$+ \frac{1}{N^{\text{on}}}\sum_{k=1}^{K}\sum_{i=1}^{N^{\text{on}}}(\phi(x_{k,i}, y_{k,i}) - \phi(x_{k,i}, y'_{k,i}))(\phi(x_{k,i}, y_{k,i}) - \phi(x_{k,i}, y'_{k,i}))^{\top}$$

with $z = R_{\max}$ when $\diamond = r$, and $z = C_{\max}$ when $\diamond = c$.

We take $b_k^{\text{r}}(x, y^{\text{rw}})$ in Eq. (29) as an example to explain the *intuition behind why including exploration bonuses $b_k^{\diamond}$ effectively encourages exploration*: When preference data do not cover $(x, y^{\text{rw}})$ well,

---

**Algorithm 2:** O-PD-DPO

---

**Input:** $\delta$, $\delta' := \frac{\delta}{4}$, $\beta$, $\pi_{\text{ref}}$, $\rho$, $\lambda_1$, $K$, $N^{\text{CE}}$, $M^{\text{CE}}$, $\gamma^{\text{on}}$, $N^{\text{on}} := 32K^2 \ln(\frac{8K|\mathcal{X}||\mathcal{Y}|}{\delta'})/(\gamma^{\text{on}})^2$, $\mathcal{D}^{\text{p}}$,
$\quad \mathcal{D}^{\text{r}} = \{(x_i^{\text{r}}, y_i^{\text{rw}}, y_i^{\text{rl}})\}_{i \in [N^{\text{r}}]}$, $\mathcal{D}^{\text{c}} = \{(x_i^{\text{c}}, y_i^{\text{cw}}, y_i^{\text{cl}})\}_{i \in [N^{\text{c}}]}$

1   $\mathcal{D}_1^{\text{r}} \leftarrow \mathcal{D}^{\text{r}}, \mathcal{D}_1^{\text{c}} \leftarrow \mathcal{D}^{\text{c}}$

2   **for** $k = 1, 2, \ldots, K$ **do**

3     Train a model using standard DPO with exploration bonuses:

$$\pi_{\hat{r}+b_k^{\text{r}}}^* \leftarrow \underset{\pi \in \tilde{\Pi}_k^{\text{r}}}{\text{argmin}} - \sum_{(x, y^{\text{rw}}, y^{\text{rl}}) \in \mathcal{D}^{\text{r}}} \log \sigma \Bigg( \beta \log \frac{\pi(y^{\text{rw}}|x)}{\pi_{\text{ref}}(y^{\text{rw}}|x)} - b_k^{\text{r}}(x, y^{\text{rw}})$$
$$- \left( \beta \log \frac{\pi(y^{\text{rl}}|x)}{\pi_{\text{ref}}(y^{\text{rl}}|x)} - b_k^{\text{r}}(x, y^{\text{rl}}) \right) \Bigg), \qquad (29)$$

    where $\tilde{\Pi}_k^{\text{r}}$ is defined in Eq. (31)

4     Train a model using a rearranged Lagrangian DPO objective with exploration bonuses:

$$\pi_k \leftarrow \underset{\pi \in \tilde{\Pi}_k^{\text{c}}}{\text{argmin}} - \sum_{(x, y^{\text{cw}}, y^{\text{cl}}) \in \mathcal{D}^{\text{c}}} \log \sigma \Bigg( \frac{1}{\lambda_k} \Bigg( \beta \log \frac{\pi_{\hat{r}_k+b_k^{\text{r}}}^*(y^{\text{cw}}|x)}{\pi_{\text{ref}}(y^{\text{cw}}|x)} - \beta \log \frac{\pi(y^{\text{cw}}|x)}{\pi_{\text{ref}}(y^{\text{cw}}|x)}$$
$$- b_k^{\text{c}}(x, y^{\text{cw}}) - \left( \beta \log \frac{\pi_{\hat{r}_k+b_k^{\text{r}}}^*(y^{\text{cl}}|x)}{\pi_{\text{ref}}(y^{\text{cl}}|x)} - \beta \log \frac{\pi(y^{\text{cl}}|x)}{\pi_{\text{ref}}(y^{\text{cl}}|x)} - b_k^{\text{c}}(x, y^{\text{cl}}) \right) \Bigg) \Bigg), \quad (30)$$

    where $\tilde{\Pi}_k^{\text{c}}$ is defined in Eq. (32)

5     Construct an estimate $\tilde{c}_k$ for $\mathbb{E}_{x \sim \mathcal{D}^{\text{p}}, y \sim \pi_k(\cdot|x)}[c^*(x, y)]$: For $i = 1, \ldots, N^{\text{CE}}$, first sample $x_i \sim \mathcal{D}^{\text{p}}$, $y_i \sim \pi_k(\cdot|x_i)$. Then, for each $(x_i, y_i)$, sample $\{Z_{i,j}\}_{j=1}^{M^{\text{CE}}} \overset{\text{i.i.d.}}{\sim} \text{Ber}(\sigma(c^*(x_i, y_i)))$. Set $\tilde{c}_k \leftarrow \frac{1}{N^{\text{CE}}} \sum_{i=1}^{N^{\text{CE}}} \sigma^{-1}(\frac{1}{M^{\text{CE}}} \sum_{j=1}^{M^{\text{CE}}} Z_{i,j})$, where $\sigma^{-1}(z) := \log(\frac{1}{1-z} - 1)$ is the inverse of the sigmoid function

6     $\lambda_{k+1} \leftarrow \text{Proj}_{[0,2\rho]}(\lambda_k + \eta\tilde{c}_k)$, where $\eta := \frac{\lambda_1}{C_{\max}\sqrt{K}}$

7     For $i = 1, \ldots, N^{\text{on}}$, sample $x_i \sim \mathcal{D}^{\text{p}}$, $y_i \sim \pi_k(\cdot|x_i)$, $y_i' \sim \pi^{\text{base}}(\cdot|x_i)$. Collect reward and cost preference feedback on $\{(x_i, y_i, y_i')\}_{i=1}^{N^{\text{on}}}$, and obtain preference data $\{(x_i, y_i^{\text{rw}}, y_i^{\text{rl}})\}_{i=1}^{N^{\text{on}}}$ and $\{(x_i, y_i^{\text{cw}}, y_i^{\text{cl}})\}_{i=1}^{N^{\text{on}}}$

8     $\mathcal{D}_{k+1}^{\text{r}} \leftarrow \mathcal{D}_k^{\text{r}} \cup \{(x_i, y_i^{\text{rw}}, y_i^{\text{rl}})\}_{i=1}^{N^{\text{on}}}, \mathcal{D}_{k+1}^{\text{c}} \leftarrow \mathcal{D}_k^{\text{c}} \cup \{(x_i, y_i^{\text{cw}}, y_i^{\text{cl}})\}_{i=1}^{N^{\text{on}}}$

9   **return** $\pi_K^{\text{out}} := \text{unif}(\pi_1, \ldots, \pi_K)$

---

$b_k^{\text{r}}(x, y^{\text{rw}})$ will be large. Then, subtracting a large value from $\beta \log \frac{\pi(y^{\text{rw}}|x)}{\pi_{\text{ref}}(y^{\text{rw}}|x)}$ encourages $\pi$ to put a higher probability on $y^{\text{rw}}$ to maintain the original value of $\beta \log \frac{\pi(y^{\text{rw}}|x)}{\pi_{\text{ref}}(y^{\text{rw}}|x)}$ which achieves the optimal value of the MLE training objective. By incorporating exploration bonuses in the training objective, the trained model $\pi_k$ has incentive to explore uncovered prompt-response space.

In addition, the constrained policy search ranges in Lines 3 and 4 also incorporate exploration bonuses, which are defined as

$$\tilde{\Pi}_k^{\text{r}} := \left\{ \pi(y|x) = \frac{\pi_{\text{ref}}(y|x) \cdot \exp\left(\frac{1}{\beta}(r(x, y) + b_k^{\text{r}}(x, y))\right)}{\sum_{y' \in \mathcal{Y}} \pi_{\text{ref}}(y'|x) \cdot \exp\left(\frac{1}{\beta}(r(x, y') + b_k^{\text{r}}(x, y'))\right)} : r \in \mathcal{R} \right\}, \qquad (31)$$

and

$$\tilde{\Pi}_k^{\text{c}} := \left\{ \pi(y|x) = \frac{\pi_{\text{ref}}(y|x) \cdot \exp\left(\frac{1}{\beta}\left(\beta \log \frac{\pi_{\hat{r}_k+b_k^{\text{r}}}^*(y|x)}{\pi_{\text{ref}}(y|x)} - \lambda_k(c(x, y) - b_k^{\text{c}}(x, y))\right)\right)}{\sum_{y' \in \mathcal{Y}} \pi_{\text{ref}}(y'|x) \cdot \exp\left(\frac{1}{\beta}\left(\beta \log \frac{\pi_{\hat{r}_k+b_k^{\text{r}}}^*(y'|x)}{\pi_{\text{ref}}(y'|x)} - \lambda_k(c(x, y') - b_k^{\text{c}}(x, y'))\right)\right)} : \right.$$

$$c \in \mathcal{C} \Bigg\}$$

$$= \Bigg\{ \pi(y|x) = \frac{\pi_{\mathrm{ref}}(y|x) \cdot \exp\left(\frac{1}{\beta}\left(\hat{r}_k(x, y) + b_k^{\mathrm{r}}(x, y) - \lambda_k\left(c(x, y) - b_k^{\mathrm{c}}(x, y)\right)\right)\right)}{\sum_{y' \in \mathcal{Y}} \pi_{\mathrm{ref}}(y'|x) \cdot \exp\left(\frac{1}{\beta}\left(\hat{r}_k(x, y') + b_k^{\mathrm{r}}(x, y') - \lambda_k\left(c(x, y') - b_k^{\mathrm{c}}(x, y')\right)\right)\right)} :$$

$$c \in \mathcal{C} \Bigg\}. \tag{32}$$

At the end of each iteration, O-PD-DPO collects reward and cost preference feedback using $\pi_k$ and a baseline policy $\pi^{\mathrm{base}}$ (Line 7). The baseline policy $\pi^{\mathrm{base}}$ is a fixed policy used in online preference data collection for ease of comparison. We make a technical assumption on $\pi^{\mathrm{base}}$:

**Assumption 2** (Baseline Policy). *The baseline policy $\pi^{\mathrm{base}}$ satisfies that for any policy $\pi$,*

$$\mathbb{E}_{x \sim \mathcal{D}^{\mathrm{p}}, y \sim \pi, y' \sim \pi^{\mathrm{base}}}\left[\left(\phi(x, y) - \phi(x, y')\right)\left(\phi(x, y) - \phi(x, y')\right)^{\top}\right] \tag{33}$$

$$\succeq C^{\mathrm{base}}\, \mathbb{E}_{x \sim \mathcal{D}^{\mathrm{p}}, y' \sim \pi^{\mathrm{base}}}\left[\phi(x, y')\phi(x, y')^{\top}\right]. \tag{34}$$

This assumption is used to guarantee that the difference of feature vectors between any policy $\pi$ and $\pi^{\mathrm{base}}$ can be connected to the feature vectors of $\pi^{\mathrm{base}}$ itself, which is useful in analysis when bounding the error due to inferring reward and cost information from preference data.

After collecting online preference data, O-PD-DPO adds these data to $\mathcal{D}_k^{\mathrm{r}}$ and $\mathcal{D}_k^{\mathrm{c}}$, which will be used in model training in the next iteration (Line 8). As the algorithm proceeds, the preference data $\mathcal{D}_k^{\mathrm{r}}$ and $\mathcal{D}_k^{\mathrm{c}}$ will cover more and more prompt-response space.

# E    PROOFS FOR ALGORITHM O-PD-DPO

In this section, we provide the proofs for algorithm O-PD-DPO in the online data setting, including the proofs for the connection to the RLHF-based procedure, suboptimality, and constraint violation.

## E.1    CONNECTION BETWEEN OUR DPO-BASED PROCEDURE AND THE RLHF-BASED PROCEDURE WITH EXPLORATION BONUSES

First, we give a result which establishes a connection between standard DPO and standard RLHF with constrained policy search ranges and exploration bonuses.

Define the following problem which first learns a reward model and then finds the optimal policy to maximize the learned reward with exploration bonuses:

$$\hat{r}_k \leftarrow \min_{r \in \mathcal{R}} - \sum_{(x, y^{\mathrm{rw}}, y^{\mathrm{rl}}) \in \mathcal{D}_k^{\mathrm{r}}} \log \sigma\left(r(x, y^{\mathrm{rw}}) - r(x, y^{\mathrm{rl}})\right) \tag{35}$$

$$\max_{\pi} \mathbb{E}_{x \sim \mathcal{D}^{\mathrm{p}}}\left[\mathbb{E}_{y \sim \pi(\cdot|x)}\left[\hat{r}_k(x, y) + b_k^{\mathrm{r}}(x, y)\right] - \beta \cdot \mathrm{KL}(\pi(\cdot|x)\|\pi_{\mathrm{ref}}(\cdot|x))\right] \tag{36}$$

**Theorem 5** (Connection between Standard DPO and Standard RLHF with Constrained Policy Ranges and Exploration Bonuses). *Problems Eqs. (29) and (36) have the same set of optimal solutions.*

*Proof.* **Step (i).** First, we prove that if $\pi$ is an optimal solution to Eq. (36), then $\pi$ is also an optimal solution to Eq. (29).

If $\hat{r}_k \in \mathcal{R}$ is an optimal solution to Eq. (35), then

$$\pi_{\hat{r}_k + b_k^{\mathrm{r}}}^*(y|x) = \frac{\pi_{\mathrm{ref}}(y|x) \cdot \exp\left(\frac{1}{\beta}\left(\hat{r}_k(x, y) + b_k^{\mathrm{r}}(x, y)\right)\right)}{\sum_{y' \in \mathcal{Y}} \pi_{\mathrm{ref}}(y'|x) \cdot \exp\left(\frac{1}{\beta}\left(\hat{r}_k(x, y') + b_k^{\mathrm{r}}(x, y')\right)\right)}$$

is an optimal solution to Eq. (36). We have that $\pi^*_{\hat{r}_k + b^r_k}$ is also an optimal solution to Eq. (29). Otherwise, there exists another $\pi' \in \tilde{\Pi}^r_k$ which achieves a smaller objective value in Eq. (29). Then, there must exist a $r' \in \mathcal{R}$ which satisfies that

$$\pi'(y|x) = \frac{\pi_{\text{ref}}(y|x) \cdot \exp\left(\frac{1}{\beta}\left(r'(x,y) + b^r_k(x,y)\right)\right)}{\sum_{y' \in \mathcal{Y}} \pi_{\text{ref}}(y'|x) \cdot \exp\left(\frac{1}{\beta}\left(r'(x,y') + b^r_k(x,y')\right)\right)},$$

i.e.,

$$r'(x,y) = \beta \log \frac{\pi'(y|x)}{\pi_{\text{ref}}(y|x)} + \beta \log Z_{r'+b^r_k}(x) - b^r_k(x,y),$$

and the objective value in Eq. (35) achieved by $r'$,

$$-\sum_{(x,y^{\text{rw}},y^{\text{rl}}) \in \tilde{\mathcal{D}}^r_k} \log \sigma\left(\beta \log \frac{\pi'(y^{\text{rw}}|x)}{\pi_{\text{ref}}(y^{\text{rw}}|x)} + \beta \log Z_{r'+b^r_k}(x) - b^r_k(x,y^{\text{rw}})\right.$$

$$\left. - \left(\beta \log \frac{\pi'(y^{\text{rl}}|x)}{\pi_{\text{ref}}(y^{\text{rl}}|x)} + \beta \log Z_{r'+b^r_k}(x) - b^r_k(x,y^{\text{rl}})\right)\right),$$

is smaller than that achieved by $\hat{r}_k$ (since $\pi'$ achieves a smaller DPO objective value), which contradicts the supposition that $\hat{r}_k$ is the optimal solution to Eq. (35).

**Step (ii).** Next, we prove that if $\pi$ is an optimal solution to Eq. (29), then $\pi$ is also an optimal solution to Eq. (36).

If $\tilde{\pi} \in \tilde{\Pi}^r_k$ is an optimal solution to Eq. (29), then there exists a $\tilde{r} \in \mathcal{R}$ which satisfies

$$\tilde{\pi}(y|x) = \frac{\pi_{\text{ref}}(y|x) \cdot \exp\left(\frac{1}{\beta}\left(\tilde{r}(x,y) + b^r_k(x,y)\right)\right)}{\sum_{y' \in \mathcal{Y}} \pi_{\text{ref}}(y'|x) \cdot \exp\left(\frac{1}{\beta}\left(\tilde{r}(x,y') + b^r_k(x,y')\right)\right)},$$

i.e.,

$$\tilde{r}(x,y) = \beta \log \frac{\tilde{\pi}(y|x)}{\pi_{\text{ref}}(y|x)} + \beta \log Z_{\tilde{r}}(x) - b^r_k(x,y). \tag{37}$$

We have that $\tilde{r}$ achieves the optimal value in Eq. (35),

$$-\sum_{(x,y^{\text{rw}},y^{\text{rl}}) \in \tilde{\mathcal{D}}^r_k} \log \sigma\left(\beta \log \frac{\tilde{\pi}(y^{\text{rw}}|x)}{\pi_{\text{ref}}(y^{\text{rw}}|x)} + \beta \log Z_{\tilde{r}}(x) - b^r_k(x,y^{\text{rw}})\right.$$

$$\left. - \left(\beta \log \frac{\tilde{\pi}(y^{\text{rl}}|x)}{\pi_{\text{ref}}(y^{\text{rl}}|x)} + \beta \log Z_{\tilde{r}}(x) - b^r_k(x,y^{\text{rl}})\right)\right). \tag{38}$$

Otherwise, there exists another $r' \in \mathcal{R}$ and then there exists a $\pi' = \pi^*_{\hat{r}} \in \tilde{\Pi}^r_k$ which gives a smaller objective value than $\tilde{\pi}$ in Eq. (38). Thus, $\tilde{r}$ achieves the optimal value in Eq. (35). Then, the optimal solution to Eq. (36) under cost model $\tilde{r}$ is

$$\pi(y|x) \propto \pi_{\text{ref}}(y|x) \cdot \exp\left(\frac{1}{\beta}\left(\tilde{r}(x,y) + b^r_k(x,y)\right)\right)$$

$$\overset{(a)}{\propto} \pi_{\text{ref}}(y|x) \cdot \exp\left(\frac{1}{\beta}\left(\beta \log \frac{\tilde{\pi}(y|x)}{\pi_{\text{ref}}(y|x)} + \beta \log Z_{\tilde{r}}(x)\right)\right)$$

$$\propto \pi_{\text{ref}}(y|x) \cdot \exp\left(\log \frac{\tilde{\pi}(y|x)}{\pi_{\text{ref}}(y|x)}\right)$$

$$= \tilde{\pi}(y|x),$$

where (a) uses Eq. (37).

Therefore, $\tilde{\pi}$ is also an optimal solution to Eq. (36). $\qquad\square$

Now we present a result which relates our rearranged Lagrangian DPO objective to the safe RLHF objective with constrained policy search ranges and exploration bonuses.

For any $k \geq 1$, define the following problem that first learns a cost model and then finds the optimal policy for the Lagrangian function under $\hat{r}_k + b_k^{\mathrm{r}}$ and $\lambda_k$:

$$
\hat{c}_k \leftarrow \min_{c \in \mathcal{C}} -\frac{1}{N^{\mathrm{c}}} \sum_{i=1}^{N^{\mathrm{c}}} \log \sigma \left( c(x_i, y_i^{\mathrm{cw}}) - c(x_i, y_i^{\mathrm{cl}}) \right) \tag{39}
$$

$$
\max_{\pi} \, \mathbb{E}_{x \sim \mathcal{D}^{\mathrm{p}}} \left[ \mathbb{E}_{y \sim \pi(\cdot|x)} \left[ \hat{r}_k(x, y) + b_k^{\mathrm{r}}(x, y) - \lambda_k \left( \hat{c}_k(x, y) - b_k^{\mathrm{c}}(x, y) \right) \right] - \beta \cdot \mathrm{KL}(\pi(\cdot|x) \| \pi_{\mathrm{ref}}(\cdot|x)) \right] \tag{40}
$$

**Theorem 6** (Connection between Our Rearranged Lagrangian DPO and Safe RLHF with Constrained Policy Ranges and Exploration Bonuses). *For any $k \geq 0$, Problems Eqs. (30) and (40) have the same set of optimal solutions.*

*Proof.* First, note that for any $\hat{c}_k$, the optimal solution to Eq. (40) is

$$
\pi^*_{\hat{r}_k + b_k^{\mathrm{r}} - \lambda_k(\hat{c}_k - b_k^{\mathrm{c}})}(y|x) = \frac{\pi_{\mathrm{ref}}(y|x) \exp\left( \frac{1}{\beta}(\hat{r}_k(x, y) + b_k^{\mathrm{r}}(x, y) - \lambda_k\left(\hat{c}(x, y) - b_k^{\mathrm{c}}(x, y)\right)) \right)}{\underbrace{\sum_{y' \in \mathcal{Y}} \pi_{\mathrm{ref}}(y'|x) \exp\left( \frac{1}{\beta}(\hat{r}_k(x, y') + b_k^{\mathrm{r}}(x, y') - \lambda_k\left(\hat{c}(x, y') - b_k^{\mathrm{c}}(x, y')\right)) \right)}_{:= Z_{\hat{r}_k + b_k^{\mathrm{r}} - \lambda_k(\hat{c}_k - b_k^{\mathrm{c}})}(x)}},
$$

$$
\forall x \in \mathcal{X}. \tag{41}
$$

Then, we have

$$
\hat{c}_k(x, y) = \frac{1}{\lambda_k} \left( \hat{r}_k(x, y) + b_k^{\mathrm{r}}(x, y) - \beta \log \frac{\pi^*_{\hat{r}_k + b_k^{\mathrm{r}} - \lambda_k(\hat{c}_k - b_k^{\mathrm{c}})}(y|x)}{\pi_{\mathrm{ref}}(y|x)} \right.
$$

$$
\left. - \beta \log Z_{\hat{r}_k + b_k^{\mathrm{r}} - \lambda_k(\hat{c}_k - b_k^{\mathrm{c}})}(x) \right) + b_k^{\mathrm{c}}(x, y)
$$

$$
\overset{(a)}{=} \frac{1}{\lambda_k} \left( \beta \log \frac{\pi^*_{\hat{r}_k + b_k^{\mathrm{r}}}(y|x)}{\pi_{\mathrm{ref}}(y|x)} + \beta \log Z_{\hat{r}_k + b_k^{\mathrm{r}}}(x) - \beta \log \frac{\pi^*_{\hat{r}_k + b_k^{\mathrm{r}} - \lambda_k(\hat{c}_k - b_k^{\mathrm{c}})}(y|x)}{\pi_{\mathrm{ref}}(y|x)} \right.
$$

$$
\left. - \beta \log Z_{\hat{r}_k + b_k^{\mathrm{r}} - \lambda_k(\hat{c}_k - b_k^{\mathrm{c}})}(x) \right) + b_k^{\mathrm{c}}(x, y),
$$

where equality (a) uses a similar derivation as Eq. (5).

Now we prove this theorem.

**Step (i).** First, we prove that if $\pi$ is an optimal solution to Eq. (40), then $\pi$ is also an optimal solution to Eq. (30).

If $\hat{c}_k \in \mathcal{C}$ is an optimal solution to Eq. (39), then $\pi^*_{\hat{r}_k + b_k^{\mathrm{r}} - \lambda_k(\hat{c}_k - b_k^{\mathrm{c}})} \in \tilde{\Pi}_k^{\mathrm{c}}$ (as shown in Eq. (41)) is an optimal solution to Eq. (40). We have that $\pi^*_{\hat{r}_k + b_k^{\mathrm{r}} - \lambda_k(\hat{c}_k - b_k^{\mathrm{c}})}$ is also an optimal solution to Eq. (30). Otherwise, there exists another $\pi' \in \tilde{\Pi}_k^{\mathrm{c}}$ which achieves a smaller objective value in Eq. (30). Then, there must exist a $c' \in \mathcal{C}$ which satisfies that

$$
\pi'(y|x) = \frac{\pi_{\mathrm{ref}}(y|x) \cdot \exp\left( \frac{1}{\beta} \left( \beta \log \frac{\pi^*_{\hat{r}_k + b_k^{\mathrm{r}}}(y|x)}{\pi_{\mathrm{ref}}(y|x)} - \lambda_k\left(c'(x, y) - b_k^{\mathrm{c}}(x, y)\right) \right) \right)}{\underbrace{\sum_{y' \in \mathcal{Y}} \pi_{\mathrm{ref}}(y'|x) \cdot \exp\left( \frac{1}{\beta} \left( \beta \log \frac{\pi^*_{\hat{r}_k + b_k^{\mathrm{r}}}(y|x)}{\pi_{\mathrm{ref}}(y|x)} - \lambda_k\left(c'(x, y') - b_k^{\mathrm{c}}(x, y')\right) \right) \right)}_{:= Z_{\beta \log \frac{\pi^*_{\hat{r}_k + b_k^{\mathrm{r}}}}{\pi_{\mathrm{ref}}} - \lambda_k(c' - b_k^{\mathrm{c}})}(x)}},
$$

i.e.,

$$c'(x,y) = \frac{1}{\lambda_k}\left(\beta\log\frac{\pi^*_{\hat{r}_k+b^r_k}(y|x)}{\pi_{\text{ref}}(y|x)} - \beta\log\frac{\pi'(y|x)}{\pi_{\text{ref}}(y|x)} - \beta\log Z_{\beta\log\frac{\pi^*_{\hat{r}_k+b^r_k}}{\pi_{\text{ref}}}-\lambda_k(c'-b^c_k)}(x)\right)$$
$$+ b^c_k(x,y),$$

and the objective value in Eq. (39) achieved by $c'$,

$$-\sum_{(x,y^{\text{cw}},y^{\text{cl}})\in\mathcal{D}^c}\log\sigma\left(\frac{1}{\lambda_k}\left(\beta\log\frac{\pi^*_{\hat{r}_k+b^r_k}(y^{\text{cw}}|x)}{\pi_{\text{ref}}(y^{\text{cw}}|x)} - \beta\log\frac{\pi'(y^{\text{cw}}|x)}{\pi_{\text{ref}}(y^{\text{cw}}|x)}\right.\right.$$

$$\left.-\beta\log Z_{\beta\log\frac{\pi^*_{\hat{r}_k+b^r_k}}{\pi_{\text{ref}}}-\lambda_k(c'-b^c_k)}(x)\right) + b^c_k(x,y^{\text{cw}}) - \frac{1}{\lambda_k}\left(\beta\log\frac{\pi^*_{\hat{r}_k+b^r_k}(y^{\text{cl}}|x)}{\pi_{\text{ref}}(y^{\text{cl}}|x)} - \beta\log\frac{\pi'(y^{\text{cl}}|x)}{\pi_{\text{ref}}(y^{\text{cl}}|x)}\right.$$

$$\left.\left.-\beta\log Z_{\beta\log\frac{\pi^*_{\hat{r}_k+b^r_k}}{\pi_{\text{ref}}}-\lambda_k(c'-b^c_k)}(x)\right) - b^c_k(x,y^{\text{cl}})\right),$$

is smaller than that achieved by $\hat{c}_k$, which contradicts the supposition that $\hat{c}_k$ is the optimal solution to Eq. (39).

**Step (ii).** Next, we prove that if $\pi$ is an optimal solution to Eq. (30), then $\pi$ is also an optimal solution to Eq. (40).

If $\pi_k \in \tilde{\Pi}^c_k$ is an optimal solution to Eq. (30), then there exists a $c_k \in \mathcal{C}$ which satisfies

$$\pi_k(y|x) = \frac{\pi_{\text{ref}}(y|x)\cdot\exp\left(\frac{1}{\beta}\left(\beta\log\frac{\pi^*_{\hat{r}_k+b^r_k}(y|x)}{\pi_{\text{ref}}(y|x)} - \lambda_k\left(c_k(x,y) - b^c_k(x,y)\right)\right)\right)}{\sum_{y'\in\mathcal{Y}}\pi_{\text{ref}}(y'|x)\cdot\exp\left(\frac{1}{\beta}\left(\beta\log\frac{\pi^*_{\hat{r}_k+b^r_k}(y'|x)}{\pi_{\text{ref}}(y'|x)} - \lambda_k\left(c_k(x,y') - b^c_k(x,y')\right)\right)\right)},$$

i.e.,

$$c_k(x,y) = \frac{1}{\lambda_k}\left(\beta\log\frac{\pi^*_{\hat{r}_k+b^r_k}(y|x)}{\pi_{\text{ref}}(y|x)} - \beta\log\frac{\pi_k(y|x)}{\pi_{\text{ref}}(y|x)} - \beta\log Z_{\beta\log\frac{\pi^*_{\hat{r}_k+b^r_k}}{\pi_{\text{ref}}}-\lambda_k(c_k-b^c_k)}(x)\right)$$
$$+ b^c_k(x,y).$$

We have that $c_k$ achieves the optimal value in Eq. (39),

$$-\sum_{(x,y^{\text{cw}},y^{\text{cl}})\in\mathcal{D}^c}\log\sigma\left(\frac{1}{\lambda_k}\left(\beta\log\frac{\pi^*_{\hat{r}_k+b^r_k}(y^{\text{cw}}|x)}{\pi_{\text{ref}}(y^{\text{cw}}|x)} - \beta\log\frac{\pi_k(y^{\text{cw}}|x)}{\pi_{\text{ref}}(y^{\text{cw}}|x)}\right.\right.$$

$$\left.-\beta\log Z_{\beta\log\frac{\pi^*_{\hat{r}_k+b^r_k}}{\pi_{\text{ref}}}-\lambda_k(c_k-b^c_k)}(x)\right) + b^c_k(x,y^{\text{cw}}) - \frac{1}{\lambda_k}\left(\beta\log\frac{\pi^*_{\hat{r}_k+b^r_k}(y^{\text{cl}}|x)}{\pi_{\text{ref}}(y^{\text{cl}}|x)} - \beta\log\frac{\pi_k(y^{\text{cl}}|x)}{\pi_{\text{ref}}(y^{\text{cl}}|x)}\right.$$

$$\left.\left.-\beta\log Z_{\beta\log\frac{\pi^*_{\hat{r}_k+b^r_k}}{\pi_{\text{ref}}}-\lambda_k(c_k-b^c_k)}(x)\right) - b^c_k(x,y^{\text{cl}})\right). \tag{42}$$

Otherwise, there exists another $c' \in \mathcal{C}$ and then there exists a $\pi' = \pi^*_{\hat{r}_k+b^r_k-\lambda_k(c'-b^c_k)} \in \tilde{\Pi}^c_k$ which gives a smaller objective value than $\tilde{\pi}_k$ in Eq. (42). Thus, $c_k$ achieves the optimal value in Eq. (39). Then, the optimal solution to Eq. (39) under cost model $c_k$ is

$$\pi(y|x) \propto \pi_{\text{ref}}(y|x)\cdot\exp\left(\frac{1}{\beta}\left(\hat{r}_k(x,y) + b^r_k(x,y) - \beta\log\frac{\pi^*_{\hat{r}_k+b^r_k}(y|x)}{\pi_{\text{ref}}(y|x)} + \beta\log\frac{\pi_k(y|x)}{\pi_{\text{ref}}(y|x)}\right.\right.$$

$$\left.\left.+\beta\log Z_{\beta\log\frac{\pi^*_{\hat{r}_k+b^r_k}}{\pi_{\text{ref}}}-\lambda_k(c_k-b^c_k)}(x)\right)\right)$$

$$\overset{(a)}{\propto} \pi_{\text{ref}}(y|x)\cdot\exp\left(\frac{1}{\beta}\left(\beta\log Z_{\hat{r}_k+b^r_k}(x) + \beta\log\frac{\pi_k(y|x)}{\pi_{\text{ref}}(y|x)}\right.\right.$$

$$+ \beta \log Z_{\beta \log \frac{\pi^*_{\hat{r}_k + b^r_k}}{\pi_{\text{ref}}} - \lambda_k(c_k - b^c_k)}(x) \Bigg) \Bigg)$$

$$\propto \pi_{\text{ref}}(y|x) \cdot \exp\left(\frac{1}{\beta}\left(\beta \log \frac{\pi_k(y|x)}{\pi_{\text{ref}}(y|x)}\right)\right)$$

$$= \pi_k(y|x),$$

where (a) uses a similar derivation as Eq. (5).

Therefore, $\pi_k$ is also an optimal solution to Eq. (23). $\qquad\square$

### E.2 SUBOPTIMALITY AND CONSTRAINT VIOLATION

In the following, we present the proof of the suboptimality and constraint violation guarantees for algorithm O-PD-DPO (Theorem 2).

Define event

$$\mathcal{G}^{\text{on}} := \Bigg\{ |\hat{r}_k(x,y) - r^*(x,y)| \leq 4\,\|\phi(x,y)\|_{(\tilde{\Sigma}_{\mathcal{D}^r_k} + \gamma^{\text{on}}I)^{-1}} \cdot$$

$$\sqrt{\frac{(\exp(R_{\max}) + \exp(-R_{\max}) + 2)^2}{N^{\text{on}}}\left(|\mathcal{X}||\mathcal{Y}| + \log\left(\frac{2K}{\delta'}\right)\right) + \gamma^{\text{on}}(R_{\max})^2} := b^r_k(x,y),$$

$$|\hat{c}_k(x,y) - c^*(x,y)| \leq 4\,\|\phi(x,y)\|_{(\tilde{\Sigma}_{\mathcal{D}^c_k} + \gamma^{\text{on}}I)^{-1}} \cdot$$

$$\sqrt{\frac{(\exp(C_{\max}) + \exp(-C_{\max}) + 2)^2}{N^{\text{on}}}\left(|\mathcal{X}||\mathcal{Y}| + \log\left(\frac{2K}{\delta'}\right)\right) + \gamma^{\text{on}}(C_{\max})^2} := b^c_k(x,y),$$

$$\forall (x,y) \in \mathcal{X} \times \mathcal{Y} \Bigg\}.$$

**Lemma 5** (MLE Guarantee with Online Data). *It holds that*

$$\Pr[\mathcal{G}^{\text{on}}] \geq 1 - 2\delta'.$$

*Proof.* According to Lemma 3.1 in (Zhu et al., 2023), we have that with probability at least $1 - \delta'$,

$$|\hat{r}_k(x,y) - r^*(x,y)|$$
$$\leq 4\,\|\phi(x,y)\|_{(\Sigma_{\mathcal{D}^r_k} + N^{\text{on}}\gamma^{\text{on}}I)^{-1}} \cdot$$

$$\sqrt{(\exp(R_{\max}) + \exp(-R_{\max}) + 2)^2\left(|\mathcal{X}||\mathcal{Y}| + \log\left(\frac{2}{\delta'}\right)\right) + N^{\text{on}}\gamma^{\text{on}}(R_{\max})^2}$$

$$= \frac{4}{\sqrt{N^{\text{on}}}}\,\|\phi(x,y)\|_{(\frac{1}{N^{\text{on}}}\Sigma_{\mathcal{D}^r_k} + \gamma^{\text{on}}I)^{-1}} \cdot$$

$$\sqrt{(\exp(R_{\max}) + \exp(-R_{\max}) + 2)^2\left(|\mathcal{X}||\mathcal{Y}| + \log\left(\frac{2}{\delta'}\right)\right) + N^{\text{on}}\gamma^{\text{on}}(R_{\max})^2}$$

$$= 4\,\|\phi(x,y)\|_{(\tilde{\Sigma}_{\mathcal{D}^r_k} + \gamma^{\text{on}}I)^{-1}} \cdot$$

$$\sqrt{\frac{(\exp(R_{\max}) + \exp(-R_{\max}) + 2)^2}{N^{\text{on}}}\left(|\mathcal{X}||\mathcal{Y}| + \log\left(\frac{2}{\delta'}\right)\right) + \gamma^{\text{on}}(R_{\max})^2}.$$

Taking a union bound over $k = 1, \dots, K$, we can obtain the first statement.

Using a similar argument, we can obtain the second statement. $\qquad\square$

**Lemma 6.** *For any $k \geq 1$, we have*

$$f(\pi^*; \hat{r}_k + b^r_k) - f(\pi_k; \hat{r}_k + b^r_k) \leq -\lambda_k \cdot \mathbb{E}_{x \sim \mathcal{D}^p, y \sim \pi_k(\cdot|x)}[\hat{c}_k(x,y) - b^c_k(x,y)].$$

*Proof.* It holds that

$$f(\pi^*; \hat{r}_k + b_k^{\mathrm{r}})$$

$$\overset{(a)}{\le} f(\pi^*; \hat{r}_k + b_k^{\mathrm{r}}) - \lambda_k \cdot \mathbb{E}_{x \sim \mathcal{D}^{\mathrm{p}}, y \sim \pi^*(\cdot|x)}[c^*(x, y)]$$

$$= \mathbb{E}_{x \sim \mathcal{D}^{\mathrm{p}}}\left[\mathbb{E}_{y \sim \pi^*(\cdot|x)}\left[\hat{r}_k(x, y) + b_k^{\mathrm{r}}(x, y) - \lambda_k \cdot c^*(x, y)\right] - \beta \cdot \mathrm{KL}(\pi^*(\cdot|x)\|\pi_{\mathrm{ref}}(\cdot|x))\right]$$

$$= \mathbb{E}_{x \sim \mathcal{D}^{\mathrm{p}}}\Big[\mathbb{E}_{y \sim \pi^*(\cdot|x)}\left[\hat{r}_k(x, y) + b_k^{\mathrm{r}}(x, y) - \lambda_k\left(\hat{c}_k(x, y) - b_k^{\mathrm{c}}(x, y)\right)\right]$$

$$- \beta \cdot \mathrm{KL}(\pi^*(\cdot|x)\|\pi_{\mathrm{ref}}(\cdot|x))\Big] + \lambda_k \cdot \mathbb{E}_{x \sim \mathcal{D}^{\mathrm{p}}, y \sim \pi^*(\cdot|x)}[\hat{c}_k(x, y) - b_k^{\mathrm{c}}(x, y) - c^*(x, y)]$$

$$\overset{(b)}{\le} \mathbb{E}_{x \sim \mathcal{D}^{\mathrm{p}}}\Big[\mathbb{E}_{y \sim \pi_k(\cdot|x)}\left[\hat{r}_k(x, y) + b_k^{\mathrm{r}}(x, y) - \lambda_k\left(\hat{c}_k(x, y) - b_k^{\mathrm{c}}(x, y)\right)\right]$$

$$- \beta \cdot \mathrm{KL}(\pi_k(\cdot|x)\|\pi_{\mathrm{ref}}(\cdot|x))\Big]$$

$$= f(\pi_k; \hat{r}_k + b_k^{\mathrm{r}}) - \lambda_k \cdot \mathbb{E}_{x \sim \mathcal{D}^{\mathrm{p}}, y \sim \pi_k(\cdot|x)}[\hat{c}_k(x, y) - b_k^{\mathrm{c}}(x, y)],$$

where inequality (a) uses the fact that $\lambda_k \ge 0$ and $\pi^*$ is feasible, and inequality (b) comes from Theorem 6. $\square$

Let

$$\bar{\Sigma}_{\mathcal{D}_k^{\mathrm{r}}} := \Sigma_{\mathcal{D}_1^{\mathrm{r}}} + \sum_{k=1}^{K} \mathbb{E}_{x \sim \mathcal{D}^{\mathrm{p}}, y \sim \pi_k(\cdot|x), y' \sim \pi^{\mathrm{base}}(\cdot|x)}\left[\left(\phi(x, y) - \phi(x, y')\right)\left(\phi(x, y) - \phi(x, y')\right)^\top\right],$$

$$\bar{\Sigma}_{\mathcal{D}_k^{\mathrm{c}}} := \Sigma_{\mathcal{D}_1^{\mathrm{c}}} + \sum_{k=1}^{K} \mathbb{E}_{x \sim \mathcal{D}^{\mathrm{p}}, y \sim \pi_k(\cdot|x), y' \sim \pi^{\mathrm{base}}(\cdot|x)}\left[\left(\phi(x, y) - \phi(x, y')\right)\left(\phi(x, y) - \phi(x, y')\right)^\top\right].$$

**Lemma 7.** *It holds that*

$$\sum_{k=1}^{K} \mathbb{E}_{x \sim \mathcal{D}^{\mathrm{p}}, y \sim \pi_k(\cdot|x)}[b_k^{\mathrm{r}}(x, y)]$$

$$\le 4\sqrt{\frac{(\exp(R_{\max}) + \exp(-R_{\max}) + 2)^2}{N^{\mathrm{on}}}\left(|\mathcal{X}||\mathcal{Y}| + \log\left(\frac{2K}{\delta'}\right)\right) + \gamma^{\mathrm{on}}(R_{\max})^2}\cdot$$

$$2\sqrt{2|\mathcal{X}||\mathcal{Y}|K\left(\log\left(\frac{\gamma^{\mathrm{on}} + 4|\mathcal{D}_1^{\mathrm{r}}| + 4K}{|\mathcal{X}||\mathcal{Y}|\gamma^{\mathrm{on}}}\right) + \frac{1}{C^{\mathrm{base}}}\log\left(\frac{\gamma^{\mathrm{on}} + |\mathcal{D}_1^{\mathrm{r}}| + C^{\mathrm{base}}K}{|\mathcal{X}||\mathcal{Y}|\gamma^{\mathrm{on}}}\right)\right)},$$

*and*

$$\sum_{k=1}^{K} \mathbb{E}_{x \sim \mathcal{D}^{\mathrm{p}}, y \sim \pi_k(\cdot|x)}[b_k^{\mathrm{c}}(x, y)]$$

$$\le 4\sqrt{\frac{(\exp(C_{\max}) + \exp(-C_{\max}) + 2)^2}{N^{\mathrm{on}}}\left(|\mathcal{X}||\mathcal{Y}| + \log\left(\frac{2K}{\delta'}\right)\right) + \gamma^{\mathrm{on}}(C_{\max})^2}\cdot$$

$$2\sqrt{2|\mathcal{X}||\mathcal{Y}|K\left(\log\left(\frac{\gamma^{\mathrm{on}} + 4|\mathcal{D}_1^{\mathrm{c}}| + 4K}{|\mathcal{X}||\mathcal{Y}|\gamma^{\mathrm{on}}}\right) + \frac{1}{C^{\mathrm{base}}}\log\left(\frac{\gamma^{\mathrm{on}} + |\mathcal{D}_1^{\mathrm{c}}| + C^{\mathrm{base}}K}{|\mathcal{X}||\mathcal{Y}|\gamma^{\mathrm{on}}}\right)\right)}.$$

*Proof.* First, we have

$$\bar{\Sigma}_{\mathcal{D}_k^{\mathrm{r}}} + \gamma^{\mathrm{on}}I = \Sigma_{\mathcal{D}_1^{\mathrm{r}}} + \gamma^{\mathrm{on}}I$$

$$+ \sum_{k=1}^{K} \mathbb{E}_{x \sim \mathcal{D}^{\mathrm{p}}, y \sim \pi_k(\cdot|x), y' \sim \pi^{\mathrm{base}}(\cdot|x)}\left[\left(\phi(x, y) - \phi(x, y')\right)\left(\phi(x, y) - \phi(x, y')\right)^\top\right]$$

$$\succeq \Sigma_{\mathcal{D}_1^{\mathrm{r}}} + \gamma^{\mathrm{on}}I + C^{\mathrm{base}}\sum_{k=1}^{K} \mathbb{E}_{x \sim \mathcal{D}^{\mathrm{p}}, y' \sim \pi^{\mathrm{base}}}\left[\phi(x, y')\phi(x, y')^\top\right],$$

and thus

$$
\left(\bar{\Sigma}_{\mathcal{D}_k^{\mathrm{r}}} + \gamma^{\mathrm{on}} I\right)^{-1} \preceq \left(\Sigma_{\mathcal{D}_1^{\mathrm{r}}} + \gamma^{\mathrm{on}} I + C^{\mathrm{base}} \sum_{k=1}^{K} \mathbb{E}_{x \sim \mathcal{D}^{\mathrm{p}}, y' \sim \pi^{\mathrm{base}}} \left[\phi(x, y') \phi(x, y')^{\top}\right]\right)^{-1}
$$

$$
= \frac{1}{C^{\mathrm{base}}} \left(\frac{1}{C^{\mathrm{base}}} \left(\Sigma_{\mathcal{D}_1^{\mathrm{r}}} + \gamma^{\mathrm{on}} I\right) + \sum_{k=1}^{K} \mathbb{E}_{x \sim \mathcal{D}^{\mathrm{p}}, y' \sim \pi^{\mathrm{base}}} \left[\phi(x, y') \phi(x, y')^{\top}\right]\right)^{-1}.
\tag{43}
$$

For ease of notation, let $d := |\mathcal{X}||\mathcal{Y}|$. Then, we have

$$
\sum_{k=1}^{K} \mathbb{E}_{x \sim \mathcal{D}^{\mathrm{p}}, y \sim \pi_k(\cdot|x)} \left[\|\phi(x, y)\|_{(\tilde{\Sigma}_{\mathcal{D}_k^{\mathrm{r}}} + \gamma^{\mathrm{on}} I)^{-1}}\right]
$$

$$
\leq \sqrt{K \sum_{k=1}^{K} \left(\mathbb{E}_{x \sim \mathcal{D}^{\mathrm{p}}, y \sim \pi_k(\cdot|x)} \left[\|\phi(x, y)\|_{(\tilde{\Sigma}_{\mathcal{D}_k^{\mathrm{r}}} + \gamma^{\mathrm{on}} I)^{-1}}\right]\right)^2}
$$

$$
\leq \sqrt{K \sum_{k=1}^{K} \mathbb{E}_{x \sim \mathcal{D}^{\mathrm{p}}, y \sim \pi_k(\cdot|x)} \left[\|\phi(x, y)\|_{(\tilde{\Sigma}_{\mathcal{D}_k^{\mathrm{r}}} + \gamma^{\mathrm{on}} I)^{-1}}^2\right]}
$$

$$
\overset{(a)}{\leq} \sqrt{2K \sum_{k=1}^{K} \mathbb{E}_{x \sim \mathcal{D}^{\mathrm{p}}, y \sim \pi_k(\cdot|x)} \left[\|\phi(x, y)\|_{(\bar{\Sigma}_{\mathcal{D}_k^{\mathrm{r}}} + \gamma^{\mathrm{on}} I)^{-1}}^2\right]}
$$

$$
= \sqrt{2K \sum_{k=1}^{K} \mathbb{E}_{x \sim \mathcal{D}^{\mathrm{p}}, y \sim \pi_k(\cdot|x), y' \sim \pi^{\mathrm{base}}(\cdot|x)} \left[\|\phi(x, y) - \phi(x, y') + \phi(x, y')\|_{(\bar{\Sigma}_{\mathcal{D}_k^{\mathrm{r}}} + \gamma^{\mathrm{on}} I)^{-1}}^2\right]}
$$

$$
\leq 2\sqrt{K} \Bigg( \sum_{k=1}^{K} \mathbb{E}_{x \sim \mathcal{D}^{\mathrm{p}}, y \sim \pi_k(\cdot|x), y' \sim \pi^{\mathrm{base}}(\cdot|x)} \left[\|\phi(x, y) - \phi(x, y')\|_{(\bar{\Sigma}_{\mathcal{D}_k^{\mathrm{r}}} + \gamma^{\mathrm{on}} I)^{-1}}^2\right]
$$

$$
+ \sum_{k=1}^{K} \mathbb{E}_{x \sim \mathcal{D}^{\mathrm{p}}, y' \sim \pi^{\mathrm{base}}(\cdot|x)} \left[\|\phi(x, y')\|_{(\bar{\Sigma}_{\mathcal{D}_k^{\mathrm{r}}} + \gamma^{\mathrm{on}} I)^{-1}}^2\right] \Bigg)^{\frac{1}{2}}
$$

$$
\overset{(b)}{\leq} 2\sqrt{K} \Bigg( 2 \log\left(\frac{\left(\frac{\gamma^{\mathrm{on}} + 4|\mathcal{D}_1^{\mathrm{r}}| + 4K}{d}\right)^d}{(\gamma^{\mathrm{on}})^d}\right) + \frac{1}{C^{\mathrm{base}}} \sum_{k=1}^{K} \mathbb{E}_{x \sim \mathcal{D}^{\mathrm{p}}, y' \sim \pi^{\mathrm{base}}(\cdot|x)} \Bigg[
$$

$$
\|\phi(x, y')\|_{\left(\frac{1}{C^{\mathrm{base}}} \left(\Sigma_{\mathcal{D}_1^{\mathrm{r}}} + \gamma^{\mathrm{on}} I\right) + \sum_{k'=1}^{k-1} \mathbb{E}_{x \sim \mathcal{D}^{\mathrm{p}}, \tilde{y} \sim \pi^{\mathrm{base}}(\cdot|x)}[\phi(x, \tilde{y})\phi(x, \tilde{y})^{\top}]\right)^{-1}}^2 \Bigg] \Bigg)^{\frac{1}{2}}
$$

$$
\overset{(c)}{\leq} 2\sqrt{K} \sqrt{2d \log\left(\frac{\gamma^{\mathrm{on}} + 4|\mathcal{D}_1^{\mathrm{r}}| + 4K}{d\gamma^{\mathrm{on}}}\right) + \frac{2}{C^{\mathrm{base}}} \log\left(\frac{\left(\frac{\frac{1}{C^{\mathrm{base}}}(\gamma^{\mathrm{on}} + |\mathcal{D}_1^{\mathrm{r}}|) + K}{d}\right)^d}{\left(\frac{\gamma^{\mathrm{on}}}{C^{\mathrm{base}}}\right)^d}\right)}
$$

$$
\leq 2\sqrt{2dK \left(\log\left(\frac{\gamma^{\mathrm{on}} + 4|\mathcal{D}_1^{\mathrm{r}}| + 4K}{d\gamma^{\mathrm{on}}}\right) + \frac{1}{C^{\mathrm{base}}} \log\left(\frac{\gamma^{\mathrm{on}} + |\mathcal{D}_1^{\mathrm{r}}| + C^{\mathrm{base}} K}{d\gamma^{\mathrm{on}}}\right)\right)},
$$

where inequality (a) comes from Lemma 13, inequality (b) uses Lemma 11 and Eq. (43), and inequality (c) is due to Lemma 11.

Thus, we can obtain the first statement.

Using a similar analysis as above, we can further obtain the second statement. $\qquad \square$

In the following, we prove Theorem 2.

*Proof of Theorem 2.* For this online setting, we also use events $\mathcal{E}$ and $\mathcal{F}$ defined in Eqs. (26) and (27).

Let $\delta' := \frac{\delta}{4}$. Then, according to Lemmas 1 and 3, we have $\Pr[\mathcal{E} \cap \mathcal{F} \cap \mathcal{G}^{\mathrm{on}}] \geq 1 - \delta$. Now it suffices to prove this theorem assuming that event $\mathcal{E} \cap \mathcal{F} \cap \mathcal{G}^{\mathrm{on}}$ holds. In the following proof, we assume that event $\mathcal{E} \cap \mathcal{F} \cap \mathcal{G}^{\mathrm{on}}$ holds.

For any $k \geq 1$ and $\bar{\lambda} \in [0, 2\rho]$, we have

$$
\left(\lambda^{k+1} - \bar{\lambda}\right)^2 = \left(\mathrm{Proj}_{[0,2\rho]}\left(\lambda_k + \eta_k \tilde{c}_k\right) - \mathrm{Proj}_{[0,2\rho]}\left(\bar{\lambda}\right)\right)^2
$$

$$
\overset{(a)}{\leq} \left(\lambda_k + \eta_k \tilde{c}_k - \bar{\lambda}\right)^2
$$

$$
= \left(\lambda_k - \bar{\lambda}\right)^2 + 2\eta_k \tilde{c}_k \left(\lambda_k - \bar{\lambda}\right) + \left(\eta_k\right)^2 \left(\tilde{c}_k\right)^2,
$$

where inequality (a) uses the nonexpansivity of the projection to $[0, 2\rho]$.

Summing the above inequality over $k = 1, \ldots, K$, we have

$$
0 \leq \left(\lambda_{K+1} - \bar{\lambda}\right)^2 \leq \left(\lambda_1 - \bar{\lambda}\right)^2 + \sum_{k=1}^{K} 2\eta_k \cdot \mathbb{E}_{x \sim \mathcal{D}^{\mathrm{p}}, y \sim \pi_k(\cdot|x)}[c^*(x, y)] \cdot \left(\lambda_k - \bar{\lambda}\right)
$$

$$
- \sum_{k=1}^{K} 2\eta_k \cdot \mathbb{E}_{x \sim \mathcal{D}^{\mathrm{p}}, y \sim \pi_k(\cdot|x)}[c^*(x, y)] \cdot \left(\lambda_k - \bar{\lambda}\right) + \sum_{k=1}^{K} 2\eta_k \tilde{c}_k \left(\lambda_k - \bar{\lambda}\right) + \sum_{k=1}^{K} \left(\eta_k\right)^2 \left(\tilde{c}_k\right)^2.
$$

Hence, we have

$$
\sum_{k=1}^{K} 2\eta_k \cdot \mathbb{E}_{x \sim \mathcal{D}^{\mathrm{p}}, y \sim \pi_k(\cdot|x)}[c^*(x, y)] \cdot \bar{\lambda} - \sum_{k=1}^{K} 2\eta_k \cdot \mathbb{E}_{x \sim \mathcal{D}^{\mathrm{p}}, y \sim \pi_k(\cdot|x)}[\hat{c}_k(x, y) - b_k^{\mathrm{c}}(x, y)] \cdot \lambda_k
$$

$$
\leq \left(\lambda_1 - \bar{\lambda}\right)^2 + \sum_{k=1}^{K} \left(\eta_k\right)^2 \left(\tilde{c}_k\right)^2 + \sum_{k=1}^{K} 2\eta_k \lambda_k \cdot \mathbb{E}_{x \sim \mathcal{D}^{\mathrm{p}}, y \sim \pi_k(\cdot|x)}[c^*(x, y) - \hat{c}_k(x, y) + b_k^{\mathrm{c}}(x, y)]
$$

$$
+ \sum_{k=1}^{K} 2\eta_k \left(\lambda_k - \bar{\lambda}\right) \left(\tilde{c}_k - \mathbb{E}_{x \sim \mathcal{D}^{\mathrm{p}}, y \sim \pi_k(\cdot|x)}[c^*(x, y)]\right).
$$

Using Lemma 6, we have

$$
\sum_{k=1}^{K} 2\eta_k \left(\mathbb{E}_{x \sim \mathcal{D}^{\mathrm{p}}, y \sim \pi_k(\cdot|x)}[c^*(x, y)] \cdot \bar{\lambda} + f(\pi^*; \hat{r}_k + b_k^{\mathrm{r}}) - f(\pi_k; \hat{r}_k + b_k^{\mathrm{r}})\right)
$$

$$
\leq \left(\lambda_1 - \bar{\lambda}\right)^2 + \sum_{k=1}^{K} \left(\eta_k\right)^2 \left(\tilde{c}_k\right)^2 + \sum_{k=1}^{K} 2\eta_k \lambda_k \cdot \mathbb{E}_{x \sim \mathcal{D}^{\mathrm{p}}, y \sim \pi_k(\cdot|x)}[c^*(x, y) - \hat{c}_k(x, y) + b_k^{\mathrm{c}}(x, y)]
$$

$$
+ \sum_{k=1}^{K} 2\eta_k \left(\lambda_k - \bar{\lambda}\right) \left(\tilde{c}_k - \mathbb{E}_{x \sim \mathcal{D}^{\mathrm{p}}, y \sim \pi_k(\cdot|x)}[c^*(x, y)]\right)
$$

$$
\overset{(a)}{\leq} \left(\lambda_1 - \bar{\lambda}\right)^2 + \sum_{k=1}^{K} \left(\eta_k\right)^2 \left(\tilde{c}_k\right)^2 + 4\sum_{k=1}^{K} \eta_k \lambda_k \cdot \mathbb{E}_{x \sim \mathcal{D}^{\mathrm{p}}, y \sim \pi_k(\cdot|x)}[b_k^{\mathrm{c}}(x, y)]
$$

$$
+ \sum_{k=1}^{K} 2\eta_k \left(\lambda_k - \bar{\lambda}\right) \left(\tilde{c}_k - \mathbb{E}_{x \sim \mathcal{D}^{\mathrm{p}}, y \sim \pi_k(\cdot|x)}[c^*(x, y)]\right),
$$

where inequality (a) uses the definition of event $\mathcal{G}^{\mathrm{on}}$.

Recall that $\eta_k = \eta$. Then, we have

$$
\sum_{k=1}^{K} \left(f(\pi^*) - f(\pi_k)\right) + \bar{\lambda} \sum_{k=1}^{K} \mathbb{E}_{x \sim \mathcal{D}^{\mathrm{p}}, y \sim \pi_k(\cdot|x)}[c^*(x, y)]
$$

$$\leq \frac{1}{2\eta} \left(\lambda_1 - \bar{\lambda}\right)^2 + \frac{\eta}{2} \sum_{k=1}^{K} (\tilde{c}_k)^2 + 2 \sum_{k=1}^{K} \lambda_k \cdot \mathbb{E}_{x \sim \mathcal{D}^{\mathrm{p}}, y \sim \pi_k(\cdot|x)}[b_k^{\mathrm{c}}(x,y)]$$

$$+ \sum_{k=1}^{K} \left(\lambda_k - \bar{\lambda}\right) \left(\tilde{c}_k - \mathbb{E}_{x \sim \mathcal{D}^{\mathrm{p}}, y \sim \pi_k(\cdot|x)}[c^*(x,y)]\right)$$

$$+ \sum_{k=1}^{K} \left(f(\pi^*) - f(\pi^*; \hat{r}_k + b_k^{\mathrm{r}})\right) - \sum_{k=1}^{K} \left(f(\pi_k) - f(\pi_k; \hat{r}_k + b_k^{\mathrm{r}})\right)$$

$$\leq \frac{1}{2\eta} \left(\lambda_1 - \bar{\lambda}\right)^2 + \frac{\eta(C_{\max})^2 K}{2} + 2 \sum_{k=1}^{K} \lambda_k \cdot \mathbb{E}_{x \sim \mathcal{D}^{\mathrm{p}}, y \sim \pi_k(\cdot|x)}[b_k^{\mathrm{c}}(x,y)]$$

$$+ \sum_{k=1}^{K} \left(\lambda_k - \bar{\lambda}\right) \left(\tilde{c}_k - \mathbb{E}_{x \sim \mathcal{D}^{\mathrm{p}}, y \sim \pi_k(\cdot|x)}[c^*(x,y)]\right)$$

$$+ K \cdot \mathbb{E}_{x \sim \mathcal{D}^{\mathrm{p}}, y \sim \pi^*(\cdot|x)}[r^*(x,y) - (\hat{r}_k(x,y) + b_k^{\mathrm{r}}(x,y))]$$

$$- \sum_{k=1}^{K} \mathbb{E}_{x \sim \mathcal{D}^{\mathrm{p}}, y \sim \pi_k(\cdot|x)}[r^*(x,y) - (\hat{r}(x,y) + b_k^{\mathrm{r}}(x,y))]$$

$$\leq \frac{1}{2\eta} \left(\lambda_1 - \bar{\lambda}\right)^2 + \frac{\eta(C_{\max})^2 K}{2} + 2 \sum_{k=1}^{K} \lambda_k \cdot \mathbb{E}_{x \sim \mathcal{D}^{\mathrm{p}}, y \sim \pi_k(\cdot|x)}[b_k^{\mathrm{c}}(x,y)]$$

$$+ \sum_{k=1}^{K} \left(\lambda_k - \bar{\lambda}\right) \left(\tilde{c}_k - \mathbb{E}_{x \sim \mathcal{D}^{\mathrm{p}}, y \sim \pi_k(\cdot|x)}[c^*(x,y)]\right) + 2 \sum_{k=1}^{K} \mathbb{E}_{x \sim \mathcal{D}^{\mathrm{p}}, y \sim \pi_k(\cdot|x)}[b_k^{\mathrm{r}}(x,y)].$$

Let $\bar{\lambda} = 0$. Recall that $\pi_K^{\mathrm{out}}$ is the uniform policy over $\pi_1, \ldots, \pi_K$ and $\eta := \frac{\lambda_1}{C_{\max}\sqrt{K}}$. Then, using Lemmas 2 and 7, we have

$$f(\pi^*) - f(\pi_K^{\mathrm{out}})$$

$$= \frac{1}{K} \sum_{k=1}^{K} \left(f(\pi^*) - f(\pi_k)\right)$$

$$\leq \frac{\lambda_1 C_{\max}}{\sqrt{K}} + \frac{2\rho}{K} \sum_{k=1}^{K} \mathbb{E}_{x \sim \mathcal{D}^{\mathrm{p}}, y \sim \pi_k(\cdot|x)}[b_k^{\mathrm{c}}(x,y)] + \frac{\rho}{K} \sum_{k=1}^{K} \left|\tilde{c}_k - \mathbb{E}_{x \sim \mathcal{D}^{\mathrm{p}}, y \sim \pi_k(\cdot|x)}[c^*(x,y)]\right|$$

$$+ \frac{2}{K} \sum_{k=1}^{K} \mathbb{E}_{x \sim \mathcal{D}^{\mathrm{p}}, y \sim \pi_k(\cdot|x)}[b_k^{\mathrm{r}}(x,y)]$$

$$= O\left( \frac{\lambda_1 C_{\max}}{\sqrt{K}} + \rho C_{\max} \sqrt{\frac{\log\left(\frac{K}{\delta}\right)}{N^{\mathrm{CE}}}} + \rho W \sqrt{\frac{\log\left(\frac{|\mathcal{X}||\mathcal{Y}|N^{\mathrm{CE}}K}{\delta}\right)}{M^{\mathrm{CE}}}} \right.$$

$$+ \rho \sqrt{\frac{(\exp(C_{\max}) + \exp(-C_{\max}) + 2)^2}{N^{\mathrm{on}}} \left(|\mathcal{X}||\mathcal{Y}| + \log\left(\frac{K}{\delta}\right)\right)} + \gamma^{\mathrm{on}}(C_{\max})^2 \cdot$$

$$\sqrt{\frac{|\mathcal{X}||\mathcal{Y}|}{K} \left(\log\left(\frac{\gamma^{\mathrm{on}} + |\mathcal{D}_1^{\mathrm{c}}| + K}{|\mathcal{X}||\mathcal{Y}|\gamma^{\mathrm{on}}}\right) + \frac{1}{C^{\mathrm{base}}} \log\left(\frac{\gamma^{\mathrm{on}} + |\mathcal{D}_1^{\mathrm{c}}| + C^{\mathrm{base}}K}{|\mathcal{X}||\mathcal{Y}|\gamma^{\mathrm{on}}}\right)\right)}$$

$$+ \sqrt{\frac{(\exp(R_{\max}) + \exp(-R_{\max}) + 2)^2}{N^{\mathrm{on}}} \left(|\mathcal{X}||\mathcal{Y}| + \log\left(\frac{K}{\delta}\right)\right)} + \gamma^{\mathrm{on}}(R_{\max})^2 \cdot$$

$$\left. \sqrt{\frac{|\mathcal{X}||\mathcal{Y}|}{K} \left(\log\left(\frac{\gamma^{\mathrm{on}} + |\mathcal{D}_1^{\mathrm{r}}| + K}{|\mathcal{X}||\mathcal{Y}|\gamma^{\mathrm{on}}}\right) + \frac{1}{C^{\mathrm{base}}} \log\left(\frac{\gamma^{\mathrm{on}} + |\mathcal{D}_1^{\mathrm{r}}| + C^{\mathrm{base}}K}{|\mathcal{X}||\mathcal{Y}|\gamma^{\mathrm{on}}}\right)\right)} \right).$$

Let $\bar{\lambda} = 2\rho$. Then, we have

$$f(\pi^*) - f(\pi_K^{\text{out}}) + 2\rho \mathbb{E}_{x \sim \mathcal{D}^{\text{p}}, y \sim \pi_K^{\text{out}}(\cdot|x)}[c^*(x,y)]$$

$$= \frac{1}{K} \sum_{k=1}^{K} (f(\pi^*) - f(\pi_k)) + \frac{2\rho}{K} \sum_{k=1}^{K} \mathbb{E}_{x \sim \mathcal{D}^{\text{p}}, y \sim \pi_k(\cdot|x)}[c^*(x,y)].$$

If $\frac{1}{K} \sum_{k=1}^{K} \mathbb{E}_{x \sim \mathcal{D}^{\text{p}}, y \sim \pi_k(\cdot|x)}[c^*(x,y)] \leq 0$, the second statement of the theorem naturally holds; Otherwise, we can replace the term $2\rho \mathbb{E}_{x \sim \mathcal{D}^{\text{p}}, y \sim \pi_K^{\text{out}}(\cdot|x)}[c^*(x,y)]$ by $2\rho [\mathbb{E}_{x \sim \mathcal{D}^{\text{p}}, y \sim \pi_K^{\text{out}}(\cdot|x)}[c^*(x,y)]]_+$ in the above inequality. Then, using Corollary 1 and Lemmas 2, 7 and 10, we obtain

$$\mathbb{E}_{x \sim \mathcal{D}^{\text{p}}, y \sim \pi_K^{\text{out}}(\cdot|x)}[c^*(x,y)]$$

$$\leq \frac{C_{\max}}{4\rho\sqrt{K}} \left( \frac{(\lambda_1 - 2\rho)^2}{\lambda_1} + \lambda_1 \right) + \frac{1}{K} \sum_{k=1}^{K} \mathbb{E}_{x \sim \mathcal{D}^{\text{p}}, y \sim \pi_k(\cdot|x)}[b_k^{\text{c}}(x,y)]$$

$$+ \frac{1}{2K} \sum_{k=1}^{K} \left| \tilde{c}_k - \mathbb{E}_{x \sim \mathcal{D}^{\text{p}}, y \sim \pi_k(\cdot|x)}[c^*(x,y)] \right| + \frac{1}{\rho K} \sum_{k=1}^{K} \mathbb{E}_{x \sim \mathcal{D}^{\text{p}}, y \sim \pi_k(\cdot|x)}[b_k^{\text{r}}(x,y)]$$

$$= O\left( \frac{C_{\max}}{\rho\sqrt{K}} \left( \frac{(\lambda_1 - 2\rho)^2}{\lambda_1} + \lambda_1 \right) + C_{\max} \sqrt{\frac{\log\left(\frac{K}{\delta}\right)}{N^{\text{CE}}}} + W \sqrt{\frac{\log\left(\frac{|\mathcal{X}||\mathcal{Y}|N^{\text{CE}}K}{\delta}\right)}{M^{\text{CE}}}} \right.$$

$$+ \sqrt{\frac{(\exp(C_{\max}) + \exp(-C_{\max}) + 2)^2}{N^{\text{on}}} \left( |\mathcal{X}||\mathcal{Y}| + \log\left(\frac{K}{\delta}\right) \right) + \gamma^{\text{on}}(C_{\max})^2} \cdot$$

$$\sqrt{\frac{|\mathcal{X}||\mathcal{Y}|}{K} \left( \log\left( \frac{\gamma^{\text{on}} + |\mathcal{D}_1^{\text{c}}| + K}{|\mathcal{X}||\mathcal{Y}|\gamma^{\text{on}}} \right) + \frac{1}{C^{\text{base}}} \log\left( \frac{\gamma^{\text{on}} + |\mathcal{D}_1^{\text{c}}| + C^{\text{base}}K}{|\mathcal{X}||\mathcal{Y}|\gamma^{\text{on}}} \right) \right)}$$

$$+ \frac{1}{\rho} \sqrt{\frac{(\exp(R_{\max}) + \exp(-R_{\max}) + 2)^2}{N^{\text{on}}} \left( |\mathcal{X}||\mathcal{Y}| + \log\left(\frac{K}{\delta}\right) \right) + \gamma^{\text{on}}(R_{\max})^2} \cdot$$

$$\left. \sqrt{\frac{|\mathcal{X}||\mathcal{Y}|}{K} \left( \log\left( \frac{\gamma^{\text{on}} + |\mathcal{D}_1^{\text{r}}| + K}{|\mathcal{X}||\mathcal{Y}|\gamma^{\text{on}}} \right) + \frac{1}{C^{\text{base}}} \log\left( \frac{\gamma^{\text{on}} + |\mathcal{D}_1^{\text{r}}| + C^{\text{base}}K}{|\mathcal{X}||\mathcal{Y}|\gamma^{\text{on}}} \right) \right)} \right).$$

$\square$

## F TECHNICAL TOOLS

In this section, we introduce several technical tools which are used in our analysis.

**Lemma 8** (Theorem 8.42 in (Beck, 2017)). *For any $\lambda \geq 0$ such that $q(\lambda) \leq u$,*

$$\lambda \leq \frac{u - f(\bar{\pi})}{-\mathbb{E}_{x \sim \mathcal{D}^{\text{p}}, y \sim \bar{\pi}(\cdot|x)}[c^*(x,y)]}.$$

*Proof.* For any $\lambda \geq 0$ such that $q(\lambda) \leq u$, we have

$$u \geq q(\lambda) \geq f(\bar{\pi}) - \lambda \cdot \mathbb{E}_{x \sim \mathcal{D}^{\text{p}}, y \sim \bar{\pi}(\cdot|x)}[c^*(x,y)].$$

Hence,

$$-\lambda \cdot \mathbb{E}_{x \sim \mathcal{D}^{\text{p}}, y \sim \bar{\pi}(\cdot|x)}[c^*(x,y)] \leq u - f(\bar{\pi}).$$

Since $\mathbb{E}_{x \sim \mathcal{D}^{\text{p}}, y \sim \bar{\pi}(\cdot|x)}[c^*(x,y)] < 0$, we have

$$\lambda \leq \frac{u - f(\bar{\pi})}{-\mathbb{E}_{x \sim \mathcal{D}^{\text{p}}, y \sim \bar{\pi}(\cdot|x)}[c^*(x,y)]}.$$

$\square$

Let $\Lambda^*$ be the set of the optimal solutions to the dual problem $\min_{\lambda \geq 0} q(\lambda)$.

**Corollary 1** (Corollary 8.43 in (Beck, 2017)). *For any $\lambda^* \in \Lambda^*$,*

$$\lambda^* \leq \frac{f(\pi^*) - f(\bar{\pi})}{-\mathbb{E}_{x \sim \mathcal{D}^{\mathrm{p}}, y \sim \bar{\pi}(\cdot|x)}[c^*(x, y)]} \leq \rho,$$

*where the second inequality comes from the definition of $\rho$.*

*Proof.* This corollary can be obtained by setting $u$ as the optimal value to the dual problem $\min_{\lambda \geq 0} q(\lambda) = f(\pi^*)$ in Lemma 8. $\qquad\square$

Define $g(\pi) := \mathbb{E}_{x \sim \mathcal{D}^{\mathrm{p}}, y \sim \pi(\cdot|x)}[c(x, y)]$. Let

$$v(u) := \max_{\pi} \{ f(\pi) : g(\pi) \leq u \},$$

$$C(u) := \{ \pi : g(\pi) \leq u \}.$$

**Lemma 9** (Theorem 3.59 in (Beck, 2017)). *For any $\lambda^* \in \Lambda^*$,*

$$v(0) + \lambda^* u \geq v(u).$$

*Proof.* For any $\pi$, we have

$$f(\pi) - \lambda^* g(\pi) \leq \max_{\pi} (f(\pi) - \lambda^* g(\pi)) = q(\lambda^*) = f(\pi^*) = v(0).$$

Thus, for any $u \in \mathbb{R}$ and $\pi \in C(u)$,

$$v(0) + \lambda^* u \geq f(\pi) - \lambda^* (g(\pi) - u) \geq f(\pi).$$

Since the above inequality holds for all $\pi \in C(u)$, by maximizing $f(\pi)$ over $\pi \in C(u)$, we have that for any $u \in \mathbb{R}$,

$$v(0) + \lambda^* u \geq v(u).$$

$\qquad\square$

**Lemma 10** (Theorem 3.60 in (Beck, 2017)). *If a policy $\tilde{\pi}$ satisfies that*

$$f(\pi^*) - f(\tilde{\pi}) + \rho'[g(\tilde{\pi})]_+ \leq L,$$

*where $L > 0$ and $\rho' \geq 2\lambda^*$, then*

$$[g(\tilde{\pi})]_+ \leq \frac{2L}{\rho'}.$$

*Proof.* From Lemma 9, we have that for any $u \in \mathbb{R}$,

$$v(0) - v(u) \geq -\lambda^* u.$$

Let $\tilde{u} := [g(\tilde{\pi})]_+$. Then, we have

$$(\rho' - \lambda^*) \tilde{u} \leq \rho' \tilde{u} + v(0) - v(\tilde{u})$$
$$\overset{(a)}{\leq} f(\pi^*) - f(\tilde{\pi}) + \rho' \tilde{u}$$
$$\leq L,$$

where inequality (a) uses the fact that $v(0) = f(\pi^*)$ and $v(\tilde{u}) \geq f(\tilde{\pi})$.

Since $\rho' \geq 2\lambda^*$, we have

$$\tilde{u} \leq \frac{L}{\rho' - \lambda^*} \leq \frac{L}{\rho' - \frac{\rho'}{2}} = \frac{2L}{\rho'}.$$

$\qquad\square$

**Lemma 11.** *Let $\psi_1, \ldots, \psi_K$ be a sequence of $d$-dimensional random vectors following distributions $\mathcal{B}_1, \ldots, \mathcal{B}_K$, respectively, and we have $\|\psi_k\| \le L$ for any $k \ge 1$. Let $A_0$ be a $d \times d$ positive definite matrix such that $\sigma_{\min}(A_0) \ge \{1, L^2\}$, and define $A_k = A_0 + \sum_{i=1}^{k} \mathbb{E}_{\psi_i \sim \mathcal{B}_i}[\psi_i \psi_i^\top]$ for any $k \ge 1$. Then, we have*

$$\sum_{k=1}^{K} \mathbb{E}_{\psi_k \sim \mathcal{B}_k} \left[ \|\psi_k\|^2_{(A_{k-1})^{-1}} \right] \le 2 \log \frac{\det(A_K)}{\det(A_0)} \le 2 \log \left( \frac{\left( \frac{\text{trace}(A_0) + KL^2}{d} \right)^d}{\det(A_0)} \right).$$

*Proof.* This proof uses a similar analytical procedure as Lemma 11 in (Abbasi-Yadkori et al., 2011).

We have

$$\begin{aligned}
\det(A_K) &= \det\left( A_{K-1} + \mathbb{E}_{\psi_K \sim \mathcal{B}_K} \left[ \psi_K \psi_K^\top \right] \right) \\
&= \det(A_{K-1}) \det\left( I + (A_{K-1})^{-\frac{1}{2}} \mathbb{E}_{\psi_K \sim \mathcal{B}_K} \left[ \psi_K \psi_K^\top \right] (A_{K-1})^{-\frac{1}{2}} \right) \\
&= \det(A_{K-1}) \det\left( I + \mathbb{E}_{\psi_K \sim \mathcal{B}_K} \left[ (A_{K-1})^{-\frac{1}{2}} \psi_K \left( (A_{K-1})^{-\frac{1}{2}} \psi_K \right)^\top \right] \right) \\
&= \det(A_{K-1}) \left( 1 + \mathbb{E}_{\psi_K \sim \mathcal{B}_K} \left[ \|\psi_K\|^2_{(A_{K-1})^{-1}} \right] \right) \\
&= \det(A_0) \prod_{k=1}^{K} \left( 1 + \mathbb{E}_{\psi_k \sim \mathcal{B}_k} \left[ \|\psi_k\|^2_{(A_{k-1})^{-1}} \right] \right).
\end{aligned}$$

Taking logarithm on both sides, we have

$$\log \det(A_K) = \log \det(A_0) + \sum_{k=1}^{K} \log \left( 1 + \mathbb{E}_{\psi_k \sim \mathcal{B}_k} \left[ \|\psi_k\|^2_{(A_{k-1})^{-1}} \right] \right).$$

Since $\sigma_{\min}(A_0) \ge \{1, L^2\}$, we have $\|\psi_k\|^2_{(A_{k-1})^{-1}} \le 1$ for any $k \ge 1$. Using the fact that $x \le 2 \log(1 + x)$, we have

$$\begin{aligned}
\sum_{k=1}^{K} \mathbb{E}_{\psi_k \sim \mathcal{B}_k} \left[ \|\psi_k\|^2_{(A_{k-1})^{-1}} \right] &\le 2 \sum_{k=1}^{K} \log \left( 1 + \mathbb{E}_{\psi_k \sim \mathcal{B}_k} \left[ \|\psi_k\|^2_{(A_{k-1})^{-1}} \right] \right) \\
&= 2 \log \frac{\det(A_K)}{\det(A_0)} \\
&\overset{(a)}{\le} 2 \log \left( \frac{\left( \frac{\text{trace}(A_0) + KL^2}{d} \right)^d}{\det(A_0)} \right),
\end{aligned}$$

where inequality (a) uses the AM-GM inequality. $\square$

**Lemma 12** (Lemma H.3 in (Agarwal et al., 2020))**.** *Let $\mathcal{B}$ be a distribution of $d$-dimensional vectors which satisfies that $\|\psi\| \le L$ if $\psi \sim \mathcal{B}$. Let $\psi_1, \ldots, \psi_M$ be $M$ i.i.d. samples from $\mathcal{B}$, and define $A = \mathbb{E}_{\psi \sim \mathcal{B}}[\psi \psi^\top]$. Then, with probability at least $1 - \delta$, we have that for any $v \in \mathbb{R}^d$,*

$$\left| v^\top \left( \frac{1}{M} \sum_{i=1}^{M} \psi_i \psi_i^\top - A \right) v \right| \le \frac{2L^2 \ln\left( \frac{8\hat{d}}{\delta} \right)}{3M} + L^2 \sqrt{\frac{2 \ln\left( \frac{8\hat{d}}{\delta} \right)}{M}},$$

*where $\hat{d} := \frac{\text{trace}(A)}{\|A\|}$ is the intrinsic dimension of $A$.*

*Proof.* For any $i \ge 1$, let $D_i := \psi_i \psi_i^\top - A$. Then, we have that $\mathbb{E}[D_i] = 0$, $\|D_i\| \le L^2$, and

$$\left\| \sum_{i=1}^{M} \mathbb{E}[(D_i)^2] \right\| \le ML^4.$$

The intrinsic dimension of $\sum_{i=1}^{M} \mathbb{E}[(D_i)^2]$ is equal to that of $A$, which is $\hat{d}$ by definition.

Using the Matrix Bernstein inequality (Theorem 7.7.1 in (Tropp et al., 2015)), we have that for any $t \geq L^2\sqrt{M} + \frac{L^2}{3}$,

$$\Pr\left[\sigma_{\max}\left(\sum_{i=1}^{M} \mathbb{E}[(D_i)^2]\right) \geq t\right] \leq 4\hat{d} \cdot \exp\left(\frac{-\frac{t^2}{2}}{L^4 M + \frac{L^2 t}{3}}\right).$$

Setting $t' = \frac{t}{M}$, we have that for any $t' \geq \frac{L^2}{\sqrt{M}} + \frac{L^2}{3M}$,

$$\Pr\left[\sigma_{\max}\left(\frac{1}{M}\sum_{i=1}^{M} \mathbb{E}[(D_i)^2]\right) \geq t'\right] \leq 4\hat{d} \cdot \exp\left(\frac{-\frac{M(t')^2}{2}}{L^4 + \frac{L^2 t'}{3}}\right).$$

When $t' = \frac{2L^2 \ln\left(\frac{4\hat{d}}{\delta}\right)}{3M} + L^2\sqrt{\frac{2\ln\left(\frac{4\hat{d}}{\delta}\right)}{M}}$, we have $4\hat{d} \cdot \exp\left(\frac{-\frac{M(t')^2}{2}}{L^4 + \frac{L^2 t'}{3}}\right) \leq \delta$.

Hence, with probability at least $1 - \delta$, we have

$$\sigma_{\max}\left(\frac{1}{M}\sum_{i=1}^{M} \mathbb{E}[(D_i)^2]\right) \leq \frac{2L^2 \ln\left(\frac{4\hat{d}}{\delta}\right)}{3M} + L^2\sqrt{\frac{2\ln\left(\frac{4\hat{d}}{\delta}\right)}{M}}.$$

We can obtain this concentration inequality in the other direction with a similar argument. Therefore, we complete the proof of this lemma. $\square$

**Lemma 13** (Lemma H.4 in (Agarwal et al., 2020)). *Let* $\mathcal{B}_1, \ldots, \mathcal{B}_K$ *be* $K$ *distributions of* $d$-*dimensional vectors. For any* $i \in [K]$, *we draw* $M$ *i.i.d. samples* $\psi_{i,1}, \ldots, \psi_{i,M}$ *from* $\mathcal{B}_i$, *and form* $\hat{A}_i = \frac{1}{M}\sum_{j=1}^{M} \psi_{i,j}\psi_{i,j}^{\top}$. *Define* $A_i = \mathbb{E}_{\psi \sim \mathcal{B}_i}[\psi\psi^{\top}]$, $A = \sum_{i=1}^{K} A_i + \gamma I$, *and* $\hat{A} = \sum_{i=1}^{K} \hat{A}_i + \gamma I$. *Setting* $M := \frac{32K^2 \ln\left(\frac{8K\tilde{d}}{\delta}\right)}{\gamma^2}$, *with probability at least* $1 - \delta$, *we have that for any* $v \in \mathbb{R}^d$,

$$\frac{1}{2}v^{\top}(A + \gamma I)^{-1}v \leq v^{\top}(\hat{A} + \gamma I)^{-1}v \leq 2v^{\top}(A + \gamma I)^{-1}v,$$

*where* $\tilde{d} := \max_{i \in [K]} \frac{\text{trace}(A_i)}{\|A_i\|}$.

*Proof.* Let $\alpha(M) =: \frac{2L^2 \ln\left(\frac{8K\tilde{d}}{\delta}\right)}{3M} + L^2\sqrt{\frac{2\ln\left(\frac{8K\tilde{d}}{\delta}\right)}{M}}$. Using Lemma 12, we have that with probability $1 - \delta$, for any $i \in [K]$,

$$A_i + \alpha(M)I + \frac{\gamma}{K}I \succeq \hat{A}_i + \frac{\gamma}{K}I \succeq A_i - \alpha(M)I + \frac{\gamma}{K}I.$$

Hence, we have

$$A + K\alpha(M)I + \gamma I \succeq \hat{A} + \gamma I \succeq A - K\alpha(M)I + \gamma I.$$

When $\gamma \geq 2K\alpha(M)$, the above inequality implies

$$(A + K\alpha(M)I + \gamma I)^{-1} \preceq \left(\hat{A} + \gamma I\right)^{-1} \preceq (A - K\alpha(M)I + \gamma I)^{-1}.$$

Let $U\Lambda U^T$ be the eigendecomposition of $A$, where $\Lambda = \text{diag}(\sigma_1, \ldots, \sigma_d)$ and $U = [u_1, \ldots, u_d]$. Then, we have

$$v^{\top}(\hat{A} + \gamma I)^{-1}v - v^{\top}(A + \gamma I)^{-1}v \leq v^{\top}\left((A - K\alpha(M)I + \gamma I)^{-1} - (A + \gamma I)^{-1}\right)v$$

$$= \sum_{i=1}^{d}\left((\sigma_i + \gamma - K\alpha(M))^{-1} - (\sigma_i + \gamma)^{-1}\right)(v^{\top}u_i)^2.$$

For any $i \in [d]$, since $\sigma_i \geq 0$, we have $\sigma_i + \gamma \geq 2K\alpha(M)$, and then $2(\sigma_i + \gamma - K\alpha(M)) \geq \sigma_i + \gamma$, which implies $(\sigma_i + \gamma - K\alpha(M))^{-1} \leq 2(\sigma_i + \gamma)^{-1}$. Therefore, we have

$$v^\top (\hat{A} + \gamma I)^{-1} v - v^\top (A + \gamma I)^{-1} v \leq \sum_{i=1}^{d} (\sigma_i + \gamma)^{-1} (v^\top u_i)^2 = v^\top (A + \gamma I)^{-1} v.$$

Using a similar analysis, we can obtain the statement in the other direction. $\qquad\square$

