# OpenReview forum: "Primal-Dual Direct Preference Optimization for Constrained LLM Alignment"
_ICLR.cc/2026/Conference — Submitted to ICLR 2026_

### Official Review · Reviewer_QQdF · 2025-10-31

**Soundness:** 2
**Presentation:** 3
**Contribution:** 2
**Rating:** 2
**Confidence:** 3

**Summary:**

The author introduce a approach called Primal-Dual Direct Preference Optimization (PD-DPO). The key contribution: "rearranged Lagrangian DPO objective" (Eq. 14). Since preferences are collected separately for rewards ($r$) and costs ($c$), DPO cannot be directly applied to the Lagrangian $L(\pi; \lambda) = r - \lambda c$. The authors rearrange the optimality conditions to express the cost preference likelihood using the reward function. They then substitute the unknown reward $r$ with the implicit reward information captured by a policy $\pi_{\hat{r}}^{\ast}$, which is pre-trained using standard DPO on reward preferences. The proposed algorithm (Algorithm 1) iteratively updates the policy (primal step) using this rearranged objective and updates the Lagrange multiplier $\lambda$ (dual step) via projected subgradient descent. The dual update relies on estimating the current policy's cost using online binary human feedback.
For experiments, they demonstrate PD-DPO’s superior performance compared to SFT and SafeDPO, while improving harmlessness over SFT. However, it is significantly less safe than the computationally intensive Safe RLHF baseline (Beaver-v3.0).

**Strengths:**

1. The challenge of applying DPO when preferences are separated by objective (reward vs. cost) is large. The author clearly expresses cost in terms of reward and the optimal Lagrangian policy, and then substituting the reward using a pre-trained standard DPO policy is a clever solution (bypassing the need for explicit reward and cost modeling during constrained optimization).
2. Extension to the online setting. The reliance on comprehensive offline data coverage is a bottleneck for DPO. The incorporation of exploration bonuses directly into the rearranged DPO objective to guide exploration in a constrained setting is novel.
3. The adoption of a primal-dual framework for the algorithm is theoretically sound compared to methods that require prior knowledge or the sweeping of the Lagrange multiplier. By integrating the dual update directly into the optimization loop, the algorithm considers the optimal tradeoff between reward maximization and constraint satisfaction.

**Weaknesses:**

1. In Appendix B, lines 700-701, you mention that you fix the Lagrange multiplier $\lambda$ to 5 due to computational limits, instead of running the dual update. This might limit the empirical validation of the paper. Compared to Beaver-v3.0, the results in Figure 1 only validate the rearranged DPO objective with a fixed penalty. Was any experimental validation done on the primal-dual part for solving the constrained problem?
2. The cost estimation procedure required for the dual update is expensive in the context of real-world LLM training. require: sampling $N^{CE}$ responses and asking $M^{CE}$ human annotators for binary feedback at every iteration K. It is an synchronous demand for human interaction inside the training loop, which is slow, expensive, and likely high variance. The massive annotation cost that convergence would incur, as suggested by the obtained bound in Theorem 1, seems to outweigh the computational gains of avoiding reward/cost model training.
3. The results of this paper rely on the accuracy of $\pi_{\hat{r}}^{\ast}$. I assume this policy, trained by the standard DPO, is used as a fixed stand-in for the reward signal on Eq. 16 throughout the constrained optimization. Therefore, any error, bias, or overly optimistic compromise in $\pi_{\hat{r}}^{\ast}$ will be carried over to the PD-DPO training. You did not account for this in your experiments; what happens if $\pi_{\hat{r}}^{\ast}$ is a poor candidate?
4. Calculating these exploration bonuses in O-PD-DPO also requires constructing and inverting covariance matrices based on the feature representations $\phi(x,y)$ (also mentioned in Appendix D). This attempt to work in the high-dimensional space of LLMs is computationally infeasible.
5. PD-DPO also shows a harmfulness gap to Beaver-v3.0 (e.g., Elo score < 1100 vs ~1400); this trade-off is unacceptable in safety-critical applications. You argued that this trade-off is expected for the gains in efficiency,  but reduction in safety suggests the method struggles to enforce constraints strictly (perhaps exacerbated by the use of a fixed $\lambda$).

**Questions:**

1. Any experimental results that validate the full Algorithm to show the trajectory of $\lambda_k$, rewards, and costs across iterations K?
2. Any clarification on the practicality of the cost estimation step (Algorithm 1, Line 4). Can you quantify the total human annotation effort required for convergence as suggested by your theory? Given this burden, how do you propose making the dual update feasible in a practical LLM pipeline? Have you considered using an offline-trained cost model just for the dual update estimation to avoid human-in-the-loop feedback?
3. How sensitive is the PD-DPO framework to the quality of the initial standard DPO model? Errors in this model directly impact the optimization objective (Eq. 16). Any ablation study on the performance of the final constrained policy when $\pi_{\hat{r}}^{\ast}$ is trained with varying amounts of reward preference data?
4. Any additional experiments conducted for O-PD-DPO?
5. [related to weakness 5]] (and just some suggestions) Have you analyzed the Pareto front achievable by PD-DPO (e.g., by varying the initial $\lambda_1$ or the optimization parameters, assuming the full algorithm is run)? Is it possible to achieve higher safety with PD-DPO, even at the cost of some helpfulness?

---

> ### Author Response · Authors · 2025-11-27
> **Response to Reviewer QQdF (Part 1/2)**
>
> Thank you very much for your time and effort in reviewing our paper! We have revised our paper according to your review and highlighted the revision in *cyan* color.
>
> For your comments W1, W5, Q1, Q4 and Q5 on the experiment part, please see our general response to all reviewers for our new experimental results. We are running experiments for the  trajectories of $\lambda$, rewards and costs, algorithm O-PD-DPO and the ablation study on  $\pi^*_{\hat{r}}$, and will add them to our revision once they finish.
>
> **W1. Experiments on the Primal-dual Part**
>
> **W2. Cost Estimation**
>
> We agree that performing real human annotation for the cost estimation scheme is expensive. This cost estimation scheme is a theoretical algorithm design in order to derive rigorous theoretical guarantees. In implementation, we can use alternative schemes to replace it, e.g., quering advanced LLMs to evaluate the costs of the responses generated by the current policy.
>
> Theoretically, the variance and error of the cost estimation scheme in our algorithm scale as $\frac{1}{N^{CE}}$ and $\frac{1}{M^{CE}}$, which will not be large if $N^{CE}$ and $M^{CE}$ are large enough. The main advantage of our algorithm is to reduce the number of required trained and loaded models from 3 to 2 compared to prior works [Dai et al., 2024; Liu et al., 2024b; Huang et al., 2024; Zhang et al., 2025], which significantly saves memory costs.
>
>
> Moreover, during rebuttal, we came up with a variant of our algorithm to solve the cost estimation implementation issue, i.e., first train a cost model and then train the reward-cost-aligned language model on reward preference data using the following objective:
>
> $$
> \min_{\pi} - \sum_{(x,y^{rw},y^{r\ell}) \in \mathcal{D}^r} \log \sigma \Bigg( \bigg( \lambda \cdot c(x,y^{rw}) + \beta \log\frac{\pi(y^{rw}|x)}{\pi_{ref}(y^{rw}|x)} \bigg) - \bigg( \lambda \cdot c(x,y^{r\ell}) + \beta \log\frac{\pi(y^{r\ell}|x)}{\pi_{ref}(y^{r\ell}|x)} \bigg) \Bigg)
> $$
>
> This variant shares the same main idea as the original version of our algorithm, and also only needs to train 2 models rather than 3 models as in [Dai et al., 2024; Liu et al., 2024b; Huang et al., 2024; Zhang et al., 2025]. Its advantage is that we can utilize the trained cost model to evaluate the cost of the current policy and update the Lagrange multiplier. We are implementing this variant, and will include its results in our revision once we finish its experiments.
>
>
> **W3. The accuracy of $\pi^*_{\hat{r}}$**
>
> We considered the error on $\pi^*_{\hat{r}}$ in both theory and experiments. That is why we represent it by the notation $\pi^*_{\hat{r}}$ instead of $\pi^*_{r^*}$.
>
> In theory, we incorporate the error brought by the imperfect learning of the reward-aligned model in the results, i.e., the $\|\phi(x,y)\|_{(\Sigma_{\mathcal{D}^r}+\gamma I)^{-1}}$ terms in Theorem 1, which depend on the number and coverage of reward preference data. In experiments, we first train $\pi^*_{\hat{r}}$ using reward preference data by ourselves, and then train the reward-cost-aligned model. Thus, the error brought by the imperfection of $\pi^*_{\hat{r}}$ is automatically incorporated in the final empirical performance of our reward-cost-aligned model.
>
> **W4. Exploration Bonuses**
>
> We agree that directly calculating exploration bonuses using $\phi(x,y)$ is computationally infeasible in practice. This is a design for the theoretical algorithm in order to derive rigorous theoretical guarantees. In practice, we can borrow methods from the deep RL literature, e.g., random network distillation (RND) [Burda et al., 2018], to construct exploration bonuses. The RND method trains a neural network to predict the state embeddings output by a randomly initialized and fixed neural network. RND takes visited states as training examples and takes prediction error as the exploration bonus, since the prediction error will be low for visited states and high for rarely visited states. With this RND method, we can efficiently construct exploration bonuses to enable exploration on the unvisited prompt-response space.
>
> **W5. Experimental Performance on Harmlessness**
>
> **Q1. Experiments on the Trajectories of $\lambda_k$, Rewards and Costs**
>
> **Q2. Implementation of Cost Estimation**
>
> Please see our reply to W2. Thank you for your insightful suggestion. Training a small (e.g., 1B) cost model just for the cost estimation in dual update may be a good idea.

---

> > ### Author Response · Authors · 2025-11-27
> > **Response to Reviewer QQdF (Part 2/2)**
> >
> > **Q3. Influence of  $\pi^*_{\hat{r}}$**
> >
> > In theory, the influence of $\pi^*_{\hat{r}}$ on the final performance of the reward-cost-aligned model is reflected in the
> >
> > $\\|\phi(x,y)\\|_{(\Sigma+\gamma I)^{-1}}$
> >
> > terms in Theorem 1, which depend on the number and coverage of reward preference data (due to the formula display issue on OpenReview, here we use $\Sigma$ to denote $\Sigma_{\mathcal{D}^r}$). The larger the number of reward preference data is, or the broader the reward preference data covers, the smaller these terms are and the better performance the final reward-cost-aligned model can achieve.
> >
> > **Q4. Experiments for O-PD-DPO**
> >
> > **Q5. Experiments on the Pareto front**
> >
> > ---
> >
> > References:
> >
> > Josef Dai, Xuehai Pan, Ruiyang Sun, Jiaming Ji, Xinbo Xu, Mickel Liu, Yizhou Wang, and Yaodong
> > Yang. Safe RLHF: Safe reinforcement learning from human feedback. ICLR, 2024.
> >
> > Zixuan Liu, Xiaolin Sun, and Zizhan Zheng. Enhancing LLM safety via constrained direct preference
> > optimization. ICLR Workshop, 2024b.
> >
> > Xinmeng Huang, Shuo Li, Edgar Dobriban, Osbert Bastani, Hamed Hassani, and Dongsheng Ding.
> > One-shot safety alignment for large language models via optimal dualization. NeurIPS, 2024.
> >
> > Botong Zhang, Shuo Li, Ignacio Hounie, Osbert Bastani, Dongsheng Ding, and Alejandro Ribeiro.
> > Alignment of large language models with constrained learning. ICML Workshop, 2025.
> >
> > Yuri Burda, Harrison Edwards, Amos Storkey, and Oleg Klimov. Exploration by random network distillation. ICLR, 2019.

---

### Official Review · Reviewer_kvKV · 2025-10-31

**Soundness:** 2
**Presentation:** 2
**Contribution:** 2
**Rating:** 4
**Confidence:** 3

**Summary:**

The authors propose a direct preference optimization like procedure for safety constrained alignment, that foregoes the use of trained reward models and constraint costs by using labels to directly construct a supervised Lagrangian objective. They provide near optimality and feasibility guarantees for problems solved by dual ascent.

**Strengths:**

All existing constrained alignment approaches explicitly train reward and cost models, learning directly from helpfulness and safety  preference binary labels is both novel in the context of constrained alignment and relevant. The paper provides near feasibility and optimality last iterate guarantees under standard assumptions.

**Weaknesses:**

The two stage approach fits a model using DPO and then uses the implicit reward given by this model in the training objective (e.q. 14). It then uses a labeling oracle for the safety cost. This is indeed a novel and reasonable approach. My main concern is that statement that the main advantage of their method is not fitting reward and cost models might be overstated in the sense that (1) the first phase is indeed fitting a reward model and (2) the second phase swaps a cost model for a labeling oracle.

The big memory gains with respect to using explicit reward an cost models disappear if the reward and costs are evaluated offline for the dataset of prompts and responses - responses are not sampled at each training step but at the end of each training epoch or primal update step. This is indeed the approach used in prior work (Huang et. al and Zhang et. al) that use DPO style losses.

Also, the formulation of Lagrangian maximization using DPO style losses was proposed at least in Huang et. al (referenced in the submission), and doing dual super-gradient descent on lambda by sampling the model and estimating the slack in Zhang et. al (also referenced in the submission) The only discussion about the distinction between this prior work and the proposed approach is the aforementioned lack of explicit cost and reward models. Although these prior works also have feasibility and optimality guarantees, there is no discussion how do the theoretical results compare to those.

Finally, the experiments present a single run with a single constrained baseline, where the method performs comparatively poorly in terms of safety. Without more experiments it is hard to evaluate how the proposed approach performs in terms of helpfulness/harmlessness trade offs (i.e. its pareto optimality) and, more importantly, wether it empirically succeeds at obtaining near feasible solutions to support the theoretical results.

**Questions:**

Can you provide additional experimental evidence supporting the performance of your method? The only point you provide does very badly in terms of the constraints compared to the baseline. The number of experiments is very limited, only a single run/performance is reported.

Can you point out the relation of your theoretical results to those in Zhang et. al (referenced in the submission) ?

Can you comment on the choice of constraint threshold and perhaps do an ablation on its impact?

Can you include plots of dual variable dynamics and or their values in the appendix?

---

> ### Author Response · Authors · 2025-11-27
> **Response to Reviewer kvKV (Part 1/2)**
>
> Thank you very much for your time and effort in reviewing our paper! We have revised our paper according to your comments and highlighted the revision in *blue* color.
>
> **W1. Advantage of Our Approach and Overstatement**
>
> **The main advantage of our approach is that it only needs to train and load 2 models**, i.e., the reward-aligned language model and reward-cost-aligned language model, without requiring prior knowledge of the optimal Lagrange multiplier, while prior approaches [Dai et al., 2024; Liu et al., 2024b; Huang et al., 2024; Zhang et al., 2025] need to train and load 3 models, i.e., the reward model (or reward-aligned language model), cost model (or cost-aligned language model) and reward-cost-aligned language model. This significantly reduces memory costs. We have revised the advantage statements in our revision to make it clear.
>
> **W2. Difference from [Huang et al., 2024; Zhang et al., 2025]**
>
> We truly appreciate prior works [Huang et al., 2024; Zhang et al., 2025], and have added more discussion and clarification on the difference from these two works in our revision.
>
> The ideas and main focuses of algorithm design and theoretical analysis in [Huang et al., 2024; Zhang et al., 2025] and our work are very different.
> They investigate the constrained alignment problem from the perspective of dual optimization.
>
> Specifically, in algorithm design, the main focus of [Huang et al., 2024] is to avoid the expensive computation of evaluating the optimal policy at every dual update step. They derive an explicit form of the dual function, which only involves the SFT model instead of the optimal policy under the current dual variable. Using this derivation, their algorithm first computes the optimal Lagrange multiplier, and then computes the optimal policy under the optimal Lagrange multiplier.
> The algorithm in [Zhang et al., 2025] alternates between updating LLM policy via Lagrangian maximization and updating the Lagrange multiplier via dual descent. While the algorithms in [Huang et al., 2024; Zhang et al., 2025] can also use DPO-style losses, their algorithms need to train and load 3 models, i.e., the reward model (or reward-aligned language model), cost model (or cost-aligned language model) and reward-cost-aligned language model.
>
> In contrast, the main focus of our algorithm design is to adopt a rearranged primal-dual DPO objective to reduce the number of required trained and loaded models from 3 to 2, without requiring prior knowledge of the optimal Lagrange multiplier.
> We do not focus on dual variable optimization or update as in [Huang et al., 2024; Zhang et al., 2025], and did not claim novelty on dual variable update.
>
> In theoretical analysis, [Huang et al., 2024] builds their analysis upon $\varepsilon$-accurate reward and cost models $\hat{r}$ and $\hat{c}$, expresses their results by this accuracy parameter $\varepsilon$, and requires an assumption that $\pi^*$ is feasible under the used cost model $\hat{c}$, which is hard to verify in practice. [Zhang et al., 2025] focuses on analyzing the primal-dual gap brought by policy parameterization, instead of the error due to learning reward and cost functions from preference data.
>
> In contrast, our analysis builds directly upon the DPO-style algorithm  (which proves the equivalence between the RLHF-style objective and our rearranged DPO-style objective), incorporates the influence of the number and coverage of preference data into results, and does not require extra assumptions.
> In addition, we provide theoretical results for the online exploration version of our algorithm, which remove the dependence on preference data coverage.
> Due to the difference in the needed assumptions and main focuses, the theoretical results in [Huang et al., 2024; Zhang et al., 2025] and our work cannot be directly compared.
>
>
> **W3 and Q1. Experiments**
>
> Please see our general response to all reviewers.
>
> **Q2. Relation of our theoretical results to those in [Zhang et al., 2025]**
>
> Please see the last and last second paragraphs of our reply to W2.
>
> In summary, [Zhang et al., 2025] focuses on analyzing the primal-dual gap brought by policy parameterization, and does not consider the error due to learning reward and cost functions from preference data. By contrast, we focus on analyzing the error brought by learning reward and cost functions from preference data using our  rearranged DPO objective and the intrinsic error of the primal-dual framework. We do not consider the error due to policy parameterization. Thus, the theoretical results in [Zhang et al., 2025] and ours cannot be directly compared.

---

> ### Author Response · Authors · 2025-11-27
> **Response to Reviewer kvKV (Part 2/2)**
>
> **Q3. Constraint Threshold and Q4. Dual Variable Dynamics**
>
> In our experiments, we set the constraint threshold to zero to keep consistent with our paper and prior work [Dai et al., 2024].
>
> Please see our general response for our new experimental results. We are running experiments for the ablation study on the constraint threshold and dual variable dynamics, and will add them to our revision once they finish.
>
> ---
>
> References:
>
> Josef Dai, Xuehai Pan, Ruiyang Sun, Jiaming Ji, Xinbo Xu, Mickel Liu, Yizhou Wang, and Yaodong
> Yang. Safe RLHF: Safe reinforcement learning from human feedback. ICLR, 2024.
>
> Zixuan Liu, Xiaolin Sun, and Zizhan Zheng. Enhancing LLM safety via constrained direct preference
> optimization. ICLR Workshop, 2024b.
>
> Xinmeng Huang, Shuo Li, Edgar Dobriban, Osbert Bastani, Hamed Hassani, and Dongsheng Ding.
> One-shot safety alignment for large language models via optimal dualization. NeurIPS, 2024.
>
> Botong Zhang, Shuo Li, Ignacio Hounie, Osbert Bastani, Dongsheng Ding, and Alejandro Ribeiro.
> Alignment of large language models with constrained learning. ICML Workshop, 2025.

---

### Official Review · Reviewer_EtwF · 2025-10-31

**Soundness:** 2
**Presentation:** 4
**Contribution:** 3
**Rating:** 4
**Confidence:** 2

**Summary:**

For both offline and online constrained alignment problem of LLMs, this work develops a novel primal-dual DPO approach which does not require trained reward and cost models or prior knowledge of the optimal Lagrange multiplier. Then the performance of this algorithm is demonstrated by theoretical convergence rate of suboptimality and cost violation, and experimental results.

**Strengths:**

The targeted problem of LLM safety alignment is important. The algorithm design looks novel, by obtaining analytical solution of cost and reward. The presentation is clear.

**Weaknesses:**

The major problem is the costly algorithm and problematic convergence bounds, as shown in the questions 1 and 2 below respectively.

**Questions:**

(1) Algorithm 1 looks costly due to the following reasons.

(1a) Eq. (15) trains $\pi _ { \hat{r} } ^ {\star}$ instead of $\hat{r}$. It seems that similar to algorithms that train $\hat{r}$, we also need to train and save two large models. What's the advantage?

(1b) Does line 4 require human annotation in every iteration? Some online DPO-type algorithms use advanced LLM to automatically annotate newly generated samples.

(2) Issues about the convergence bounds: The bounds in Theorem 1 contains $B$, whose last two constants contains algorithm generated variables $\pi_k$ and cannot be guaranteed small. The final term of $B^{\rm on}$ for Theorem 2 seems to go to $+\infty$ as $K\to+\infty$. Could you prove that it goes to 9 as $K\to+\infty$?

**I'd like to raise my rating if questions 1-2 can be solved well.**

(3) Right above Eq. (8), it seems that safe RLHF has access to only $\mathcal{D}^c$ but not $\mathcal{D}^r$?

(4) In the experiment, what evaluation model is used in model-based evaluation? Could you list the hyperparameters for the other algorithms?

(5) Optional: Do you think the primal algorithm [1] will work on constrained alignment problem, which does not require Lagrange multipliers?

[1] Xu, T., Liang, Y., \& Lan, G. (2021, July). Crpo: A new approach for safe reinforcement learning with convergence guarantee. In International Conference on Machine Learning (pp. 11480-11491). PMLR.

---

> ### Author Response · Authors · 2025-11-27
> **Response to Reviewer EtwF (Part 1/2)**
>
> Thank you very much for your insightful and valuable comments! We have revised our paper according to your comments and highlighted the revision in *orange* color.
>
> **Q(1a). Advantage of Our Approach**
>
> The main advantage of DPO compared to RLHF is to reduce the number of required trained models from 2 to 1 (i.e., avoid explicitly training the reward model), which greatly saves memory costs, and our work is motivated by this fact.
>
> **Our approach only needs to train and save 2 models**, i.e., the reward-aligned language model
>
> $
> \pi^*_{\hat{r}}
> $
>
> and reward-cost-aligned language model
>
> $
> \pi^*_{\hat{r}-\lambda\hat{c}}
> $
>
> , by using our rearranged primal-dual DPO objective Eq. (14), **while prior approaches [Dai et al., 2024; Liu et al., 2024b; Huang et al., 2024; Zhang et al., 2025] need to train and save 3 models**, i.e., the reward model $\hat{r}$, cost model $\hat{c}$ and reward-cost-aligned language model $\pi^*_{\hat{r}-\lambda\hat{c}}$. Thus, our approach significantly reduces memory costs.
>
>
> **Q(1b). Line 4 of Algorithm PD-DPO**
>
> For Line 4 in algorithm PD-DPO, we can use advanced LLMs to annotate cost binary feedback, instead of performing real human annotation, as done in the RLHF and DPO literature.
>
> **Q(2). Theorem 2**
>
> Thank you for pointing out this question. The $B^{on}$ term in Theorem 2 in our original submission **has a typo**. The term
> $$
> \sqrt{ |\mathcal{X}| |\mathcal{Y}| K \bigg(  \log \Big( \frac{ \gamma^{on} + |\mathcal{D}^r_1| + K }{|\mathcal{X}| |\mathcal{Y}| \gamma^{on}} \Big) + \frac{1}{C^{base}} \log \Big( \frac{ \gamma^{on} + |\mathcal{D}^r_1| + C^{base} K }{|\mathcal{X}| |\mathcal{Y}| \gamma^{on}} \Big) \bigg) }  \qquad Eq.(i)
> $$
> in $B^{on}$ should be
> $$
> \sqrt{ \frac{|\mathcal{X}| |\mathcal{Y}|}{K}  \bigg(  \log \Big( \frac{ \gamma^{on} + |\mathcal{D}^r_1| + K }{|\mathcal{X}| |\mathcal{Y}| \gamma^{on}} \Big) + \frac{1}{C^{base} K} \log \Big( \frac{ \gamma^{on} + |\mathcal{D}^r_1| + C^{base} K }{|\mathcal{X}| |\mathcal{Y}| \gamma^{on}} \Big) \bigg) } .
> $$
> It is because **we forgot to divide Eq.(i) by $K$ when presenting the final result**. Eq.(i) stands for the factor of the sum of exploration bonuses $b^r_k$ and $b^c_k$ over all iterations $k$, which is of order $\tilde{O}(\sqrt{K})$. When presenting the final result of the output policy
>
> $\pi^{out}_K:=unif(\pi_1,\dots,\pi_K),$
>
> we need to compute $\frac{1}{K}\sum_{k=1}^{K}(f(\pi^*)-f(\pi_k))$, which involves $\frac{1}{K} \sum_{k=1}^{K} b^r_k$ and $\frac{1}{K} \sum_{k=1}^{K} b^c_k$, and thus we need to use $\frac{1}{K} \cdot Eq.(i)$. After fixing this typo, our results in Theorem 2 are of order $\tilde{O}(\frac{1}{\sqrt{K}})$, which converge to zero when $K$ goes to infinity.
>
> Other than this typo, our proofs are all correct. We have fixed this typo in our revision.
>
> **Q(3). Safe RLHF**
>
> The safe RLHF approach [Dai et al., 2024] has access to both reward preference data $\mathcal{D}^r$ and cost preference data $\mathcal{D}^c$.
> Safe RLHF trains a reward model and a cost model on $\mathcal{D}^r$ and $\mathcal{D}^c$, respectively, and then fine-tunes the language model to maximize the Lagrangian function constituted by the learned reward and cost functions.
> We have added more clarification above Eq. (8) to avoid confusion in our revision.
>
> **Q(4). Experiments**
>
> In our experiments, for model-based evaluation, we use the open-source reward model Beaver-7b-unified-reward and cost model Beaver-7b-unified-cost [Dai et al., 2024], which was also used in the model-based evaluations in many prior works, e.g., [Dai et al., 2024; Huang et al., 2024; Zhang et al., 2025]. We have mentioned this in the caption of Figure 1 and Section 6 in our original submission.
>
> In evaluation, we directly access the open-resource models SFT (Alpaca-7b-reproduced) and Beaver-v3.0 [Dai et al., 2024] via their Hugging Face websites, and did not tune it by ourselves. Thus, their hyperparameters are the same as in their paper [Dai et al., 2024].
> For algorithm SafeDPO [Kim et al., 2025], its model is not released, and we ran their code with the same hyperparameters as in their paper. Following your suggestion, we have added the hyperparameters of the compared algorithms in Appendix B.
>
> **Q(5). Application of CRPO to Constrained LLM Alignment**
>
> Thank you for raising this interesting question. Yes, we think that algorithm CRPO [Xu et al., 2021] may be applied to constrained alignment problem in the RLHF manner, e.g., first train a reward model and a cost model, and then use CRPO to fine-tune a language model by replacing the reward and constraint value functions in CRPO by the trained reward and cost models, respectively. However, such application of CRPO still needs to train 3 models, while our algorithm only needs to train 2 models, which greatly saves memory costs.

---

> > ### Author Response · Authors · 2025-11-27
> > **Response to Reviewer EtwF (Part 2/2)**
> >
> > References:
> >
> > Josef Dai, Xuehai Pan, Ruiyang Sun, Jiaming Ji, Xinbo Xu, Mickel Liu, Yizhou Wang, and Yaodong
> > Yang. Safe RLHF: Safe reinforcement learning from human feedback. ICLR, 2024.
> >
> > Zixuan Liu, Xiaolin Sun, and Zizhan Zheng. Enhancing LLM safety via constrained direct preference
> > optimization. ICLR Workshop, 2024b.
> >
> > Xinmeng Huang, Shuo Li, Edgar Dobriban, Osbert Bastani, Hamed Hassani, and Dongsheng Ding.
> > One-shot safety alignment for large language models via optimal dualization. NeurIPS, 2024.
> >
> > Botong Zhang, Shuo Li, Ignacio Hounie, Osbert Bastani, Dongsheng Ding, and Alejandro Ribeiro.
> > Alignment of large language models with constrained learning. ICML Workshop, 2025.
> >
> > Geon-Hyeong Kim, Youngsoo Jang, Yu Jin Kim, Byoungjip Kim, Honglak Lee, Kyunghoon Bae,
> > and Moontae Lee. SafeDPO: A simple approach to direct preference optimization with enhanced
> > safety. arXiv, 2025.
> >
> > Tengyu Xu, Yingbin Liang, and Guanghui Lan. CRPO: A new approach for safe reinforcement learning with convergence guarantee. ICML, 2021.

---

> ### Comment · Reviewer_EtwF · 2025-11-27
> **Would like to raise to rating=6**
>
> Solved my concerns.
> I woud like to raise my rating to 6, but found there is no edit button for that.
>
> For Q2, the last 2 terms of B of Theorem 1 involves algorithm generated policies $\pi _ k$. Is there any justification for that? For example, $||\phi(x,y)|| _ {(\Sigma _ {\mathcal{D}^{\mathrm{c}}}+\gamma I)^{-1}}$ may be bounded for any x, y?
>
> Reviewer EtwF

---

> > ### Author Response · Authors · 2025-11-28
> > **Thank you for your willingness to raise your score and the answer to your question**
> >
> > Thank you very much for your willingness to raise your score!
> >
> > In general, let $B_{\phi}$ denote a universal upper bound that satisfies $\\|\phi(x,y)\\| \leq B_{\phi}$ for any $(x,y)$. Then, we have a universal upper bound $
> > \\|\phi(x,y)\\| (\Sigma_{\mathcal{D}^{\diamond}}+\gamma I)^{-1} \leq \frac{B_{\phi}}{\sqrt{\gamma}}
> > $ for any $(x,y)$, where $\diamond \in \\{r,c\\}$.
> > Thus, $\frac{1}{K} \sum_{k=1}^{K} \mathbb{E} ((x,y) \sim \mathcal{D}^p \times \pi_k) [\\|\phi(x,y)\\|   (\Sigma_{\mathcal{D}^{\diamond}}+\gamma I)^{-1} ]$ is bounded independent of $\pi_k$. (Due to the formula display issue of OpenReview, here $(\Sigma_{\mathcal{D}^{\diamond}}+\gamma I)^{-1}$ denotes the subscript of $\\|\cdot\\|$, and $((x,y) \sim \mathcal{D}^p \times \pi_k)$ denotes the subscript of $\mathbb{E}$.)

---

### Official Review · Reviewer_VPsQ · 2025-11-03

**Soundness:** 2
**Presentation:** 2
**Contribution:** 2
**Rating:** 2
**Confidence:** 2

**Summary:**

The paper studies a constrained alignment problem in the framework of direct preference optimization (DPO). The authors reduce this problem to a Lagrangian dual problem, where the Lagrangian maximizer is evaluated by minimizing a preference-based optimization objective function. A key idea is to introduce a preference-based presentation of a reward model. Based on this formulation, the authors propose a preference-based primal-dual algorithm: PD-DPO. Furthermore, the authors provide the optimality and constraint violation guarantees, both with and without the preference data coverage assumption. Finally, the authors conduct a Safe RLHF experiment to show effectiveness.

**Strengths:**

- The authors study a constrained alignment for LLMs based on preference data. This is an important direction for aligning LLMs with specific human values, since constrained generation is required in LLM applications.

- The authors present a method that allows us to solve the Lagrangian dual problem in the DPO style. A key feature is that the proposed training algorithm: PD-DPO doesn't require direct knowledge of reward and cost models.

- The authors provide iteration complexity guarantees of the proposed algorithm in terms of objective and constraint functions, both with and without the preference data coverage assumption.

**Weaknesses:**

- It would be helpful if the authors could have a table to compare the proposed method with previous preference-based methods in terms of model/algorithm assumptions and computational efficiency. For instance, the authors mention previous preference-based methods (Liu et al. (2024b); Huang et al. (2024); Zhang et al. (2025); Kim et al. (2025)) need to regenerate preference data.

- The main idea of the proposed method is the policy-based representation of the combined reward and cost function in Equation (12). However, this assumes the Bradley-Terry model for a mixed human preference. It is important to explain in what extent this assumptions is valid in practice.

- The proposed algorithm: PD-DPO utilizes the cost binary feedback from human annotators, which assumes an off-shell cost model. This type of labeling assumption is also used in previous works (Liu et al. (2024b); Huang et al. (2024)). It is useful to clarify their differences.

- The iteration complexity guarantees seem to be limited to theoretical interest, since it assumes optimization steps of PD-DPO are solved exactly.

- The data coverage assumption in Section 4.3 is strong, since it assumes coverage over polices for all iterations. The exploration bonus analysis in Section 5 assumes the Bradley-Terry model, which is not explicitly mentioned in the main paper.

- Another weakness of this paper is the limited experimental evaluation in terms of data sets, and baseline methods.

**Questions:**

Additional to suggestions in Weaknesses, below are some other questions.

- Notation c is abused in Section 4.1.

- Since reward and cost are unknown, how to determine bound constants in (17) and (18)?

- What is rho in Assumption 1? How to choose it based on preference data?

- How large is the data coverage-related constants in Theorems 1 and 2?

- Math writing should be improved.

---

> ### Author Response · Authors · 2025-11-27
> **Response to Reviewer VPsQ (Part 1/3)**
>
> Thank you very much for your time and effort in reviewing our paper! We have revised our paper according to your comments and highlighted the revision in *purple* color.
>
> **W1. Comparison with Prior Works using A Table**
>
> Thank you for this valuable suggestion, and we have added a comparison table in Appendix A in our revision.
> The following table compares the assumptions,  the number of required trained and loaded models, and theoretical guarantees on the output policy of our work and prior works on constrained LLM alignment.
>
> | Algorithms | Assumptions  | # The required trained and loaded models  | Theoretical guarantees of the output policy  |
> | ------------ | ------------ | ------------ | ------------ |
> | PD-DPO (ours)  |  (i) Bradley-Terry model; (ii) Slater’s condition | 2: the reward-aligned  language model, reward-cost-aligned language model | Suboptimality and constraint violation  | None |
> | Safe RLHF [Dai et al., 2024]  | Bradley-Terry model  | 3: the reward model, cost model, reward-cost-aligned language model  |  None  |
> | C-DPO [Liu et al., 2024b]  |  Bradley-Terry model |  3: the reward model, cost model, reward-cost-aligned model |  None  |
> | MoCAN, PeCAN [Huang et al., 2024] |  (i) Bradley-Terry model; (ii) Slater’s condition; (iii) The optimal policy is feasible under the used cost model | 3: the reward model (the reward-aligned language model), cost model (cost-aligned language model), reward-cost-aligned language model  | Suboptimality and constraint violation (reply on their Assumption (iii), which is hard to verify) |
> | CAID [Zhang et al., 2025] |  (i) Bradley-Terry model; (ii) Slater’s condition; (iii) Boundedness of the policy parameterization gap; (iv) Strong convexity of the dual function | 3: the reward model, cost model, reward-cost-aligned language model  | Suboptimality and constraint violation (focus on the primal-dual gap brought by policy parameterization)  |
> | SACPO [Wachi et al., 2024]  | (i) Bradley-Terry model; (ii) Slater’s condition; (iii) Knowledge of the optimal Lagrange multiplier  |  2: the reward-aligned model, reward-cost-aligned  language model  |  Suboptimality and constraint violation (have an unbounded term of the gap between the used and optimal Lagrange multipliers) |
> | SafeDPO [Kim et al., 2025] |  (i) Bradley-Terry model; (ii) For any prompt, there exists a safe response and the SFT model assigns a positive probablity to this safe response | 1: the reward-cost-aligned language model  | None  |
>
> The advantage of our algorithm is to **reduce the required trained and loaded models from 3 to 2 without requiring prior knowledge of the optimal Lagrange multiplier**, which significantly saves memory costs, and provide rigorous theoretical guarantees on the output policy. As shown in the table, compared to [Dai et al., 2024; Liu et al., 2024b; Huang et al., 2024; Zhang et al., 2025], we only need to train and load 2 models rather than 3, and greatly save memory costs. Compared to [Wachi et al., 2024] which also only needs to train 2 models, we do not require prior knowledge of the optimal Lagrange multiplier. While [Kim et al., 2025] only needs to train 1 model, their algorithm performs worse than our algorithm in experiments, and does not have theoretical guarantees on the output policy.
>
> Yes, [Liu et al., 2024b; Huang et al., 2024] need to regenerate preference data. But our main advantage is not on this point, and is instead on the reduction of the required trained and loaded models and rigorous theoretical guarantees on the output policy without extra assumptions.
>
> Since the algorithmic frameworks used in the compared algorithms are very different (using the RLHF/DPO framework, and performing dual update or not), it is unclear how to fairly analyze and compare computational efficiency for theses algorithms. We have revised our statement "reduces computational and memory costs" to "reduces memory costs" in revision.

---

> ### Author Response · Authors · 2025-11-27
> **Response to Reviewer VPsQ (Part 2/3)**
>
> **W2. The Bradley-Terry Model**
>
> The Bradley-Terry model is a standard and widely-used assumption to characterize human preference in the LLM alignment literature. It was used in the famous RLHF [Ouyang et al., 2022] and DPO [Rafailov et al., 2023] works. Actually, both the reward model training objective of RLHF and the DPO training objective are derived using the Bradley-Terry model. Most of prior works, e.g., [Dai et al., 2024; Liu et al., 2024b; Huang et al., 2024; Zhang et al., 2025; Wachi et al., 2024; Kim et al., 2025], also assumed the Bradley-Terry model.
>
> We agree that the Bradley-Terry model may not fully characterize complex human preference in practice. However, the algorithms derived from the Bradley-Terry model, e.g., the algorithms in prior works and our work, still perform well in practice as shown in experiments. In addition, the main idea of our approach, i.e.,  a rearranged primal-dual DPO objective utilizing a standard DPO-trained model, can also be applied to a general human preference model by combining with the formulation in the work $\Psi$PO [Azar et al., 2024], which uses a general preference-based utility function $\Psi: [0,1]\rightarrow \mathbb{R}$ to generalize the Bradley-Terry model.
>
> **W3. Cost Binary Feedback**
>
> [Liu et al., 2024b; Huang et al., 2024] did not use cost binary feedback.
> [Dai et al., 2024; Kim et al., 2025] used cost binary feedback. Our assumption on the generation of cost binary feedback is the same as that in [Dai et al., 2024; Kim et al., 2025], i.e., $\Pr[ Z(y)=1 | x ] = \frac{1}{1+\exp(-c^*(x,y))}$, where $Z(y)=1$ denotes that response $y$ is unsafe, and $c^*$ is the groundtruth cost function. The difference is that we assume that we can collect cost binary feedback online, while [Dai et al., 2024; Kim et al., 2025] use offline cost binary data.
>
> **W4. The Optimization Steps**
>
> It is common to consider that the optimization steps are solved exactly in RLHF/DPO theoretical analysis, in order to focus on analyzing the error brought by reward/cost function learning from preference data and the used algorithmic frameworks, e.g., primal-dual methods. Prior works, e.g., [Wachi et al., 2024; Huang et al., 2024; Zhang et al., 2025], also considered that the optimization steps on policies are solved exactly in their analysis.
>
> We can also incorporate the error in the optimization steps into our theoretical results, and express our results with this error.
>
> **W5. Data Coverage and the Bradley-Terry Model**
>
> We agree that the data coverage terms in Theorem 1 in Section 4.3 can be large. Thus, we design an online exploration version of algorithm PD-DPO equipped with exploration bonuses to remove the  data coverage dependence in Theorem 2 in Section 5.
>
> We have stated that we assume the Bradley-Terry model in the Preliminaries Section (Section 3) in our original submission, which also applies to Section 5. However, following your suggestion, we have added clarification on this point in Section 5 in our revision.
>
> **W6. Experiments**
>
> Please see our general response to all reviewers.
>
>
> **Q1. Abused Notation $c$**
>
> Thank you for pointing out this question. $c(x,y)$ denotes the cost function. The superscript c stands for the quantities related to cost preference. We have revised the superscript c using the text font, instead of the math font, to avoid confusion in our revision.
>
> **Q2. Constants $R_{\max},C_{\max}$ in Eqs. (17) and (18)**
>
> The reward and cost ranges $r \in [-R_{\max},R_{\max}]$ and $c \in [-C_{\max},C_{\max}]$ in Eqs. (17) and (18) are a design for the theoretical algorithm in order to derive rigorous guarantees on the output policy. In practical implementation, we do not impose the reward and cost range constraints, and instead directly perform optimization on the parameters of language models using our rearranged primal-dual DPO objective Eq. (14). We have mentioned this point in Appendix B (implementation details) in our original submission.
>
> **Q3. $\rho$ in Assumption 1**
>
> $\rho$ is a parameter representing the ratio of the suboptimality of a feasible policy to its feasibility margin to the constraint threshold. $\rho$ serves as an upper bound on the optimal Lagrange multiplier, which is often used in constrained optimization analysis [Beck, 2017]. We need $\rho$ only in the theoretical algorithm to keep the updated Lagrange multiplier upper bounded. In practial implementation, we just create an optimizer for the Lagrange multiplier and perform subgradient descent, and we do not project it to $[0,\rho]$ at every step.

---

> ### Author Response · Authors · 2025-11-27
> **Response to Reviewer VPsQ (Part 3/3)**
>
> **Q4. Data Coverage-related Constants**
>
> In Theorem 1, the data coverage-related constants can be large, since we do not consider online exploration in Section 4. In Theorem 2, there is no data coverage-related constant, since in Section 5, we design an online exploration version of  algorithm PD-DPO, which employs exploration bonuses in the rearranged primal-dual DPO objective to enable exploration and removes the data coverage dependence in the results.
>
> **Q5. Math Writing**
>
> Thank you for your suggestion. We have revised the superscripts $r$ and $c$ to r and c using the text font, respectively, to avoid confusion in our revision.
>
> ---
>
> References:
>
> Josef Dai, Xuehai Pan, Ruiyang Sun, Jiaming Ji, Xinbo Xu, Mickel Liu, Yizhou Wang, and Yaodong
> Yang. Safe RLHF: Safe reinforcement learning from human feedback. ICLR, 2024.
>
> Zixuan Liu, Xiaolin Sun, and Zizhan Zheng. Enhancing LLM safety via constrained direct preference
> optimization. ICLR Workshop, 2024b.
>
> Xinmeng Huang, Shuo Li, Edgar Dobriban, Osbert Bastani, Hamed Hassani, and Dongsheng Ding.
> One-shot safety alignment for large language models via optimal dualization. NeurIPS, 2024.
>
> Botong Zhang, Shuo Li, Ignacio Hounie, Osbert Bastani, Dongsheng Ding, and Alejandro Ribeiro.
> Alignment of large language models with constrained learning. ICML Workshop, 2025.
>
> Akifumi Wachi, Thien Tran, Rei Sato, Takumi Tanabe, and Youhei Akimoto. Stepwise alignment for
> constrained language model policy optimization. NeurIPS, 2024.
>
> Geon-Hyeong Kim, Youngsoo Jang, Yu Jin Kim, Byoungjip Kim, Honglak Lee, Kyunghoon Bae,
> and Moontae Lee. SafeDPO: A simple approach to direct preference optimization with enhanced
> safety. arXiv, 2025.
>
> Long Ouyang et al. Training language models to follow
> instructions with human feedback. NeurIPS, 2022.
>
> Rafael Rafailov, Archit Sharma, Eric Mitchell, Christopher D Manning, Stefano Ermon, and Chelsea
> Finn. Direct preference optimization: Your language model is secretly a reward model. NeurIPS, 2023.
>
> Mohammad Gheshlaghi Azar, Zhaohan Daniel Guo, Bilal Piot, Remi Munos, Mark Rowland, Michal
> Valko, and Daniele Calandriello. A general theoretical paradigm to understand learning from
> human preferences. AISTATS, 2024.
>
> Amir Beck. First-order methods in optimization. SIAM, 2017.

---

### Author Response · Authors · 2025-11-27
**General Response on Experiments to All Reviewers**

According the reviewers' suggestion, we added more baselines SACPO and P-SACPO [Wachi et al., 2024] and ran our algorithm with different values of the Lagrange multiplier ($\lambda$) to show the Pareto front. During rebuttal, we  updated our hyper-parameters to align with the DPO hyper-parameters in [Dai et al., 2024], and we observed that the empirical performance significantly improved. We have listed the hyper-parameters of all compared algorithms in Appendix B in our revision.

We have included all released models of related work to the best of our knowledge ([Huang et al., 2024; Zhang et al., 2025] did not release their models, and it takes time to reproduce their algorithms). Below we report preliminary results for model-based evaluation, i.e., the rewards and negated costs of the responses generated by compared language models when evaluated by Beaver-7b-unified-reward and Beaver-7b-unified-cost [Dai et al., 2024].

| Algorithm | Helpfulness (Reward) | Harmlessness (Negated Cost) |
| ------- | ------- | ------- |
|    SFT (Alpaca-7b-reproduced)     |    -1.0648     |    -6.7657     |
| SafeDPO [Kim et al., 2025] | -0.8342 | -5.8312 |
| SACPO [Wachi et al., 2024] | -0.0845 | 9.6723 |
| P-SACPO [Wachi et al., 2024] | 0.5574 | 8.8926 |
| Beaver-v3.0 [Dai et al., 2024] | 1.0453 | 16.3837 |
| PD-DPO ($\lambda$=1) | 3.8082 |  -2.8584 |
| PD-DPO ($\lambda$=2) | 1.3692 | 6.1695 |
| **PD-DPO ($\lambda$=3)** | **0.1701** |  **9.6908** |
| PD-DPO ($\lambda$=5) | -0.8847 | 12.7177 |

From the table, we see that our PD-DPO  ($\lambda=3$) outperforms SFT, Safe-DPO [Kim et al., 2025] and SACPO [Wachi et al., 2024] in both harmlessness and helpfulness.
The performance of PD-DPO ($\lambda=3$) is comparable to P-SACPO [Wachi et al., 2024]. However, PD-DPO does not require prior knowledge of the optimal Lagrange multiplier as in SACPO and P-SACPO.
While PD-DPO has worse performance than Beaver-v3.0 [Dai et al., 2024], PD-DPO only needs to train 2 models rather than 3 models as in Beaver-v3.0. In addition, Beaver-v3.0 requires much more memory costs than PD-DPO (cannot be run on a single GH200 GPU with 96GB memory), and  does not have rigorous theoretical guarantees as PD-DPO. This trade-off between performance and memory costs is similar to the trade-offs between DPO and RLHF that have been reported in the literature.

We have incorporated these results using a figure in Section 6 in our revision.
We are running more experiments, including GPT-evaluation, the dynamics of $\lambda$, reproduction of [Huang et al., 2024; Zhang et al., 2025], and  ablation study on the constraint threshold and reward-aligned model $\pi^*_{\hat{r}}$. We will add these experiments to our revision once they finish.

---

References:

Josef Dai, Xuehai Pan, Ruiyang Sun, Jiaming Ji, Xinbo Xu, Mickel Liu, Yizhou Wang, and Yaodong
Yang. Safe RLHF: Safe reinforcement learning from human feedback. ICLR, 2024.

Xinmeng Huang, Shuo Li, Edgar Dobriban, Osbert Bastani, Hamed Hassani, and Dongsheng Ding.
One-shot safety alignment for large language models via optimal dualization. NeurIPS, 2024.

Botong Zhang, Shuo Li, Ignacio Hounie, Osbert Bastani, Dongsheng Ding, and Alejandro Ribeiro.
Alignment of large language models with constrained learning. ICML Workshop, 2025.

Akifumi Wachi, Thien Tran, Rei Sato, Takumi Tanabe, and Youhei Akimoto. Stepwise alignment for
constrained language model policy optimization. NeurIPS, 2024.

Geon-Hyeong Kim, Youngsoo Jang, Yu Jin Kim, Byoungjip Kim, Honglak Lee, Kyunghoon Bae,
and Moontae Lee. SafeDPO: A simple approach to direct preference optimization with enhanced
safety. arXiv, 2025.

---

### Meta-Review · Area_Chair_qdxv · 2025-12-31

**Summary:**

This paper proposes PD-DPO, a primal–dual direct preference optimization method for constrained LLM alignment that aims to avoid explicit reward/cost model training (reducing “3-model” pipelines to 2 models) while providing theoretical guarantees on suboptimality and constraint violation. Reviewer EtwF and kvKV find the formulation promising and the theoretical framing relevant, but the overall panel is split: VPsQ and QQdF remain unconvinced due to strong assumptions and limited/unclear empirical validation of the full primal–dual procedure and its practicality. The rebuttal adds missing baselines and a Pareto-style sweep over $\lambda$, clarifies assumptions, and fixes at least one stated theoretical typo, improving the paper but leaving some key deployment concerns unresolved.

**Reviewer Concerns:**

Concerns that were substantially addressed include: (i) missing baseline coverage and tradeoff reporting (authors added SACPO/P-SACPO and multiple $\lambda$ settings / Pareto front), (ii) requests for clearer comparison to prior constrained-alignment work (authors added a comparison table and clarified the “2 vs 3 models” claim), and (iii) specific technical questions/typos in theory (notably EtwF’s concern about Theorem 2, which the authors attribute to a typo and correct in the revision; plus clarifications about Bradley–Terry usage and notation raised by VPsQ).

Concerns that remain outstanding are: (a) practicality of the dual update / cost estimation in real training loops (human-in-the-loop cost labels, variance, and the “theory vs implementation” gap flagged by QQdF and EtwF), (b) novelty positioning relative to Huang et al. / Zhang et al. beyond the “no explicit cost/reward model” angle (raised by kvKV), and (c) empirical validation completeness, especially demonstrating the actual primal–dual dynamics (trajectories of $\lambda$, reward/cost over iterations) and online/exploration variant results (requested by QQdF, kvKV, and VPsQ; authors indicate some are “running” but not yet evidenced in the discussion).

**Reviewer Scores:**

EtwF (4): explicitly states their concerns are solved and they would raise to 6 if they could edit, contingent mainly on minor clarification about boundedness terms in Theorem 1 (which the authors addressed with a boundedness argument).

kvKV (4): likely shifts modestly upward (e.g., 4 → 4/6) given the added baselines/Pareto sweep and expanded discussion of differences from prior work, but may remain cautious because the “memory savings” narrative and empirical constraint-satisfaction evidence are still debated.

VPsQ (2): may move slightly (e.g., 2 → 2/4) after the added comparison table, clarified Bradley–Terry assumptions, and expanded experimental baseline set; however, they flagged broad issues (strong assumptions, exact-solve steps, data coverage strength, limited experiments) that are only partially mitigated.

QQdF (2): likely remains 2 (or at most marginally higher) because their central critique is that the empirical section does not convincingly validate the full primal–dual algorithm (dual update and dynamics) and the proposed cost-estimation + exploration mechanisms appear impractical as stated; the rebuttal acknowledges these issues but largely defers full evidence.

---

### Decision · Program_Chairs · 2026-01-26

Reject